

# Inelastic compaction and permeability evolution in volcanic rock

Jamie I. Farquharson, Patrick Baud, Michael J. Heap

Institut de Physique de Globe de Strasbourg (UMR 7516 CNRS), EOST, Université de Strasbourg, France

*Correspondence to*: Jamie I. Farquharson (farquharson@unistra.fr)

**Abstract.**

Active volcanoes are mechanically dynamic environments, and edifice-forming material may often be subjected to
significant amounts of stress and strain. It is understood that porous volcanic rock can compact inelastically under a wide
range of *in situ* conditions. In this contribution, we explore the evolution of porosity and permeability – critical properties
influencing the style and magnitude of volcanic activity – as a function of inelastic compaction of porous andesite under
triaxial conditions. Progressive strain accumulation is associated with progressive porosity loss. The efficiency of
compaction was found to be related to the effective confining pressure under which deformation occurred: at higher effective
pressure, more porosity was lost for any given amount of strain. Permeability evolution is more complex, with small
amounts of stress-induced compaction (< 0.05, i.e. less than 5 % reduction in sample length) yielding an increase in
permeability under all effective pressures tested, occasionally by almost one order of magnitude. This phenomenon is
considered here to be the result of improved connectivity of formerly isolated porosity during triaxial loading. This effect is
then overshadowed by a decrease in permeability with further inelastic strain accumulation, especially notable at high strains
(> 0.20) where samples may undergo a reduction in permeability by two orders of magnitude relative to their initial values.
A physical limit to compaction is discussed, which we suggest is echoed in a limit to the potential for permeability reduction
in compacting volcanic rock. Compiled literature data illustrate that at high strain (both in the brittle and ductile regimes),
porosity $\phi$ and permeability $k$ tend to converge towards intermediate values (i.e. $0.10 \leq \phi \leq 0.20$; $10^{-14} \leq k \leq 10^{-13}$ m²). These
results are discussed in light of their potential ramifications for impacting edifice outgassing  – and in turn, eruptive activity
– at active volcanoes.

## 1 Introduction

Active volcanoes are inherently high-strain environments (e.g. Ōmori, 1920; Mogi, 1958; Dzurisin, 2003). Magma is a
constantly developing multi-phase medium, and as it migrates through the crust, vesiculating and crystallising along the way,
it can impart significant mechanical stress on the surrounding edifice rock (e.g. Sparks, 1997; Voight *et al*., 1998; Denlinger
and Hoblitt, 1999; Clarke *et al*., 2007; Heimisson *et al*., 2015). Deeper in the crust, magma chamber deformation can
pressurise conduit and dyke systems above, in turn displacing the edifice (e.g. Melnik and Sparks, 1999; Melnik and Sparks,
2005; Wadge *et al*., 2006). Much of the edifice is likely to be fluid-saturated (e.g. Day, 1996; Delcamp *et al*., 2016), a
function of permeability, permeability anisotropy, and heat flux (Hurwitz *et al*., 2003; Finn *et al*., 2007): migration of fluids
can serve to adjust the *in situ* stress conditions and influence the short-term failure strength of edifice-forming rocks (e.g.
Farquharson *et al*., 2016a). Moreover, the continued loading of the edifice due to iterative emplacement of erupted material
will serve to increase the overburden (i.e. the confining pressure) in any given region of the edifice (e.g. Heap *et al*. 2015a).
It is generally understood that all volcanoes deform under their own weight to some extent (Shteynberg and Solov'yev, 1976;
Dieterich, 1988; van Wyk de Vries and Borgia, 1996), driven largely by gravitational spreading and substratum flexure (van
Wyk de Vries and Matela, 1998), which indicates that the local and far-field stresses operative in a volcanic edifice will



evolve over time. Further, the overlying stress may be influenced by loading and unloading of volcanoes by ice, for example, either seasonally or due to longer-scale climatic variations (e.g. Sigmundsson *et al.*, 2010).

Thus, edifice-forming volcanic rock may be subjected to a range of stress and strain conditions over time. This is an
important consideration, as the response of rock to imposed stress can have a significant impact on its fluid transport properties. In particular, permeability – the capacity of a material to transmit fluids through interconnected pore space – is a critical property in the context of volatile loss and pressure dissipation. The expansion of exsolved gas species which occurs as volatile-laden magma approaches the surface of the Earth can generate overpressures in the magma: the kinetic engine that typically drives explosive fragmentation (e.g. Sparks, 1978). If the permeability of a volcanic system (including the
edifice) is high, volatiles may be effectively siphoned out of the magma and the propensity for explosive behaviour reduced; on the other hand, low system permeability could promote pressure build-up and violent eruptive activity (e.g. Eichelberger et al., 1986, Woods and Koyaguchi, 1994, Rust et al., 2004, Edmonds and Herd, 2007, Mueller et al., 2008, Mueller et al., 2011, Nguyen et al., 2014, Castro et al., 2014, Okumura and Sasaki, 2014; Gaunt et al., 2014).

Volcanic rock can either dilate or compact in response to an applied differential stress or a change in pore pressure. Dilatant brittle failure comprises an increase in porosity through the formation, growth, and coalescence of stress-induced fractures (e.g. Fortin *et al.*, 2011). In compression, this culminates in an axial split or shear fracture. On the other hand, an overall reduction in porosity can be brought about by homogeneous or localised inelastic compaction. In Yakuno basalt, Shimada *et al.* (1989) showed that ductile deformation was associated with distributed microcracking, granulation, and pore occlusion.
The underlying physical mechanism – cataclastic pore collapse – is described in detail by Zhu *et al.* (2010), and has since been interpreted in triaxially deformed tuff (Zhu *et al.*, 2011; Alam *et al.*, 2014; Heap *et al.*, 2015b), basalt (Adelinet *et al.*, 2013; Zhu *et al.*, 2016), andesite (Heap *et al.*, 2015a), trachyandesite (Loaiza *et al.* 2012), and dacite (Heap *et al.*, 2016). This micromechanism can be distributed (e.g. Zhu *et al.*, 2011; Heap *et al.*, 2015b) or localised (e.g. Loaiza *et al.*, 2012; Adelinet *et al.*, 2013; Heap *et al.* 2015a) in the form of bands of compacted pores. The predisposition for strain localisation
in volcanic rock has been suggested as a function of stress nucleation around equant porosity (Adelinet *et al.*, 2013), due to preferential alignment of pores (Loaiza *et al.*, 2013), or along pre-existing planes of weakness such as zones of amœboid pores or microfractures (Heap *et al.*, 2015a).

While manifestations of inelastic compaction tend not to be immediately obvious in the field (unlike fractures, which are
ubiquitous at all scales in volcanic environments), abundant indirect evidence for this process exists. Bulk rock density (and reciprocally, rock porosity) has been estimated during scientific drilling projects – in concert with gravimetric and other geophysical data – at, for example, Campi Flegrei, Italy (Barberi *et al.*, 1991) and Mount Unzen (Sakuma *et al.*, 2008). These studies highlight a predictable decrease in porosity with increasing depth, supported in either case by a general increase in ultrasonic wave velocities with increasing depth, indicative of a reduction in porosity with increasing lithostatic
pressure. This observation is borne out by experimental deformation studies, which show that the propensity for compactant – rather than dilatant – behaviour of volcanic rock is intrinsically linked to the confining pressure under which the sample is deformed (e.g. Shimada *et al.*, 1989; Heap *et al.*, 2015a; Zhu *et al.*, 2016), as well as being heavily reliant on its initial porosity (Heap *et al.*, 2015a; Zhu *et al.*, 2016) and other factors such as temperature (Violay *et al.*, 2012) and alteration (Siratovich *et al.*, 2016). In detail, high effective pressures and/or high initial porosity promote ductile behaviour, whereas
dilatant brittle failure is favoured in low-porosity volcanic rock deformed under a range of effective pressures.

While pre-failure permeability has been explored in plutonic (Zoback and Byerlee, 1975; Kiyama *et al.*, 1996; Mitchell and Faulkner, 2008) and volcanic (Faoro *et al.*, 2013) rocks, studies of post-failure permeability change have been generally



limited to investigations into sedimentary and synthetic materials (e.g. Mordecai and Morris, 1970; Peach and Spiers, 1996; Regnet *et al.*, 2015). However, a recent study (Farquharson *et al.*, 2016b) explored the influence of brittle failure and progressive stress-induced dilation of low- to intermediate-porosity volcanic rock. Permeability was found to increase with ongoing strain accumulation under triaxial conditions. With regards to the influence of inelastic compaction, research has

yielded both decreases (Zhu and Wong, 1997; Baud *et al.*, 2012) and increases (Xiaochun *et al.*, 2003) in the permeability of porous sandstone. Alam *et al.* (2014) investigated the permeability evolution of welded tuff from Shikotsu (Hokkaidō Prefecture, Japan), finding that permeability decreased monotonously with triaxial compression (both in the dilation and compaction regimes) and that the rate of permeability decrease was tied to the effective pressure under which deformation was performed. Pilot experiments on porous andesite – described in Heap *et al.* (2015a) – also indicate permeability loss as a

result of inelastic compaction. Building on the work of these studies, this contribution investigates the response of the physical properties of volcanic rock – i.e. porosity and permeability – as a function of inelastic compaction under conditions anticipated in volcanic environments. We then expound these results in light of the potential influence of edifice rock compaction on volcanic activity.

## 2 Materials and methods

### 2.1 Sample preparation and deformation

To assess the influence of inelastic compaction on volcanic rock permeability, a porous andesite from Volcán de Colima (Mexico) was used. The construction history, geomorphology and eruptive style of Volcán de Colima make it a useful analogue for other active andesitic stratovolcanoes around the world, such as Gunung Merapi (Indonesia), Ruapehu (New Zealand), Volcán Rincon de la Vieja (Costa Rica), Santa María (Guatemala), Tungurahua (Ecuador) and many, many more.

Core samples were prepared from a block of andesite approximately 1 m³, collected in May 2014 from the La Lumbre debris-flow track (*barranca*) on the south-western flank of the volcano. The andesite – "LLB" – is a vesicular porphyritic andesite containing subhedral phenocrysts and microphenocrysts, of unknown age. Bulk geochemical analysis is given in Table 1.

This andesite was chosen because its relatively high initial connected porosity $\phi$ means that it can be deformed in the ductile regime under pressure conditions relevant to a volcanic edifice (and easily attainable in the laboratory: Heap *et al.*, 2015a). Ten sample cores were prepared with a diameter of 20 mm, and were ground flat and parallel to a nominal length of 40 mm. Samples were dried in a vacuum oven for at least 48 hours, and the following steps carried out (adopting the protocol of Farquharson *et al.* 2016b):

1.  Physical properties (porosity, permeability) were measured,
        2.  Samples were saturated, then deformed triaxially in compression under a set effective pressure to a given degree of axial strain,
        3.  Samples were unloaded, dried for 48 hours, and their permeability was re-measured.

Each of these stages are described in more detail hereafter. Helium pycnometry was used to measure the bulk and powder

densities of LLB samples ($\rho_b$ and $\rho_p$, respectively), whilst measurements of sample dimensions allow the calculation of the volumetric mass density $\rho_v$. In turn, porosity (connected $\phi$, total $\phi_t$, and unconnected $\phi_u$) can be calculated:

$$\phi = \left( \frac{\rho_b - \rho_v}{\rho_v} \right) \quad (1.1)$$
$$\phi_t = 1 - \left( \frac{\rho_b}{\rho_p} \right) \quad (1.2) \quad\quad\quad\quad (1)$$
$$\phi_u = \phi_t - \phi \quad (1.3),$$



where a mean value of 2653 kg m⁻³ is used for $\rho_p$. Gas permeability was measured under steady-state conditions with a confining pressure of 1 MPa using the setup described in Farquharson *et al.* (2016c). Where necessary, a correction was applied to the measured permeability values to account for turbulent flow (see Forchheimer, 1901), the effects of which can become non-negligible when measuring the permeability of high-porosity media.

Samples were then encased in a copper foil jacket (which serves to retain bulk sample cohesion after deformation), saturated with distilled water, and loaded into the triaxial deformation rig at Université de Strasbourg (see Fig. 1). Throughout deformation we assume a simple effective stress law, whereby the effective confining pressure $p_{eff}$ experienced by a sample is a function of the confining pressure $p_c$ around the sample and the pressure of pore fluid $p_p$ within the sample, such that $p_{eff} = p_c - \alpha \cdot p_p$. Recent experimental work (Farquharson *et al.*, 2016a) shows that $\alpha = 1$ is a reasonable assumption for

porous andesite.

For each test, the confining and pore pressures were increased slowly until a targeted effective pressure (i.e. hydrostatic pressurisation). Assuming a pycnometry-derived value for bulk density $\rho_b$ of approximately 2100 kg m⁻³, the imposed effective pressures of 10, 30, 50, and 70 MPa are analogous to depths ranging from the upper 500 m of the edifice to greater

than 3 km in depth (given that $p_c \propto \rho_b \cdot g z$, where $g$ and $z$ are surface gravitational acceleration and depth, respectively). The sample would then be left overnight to allow microstructural equilibrium. During the deformation experiments, a differential stress was introduced in the direction of the sample axis by advancing an axial piston (see Fig. 1) under servo-control, such that the sample is subjected to a constant strain rate of 10⁻⁵ s⁻¹. Confining pressure and pore pressure were servo-controlled throughout.  During hydrostatic and nonhydrostatic loading, the response of the pore fluid pump reflects

variations in pore volume, which – normalised to the initial sample volume – corresponds to the porosity change $\delta\phi$. For a porous material, $\delta\phi$ can be considered equal to the volumetric strain. Indeed, the response of the confining pressure pump provides an independent estimation of the volumetric strain, found to be in perfect agreement with the inferred $\delta\phi$ (see Baud et al., 2014 for details). When this differential $\delta\phi$ is positive it signifies dilation (an increase in porosity) and when it is negative, it indicates compaction (a decrease in porosity). Deformation was allowed to continue for different amounts of

axial strain accumulation – sample shortening relative to its original length ($\varepsilon_t$) – then unloaded. The strain recovered during the unloading phase is subtracted from the total axial strain $\varepsilon_t$ to give the inelastic (non-recoverable) strain accrued by the sample ($\varepsilon_i$). Similarly, the elastic porosity change recovered during unloading is subtracted from the porosity change at $\varepsilon_t$, to give the inelastic porosity change $\delta\phi_i$. Samples were subsequently vacuum-dried once again and gas permeability re-measured.


### 2.2 Post-deformation permeability

It has been shown in recent studies (e.g. Nara *et al.*, 2011) that the permeability of volcanic materials is influenced by the effective pressure under which it is measured: permeability tends to decrease with increasing effective pressure. As such, we acknowledge the limitation that post-deformation measurements do not represent the permeability under the deformation

conditions *sensu stricto*. Nevertheless, we choose to measure permeability under the conditions described above for a host of reasons. Investigations towards determining poroelastic constant $\alpha$ for properties (including permeability) other than rock strength indicate that this coefficient may differ as a function of porosity, pore geometry, and other factors, which is to say that $\alpha$ for one property may not be the same as $\alpha$ for a second property (Bernabé, 1986). Given the lack of constraint on $\alpha$ for the permeability of volcanic rocks, permeability is measured at the lowest possible confining pressure (1 MPa, rather than

at "*in situ*" pressures) and without imposing a differential stress, in order to allow comparison within and between sample sets (indeed, we compare our data with compiled literature data in Sect. 4.1). This procedure also avoids the potential for creep – a mechanism of time-dependent deformation whereby sub-critical crack growth induces damage and possibly even



failure at stresses below the short-term strength of the rock (*e.g.* Brantut *et al.*, 2013; Heap *et al.*, 2015c) – as well as precluding other phenomena such as stress relaxation that may arise when measuring permeability under triaxial conditions.

Measuring permeability requires that the sample dimensions, specifically length and cross-sectional area, are be well constrained. Prior to initial measurements of permeability, sample dimensions are measured accurately using digital callipers. However, after mechanical deformation samples are often barrelled and thus non-cylindrical, making their mean radii nontrivial to determine. Assuming that the solid volume $V_s$ remains constant throughout deformation, then the post-deformation volume is equal to the sum of solid volume, the initial porosity, and the pore volume change after deformation. The post-deformation cross-sectional area $A_{post}$ can therefore be determined such that

$$A_{post} = \left( \frac{V_s}{1 - [\phi + \delta\phi_i]} \right) \cdot l_{post}^{-1}, \tag{2}$$

where $l_{post}$ is the mean sample length after deformation.

## 3 Results

Table 2 gives the deformation conditions ($p_{eff}$, $\varepsilon_t$, $\varepsilon_i$) for each test, as well as pre- and post-deformation values of porosity and permeability. Mechanical data for all experiments – performed under a range of effective pressures to differing amounts of strain – is shown in Fig. 2a, plotted as differential stress against axial strain. In each case, the stress-strain curve is concave upwards in the initial phase of sample loading (1 in Fig. 2a *inset*), which is followed by a period of linear elastic behaviour (2). Beyond a critical stress state (3), termed $C^*$ (Wong *et al.*, 1997), the sample is no longer deforming poroelastically (i.e. additional differential stress causes inelastic compaction: this is known as "shear-enhanced" compaction). This threshold – the compactive yield stress – signals the onset of shear-enhanced compaction. Thereafter (4), the material may continue to accommodate approximately the same amount of stress, or accumulate additional stress (a phenomenon known as strain-hardening), where the stress-strain curve tends upwards post-failure and the sample strengthens with increasing strain. In many of the samples, the stress-strain curve is variably interposed by stress drops. This compactant behaviour is illustrated in Fig. 2b: porosity tends to decrease monotonously with increasing axial strain, and the net porosity change is always negative. The trend of progressive compaction differs in one experiment (LLB-13): after a threshold strain the trajectory of the porosity change curve becomes positive (indicating dilation). This phenomenon is discussed in Sect. 4.2.

Figure 3a shows the inelastic change in sample porosity as a function of inelastic axial strain. The initial samples contained connected porosities ranging between 0.19 and 0.23 (see Table 2), which invariably decreased after accumulating strain in the ductile regime. The minimum porosity change was -0.005, for a sample deformed under an effective pressure of 10 MPa to an inelastic strain of 0.006 (a 0.6 % reduction in sample length). The maximum change in porosity was -0.074, for a sample deformed at an effective pressure of 50 MPa to an inelastic strain of 0.174 (i.e. a 17.4 % reduction in sample length). For a given amount of inelastic strain, the volume of porosity lost through compaction is dependent on the effective pressure: more efficient compaction is evident at higher effective pressures. Figure 3b shows post-deformation permeability $k_e$ as a function of inelastic axial strain. At low strains (< 0.05), sample permeability tends to increase, from around $5 \times 10^{-13}$ m$^2$ to as high as $3.71 \times 10^{-12}$ m$^2$ (almost an order of magnitude). This behaviour overlies a general trend of permeability decrease with increasing strain: at higher strains (0.06 - 0.24) permeability can decrease relative to its original value by as much as two orders of magnitude. The largest decrease (sample LLB-13) was from $4.84 \times 10^{-12}$ m$^2$ to $5.51 \times 10^{-14}$ m$^2$ after an inelastic strain accumulation of 0.233.



## 4 Discussion

### 4.1 Microstructural controls on permeability evolution

The underlying micromechanical mechanism driving inelastic compaction in volcanic rocks has been shown to be cataclastic pore collapse (Zhu *et al.*, 2011; Heap *et al.*, 2015a; Zhu *et al.*, 2016). Figure 4 illustrates this process by showing images of

an intact and a deformed sample. Figure 4a is a backscattered scanning electron microscope image of an as-collected sample of LLB andesite, whereas the images in Fig. 4b and 4c (from the same sample suite) are of a samples that has accumulated high strain (> 0.20) under an effective pressure of 30 MPa. The undeformed sample (Fig. 4a) is pervasively microcracked, with highly amœboid pores ranging from < 10 μm to around 80 μm in diameter. Cataclastic pore collapse involves intense microcracking, which develops in a concentric damage zone around a pore. As the process of cataclasis – progressive

fracturing and comminution – continues, fragments can spall into the void space, thus reducing porosity (Zhu *et al.*, 2010). Figure 4b clearly shows abundant fractures created during triaxial deformation, both within the groundmass and crystals. In many areas, fragments have been comminuted to the micron-scale. In these samples, as observed in previous experimental studies of volcanic rock (Loaiza *et al.*, 2012; Adelinet *et al.*, 2013; Heap *et al*. 2015a), cataclastic pore collapse is localised in the form of bands traversing the sample. The occurrence of these bands has been shown to correspond to periodic stress

drops (Heap *et al.*, 2015a), which are abundant in the mechanical data of Fig. 2a.

Our experimental data (Fig. 2b and Fig. 3a) shows that cataclastic pore collapse progressively reduces the porosity of these andesites. We note that porosity loss is seemingly tied to the effective pressure under which compaction occurs (as observed in previous studies concerned with triaxial rock deformation, e.g. Wong *et al.*, 1997; Baud *et al.*, 2006; Heap *et al.*, 2015a):

for a given amount of strain, the porosity lost by a sample is always greater at a higher effective pressure. This phenomenon is true both for total porosity change (Fig. 2b) and for inelastic porosity loss (Fig. 3a), which is to say that the inelastic compaction factor $\delta\phi_i/\varepsilon_i$ always decreases as effective pressure increases (Baud *et al.*, 2006). While the mechanism of cataclastic pore collapse is governed by the pore sizes (e.g. Zhu *et al.*, 2010; 2011), it has also been demonstrated that the local stress field around a pore increases as a function of the incumbent confining pressure (Zhu *et al.*, 2010). A classic study

of fault gouge formation in sandstone (Engelder, 1974) shows that fault-zone fragments are smaller when generated at higher confining pressures, and a similar effect (albeit less pronounced) was noted by Kennedy and Russell (2012), who investigated fault gouge formation in dacitic dome rock. We suggest that cataclasis may become more efficient as the local stress field increases in line with the confining pressure; in turn, a finer distribution of fragments will more readily occlude the pores around which they develop. Whether a change in the mean fragment size generated during cataclastic pore collapse

underlies the observed evolution of $\delta\phi_i/\varepsilon_i$ remains open to a targeted microstructural study.

As would be expected (e.g. Zhu and Wong, 1997), permeability reduction follows the same general trend (Fig. 3b) as porosity reduction (Fig. 3a), with samples accumulating high strains showing a correspondingly large reduction in permeability. However at low strains, there is no one-to-one relationship between permeability and porosity after

deformation. Rather, permeability tends to increase moderately at inelastic strains less than around 0.05 (i.e. a 5 % shortening in sample length). A similar phenomenon was also observed by Loaiza *et al.* (2012), who noted that permeability of Açores trachyandesite increased beyond a critical stress state during hydrostatic pressurisation. This critical stress (known as $P^*$: Zhang *et al.* 1990) signals the onset of lithostatic inelastic compaction; Loaiza *et al.* (2012) show that stress-induced cracks coalesce between collapsed pores during hydrostatic compaction, improving connectivity and, in turn, increasing

permeability. Prior to deformation, the samples of porous andesite used in our experiments – LLB – contained an isolated porosity of 0.01, on average (Table 2). Similar to the mechanism posited by Loaiza *et al.* (2012), we suggest that distributed microcracking the initial stages of ductile deformation serves to interconnect this isolated porosity, creating efficient pathways for fluid flow. Mechanical data of all the experiments (Fig. 1a) exhibit intermittent stress drops – even in the




instances where permeability was observed to increase relative to the initial value – which suggests that compaction localisation in these andesites does little in the way of forming a barrier to fluid flow. Loaiza *et al.* (2012), Adelinet *et al.* (2013) and Heap *et al.* (2015a) each examine microstructure of compaction bands formed in porous volcanic rocks. In all cases, they are irregular in shape and thickness, but do not necessarily constitute a contiguous surface of collapse pores (i.e. a

layer of reduced porosity). Their characteristic tortuosity – and the fact that the reduction in porosity relative to the host sample is remarkably less than observed in sandstones (Baud *et al.*, 2012) – may explain the lack of an obvious observed influence on sample permeability in this study. Thus, counter-intuitively, small amounts of stress-induced compaction may actually increase permeability in volcanic materials. At higher strains however, this effect is overtaken by the global reduction in sample porosity, which serves to decrease the mean flow path aperture and forces fluids to travel through more

tortuous routes.

### 4.2 A limit to compaction and permeability reduction

Porosity exerts a first order control on the brittle-ductile transition of porous rocks (e.g. Wong and Baud, 2012): high

porosity fosters ductile behaviour in response to an applied differential stress whereas dilatant brittle behaviour is anticipated in low-porosity materials. If a high-porosity volcanic rock undergoes progressive compaction, it will eventually achieve a porosity low enough to respond in a dilatant fashion. The critical stress state at this transition is known as $C^{*\prime}$ has been previously described in sedimentary materials (e.g. Schock *et al.*, 1973; Baud *et al.*, 2000; Vajdova *et al.*, 2004; Baud *et al.*, 2006; Regnet *et al.*, 2015), and recently in andesite from Volcán de Colima (Heap *et al.*, 2015a). As anticipated, beyond a

threshold stress-strain accumulation, sample LLB-13 exhibits a transition from compactant deformation to dilatant behaviour (highlighted by the arrow in Fig. 1b). In volcanic rock – at the sample-scale – this is characterised by significant shortening and barrelling of the sample (a consequence of the high strains required to achieve and exceed $C^{*\prime}$) and the generation of an extensive dilatant shear zone (Heap *et al.*, 2015a), characteristically similar to highly strained fault zones observed in volcanic rock (Farquharson *et al.*, 2016b). Figure 4 illustrates these separate mechanisms, with evidence of cataclastic pore

collapse being shown in Fig. 4b, and the dilatant shear zone shown in detail in Fig 4c. As well as inhibiting net compaction, the transition to dilatant behaviour after a critical stress threshold may well constitute a limit to permeability reduction. Indeed, Regnet *et al.* (2015) observed that the permeability of an oolitic limestone increased when deformed triaxially beyond $C^{*\prime}$, to a level greater than its original value. This suggests that the volumetric increase associated with continued deformation after this critical stress state is linked to the generation of an efficient flow path for transmitted fluids. The

concept of a limit to porosity and permeability reduction is supported by compiled data shown in Fig. 5, and suggests that for a volcanic rock of given initial porosity, there is a limited range of strain-induced subsolidus $k–\phi$ states in which it can exist: one cannot compact indefinitely without promoting dilatant mechanisms.

Figure 5 compiles data from this study with that of Farquharson *et al.* (2016b) and Heap *et al.* (2015a). Farquharson *et al.*

(2016b) performed triaxial experiments on low- and intermediate-porosity volcanic rocks (basalt from Mount Etna, Italy, and andesites from Volcán de Colima and Kumamoto, Japan), exploring the evolution of physical properties as a function of stress-induced dilation. Heap *et al.* (2015a) performed compaction experiments comparable to those described in this study, with pre- and post-deformation permeability being assessed for two samples of San Antonio (C8) andesite from Volcán de Colima. Notably, data for porosity (Fig. 5a) and permeability (Fig. 5b) of all samples tends to converge towards intermediate

values with ongoing strain accumulation. For this set of samples, an initial porosity range of 0.05 - 0.23 reduces to values between 0.07 and 0.19 after an axial strain of 0.05, and thereafter to a range of 0.08 to 0.17 after an axial strain of 0.10. A similar trend can be observed in the permeability data, albeit with appreciably more scatter. Assuming that stress-induced fracture and compaction are common in active volcanic environments, this phenomenon of convergence towards



intermediate values (i.e. $0.10 \leq \phi \leq 0.20$; $10^{-14} \leq k \leq 10^{-13}$ m$^2$) might partially explain why the modal porosity of large datasets of edifice-forming material tends to fall between 0.10 and 0.20 (e.g. Mueller *et al.*, 2011; Bernard *et al.*, 2015; Farquharson *et al.*, 2015; Lavallée *et al.*, 2017).

### 4.3 Implications for volcanology

Our experimental results highlight that permeability evolution during stress-induced compaction may be complex, but an overarching trend of decreasing permeability is anticipated, especially if volcanic rock can compact to relatively high amounts of strain (i.e. > 0.06). This section examines these results in a broader context: if we can expect permeability of edifice-forming rocks to vary due to porosity, effective pressure and stress, then what influence does this have on volcanic activity?

Due to different histories of degassing, ascent, and eruption processes for different volcanic ejecta and effusive products (*e.g.* Mueller *et al.*, 2011), heterogeneous edifice porosity may arise over time. Volcán Rincon de la Vieja, for example, exhibits contrasting flank compositions as tephra is predominantly deposited on the western side due to the prevailing trade winds, whereas dense lava flows have been historically concentrated to the north and south (Kempter *et al.*, 1996). Similarly, Volcán Casita (Nicaragua) is composed primarily of pyroclastic units on the southwest (again a consequence of the prevailing wind direction), whilst the majority of relatively denser lava flows extend to the east because of an asymmetric crater morphology (van Wyk de Vries *et al.*, 2000). Indeed, geophysical surveys of active and historically active volcanoes indicate that significant variations in density (and hence, porosity) may be a common feature of stratovolcanic edifices worldwide. For example, Tiede *et al.* (2005) use gravimetric inversion to explore edifice density at Gunung Merapi, identifying a relatively low density unit on the western flank. These authors calculate an average porosity of 0.21 for this unit: a high value compared to the average edifice porosity of around 0.15 determined by Setiawan (2002) and the range of 0.05 - 0.10 estimated by Commer *et al.* (2005) for the region directly below the Merapi summit. These values are generally consistent with measured laboratory values of porosity for Merapi samples (Le Pennec *et al.*, 2001; Kushnir *et al.*, 2016). Similar contrasts in density have been inferred from gravimetric studies of several other volcanic regions such as Mauna Loa, Hawai'i (Zucca *et al.*, 1982), Campi Flegrei, Italy (Cubellis *et al.*, 1995), Cordón Caulle, Chile (Sepúlveda *et al.*, 2005), and in the Central Volcanic Complex of Tenerife, Spain (Gottsmann *et al.*, 2008).

In agreement with previous experimental studies (Loaiza *et al.*, 2012; Adelinet *et al.*, 2013; Zhu *et al.*, 2016), the data presented here show that relatively porous edifice rock is prone to compact, even under low effective pressure (*i.e.* at shallow depths), whereas Heap *et al.* (2015a) and Farquharson *et al.* (2016b), for example, show that volcanic rocks with a low initial porosity will preferentially dilate when subject to stress. Thus one or other of these two processes will be dominant in different regions of a volcanic edifice. If we take Gunung Merapi as an example, the relatively porous western flank should compact – if subject to an imposed differential stress – in the shallow edifice; on the other hand, the denser regions will fracture near the surface but facilitate compaction at depth by increasing the overlying lithostatic pressure for a given depth (*i.e.* the effective confining pressure). Accordingly, permeability reduction should occur over time and with increased strain in the western portion of the edifice, whereas permeability of the other flanks should increase. This is in agreement with the broad trend of outgassing observed at Gunung Merapi: quiescent outgassing occurs through and around the central vent, as well as through fumarole fields located to the east, southeast, and south of the main crater (Le Cloarec and Gauthier, 2003), while fumarole fields are absent to the west of the crater.

This field evidence underscores the importance of the spatial distribution of edifice rock with differing physical and mechanical properties in terms of the evolution of permeability and outgassing routes in volcanic systems. Highly porous



edifice rock may constitute an effective means of passive outgassing during periods of eruptive quiescence; when subject to stress, however – for example as a result of shallow fluid migration (*e.g.* Denlinger and Hoblitt 1999; Clarke *et al.* 2007) or deep-seated magma chamber deformation (*e.g.* Melnik and Sparks 2005; Wadge *et al.*, 2006) – its permeability will tend to decrease. In turn, this may hinder outgassing and promote explosive activity. However, as our data show (Fig. 3b), low amounts of inelastic strain may actually increase permeability, possibly by connecting isolated porosity. In this case, stress-induced compaction may yield a temporary increase in permeability and flank outgassing, belying a longer-term trend of permeability reduction. This implies that the spatial and temporal distribution of fumaroles and the vigour with which they outgas may contain important information regarding subsurface strain accumulation: in a quiescent system, changes in the volume of passive outgassing of magmatic gas species may reflect stress-induced compaction, portending a build-up of pressure and potential explosive activity.

## 5 Conclusions

Volcanic rock of relatively high porosity (i.e. >0.20) can compact as a function of inelastic strain accumulation. We performed a series of triaxial deformation experiments on a suite of porous andesite in order to explore the influence of stress-induced compaction on porosity and permeability evolution. The efficiency of compaction was found to be a function of the effective pressure under which deformation occurred: at higher effective pressure, a greater volume of porosity was lost for any given amount of inelastic strain, reflecting the reduction in the inelastic compaction factor as a function of increasing effective pressure. We suggest that the associated underlying physical mechanism is progressively more efficient pore occlusion through cataclastic pore collapse at higher confining pressures, due to enhanced communution. At low strains (<0.05), compaction tends to result in a moderate increase in permeability, which we suggest is a result of increased pore connectivity due to distributed microcracking. This effect is outweighed by progressive compaction at higher strains, resulting in a general trend of decreasing permeability with ongoing inelastic compaction. There exists a physical limit to compaction, which we suggest is echoed in a limit to the potential for permeability reduction in a deforming sample. Compiled data show that at high strain, porosity and permeability tend to converge towards intermediate values (i.e. $0.10 \leq \phi \leq 0.20$; $10^{-14} \leq k \leq 10^{-13}$ m$^2$). Field evidence from the literature emphasises the importance of understanding the physical and mechanical properties of rock in active volcanic environments, in particular the evolution in a rock's capacity to effectively transmit magmatic volatiles.

**Author contributions**

J.F., P.B., and M.H. designed the experiments, which were performed by J.F. J.F. prepared the manuscript with contributions from all authors.

**Competing interests**

The authors declare that they have no conflict of interest.

**Data availability**

Data are presented in Tables 1 and 2 in the text, or are available on request to the corresponding author.

**Acknowledgements**

Thierry Reuschlé, Alex Schubnel, Luke Griffiths, and Alexandra Kushnir are thanked for inspiring discussions. Fieldwork was funded in part by the framework of the LABEX ANR-11-LABX-0050_G-EAU-THERMIE-PROFONDE and therefore benefits from a funding from the state managed by the French National Research Agency as part of the Investments for the future program. JIF acknowledges an Initiative d'Excellence (IDEX) "Contrats doctoraux" grant from the French State. MJH acknowledges Initiative d'Excellence (IDEX) Attractivité grant "VOLPERM" and a CNRS INSU grant. We are grateful to Nick Varley and Oliver Lamb for field assistance at Volcán de Colima.



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



**Tables**

**Table 1: Major element (oxide) composition, determined via X-ray fluorescence analysis. All values are given in weight percent (wt. %). $SiO_2$ = Silicon dioxide; $Al_2O_3$ = Aluminium oxide; $Fe_2O_3$ = Iron oxide; MnO = Manganese(II) oxide; MgO = Magnesium oxide; CaO =Calcium oxide; $Na_2O$ = Sodium oxide; $K_2O$ = Potassium oxide; $TiO_2$ = Titanium oxide; $P_2O_5$ = Phosphorus**
5  **pentoxide; $Cr_2O_3$ = Chromium(III) oxide; $V_2O_5$ = Vanadium(V) oxide; NiO = Nickel(II) oxide; LOI = loss on ignition.**

| $SiO_2$ | $Al_2O_3$ | $Fe_2O_3$ | MnO | MgO | CaO | $Na_2O$ |
|---|---|---|---|---|---|---|
| 61.260 | 17.330 | 5.745 | 0.100 | 3.725 | 5.505 | 4.455 |
| ± 0.270 | ± 0.070 | ± 0.085 | ± 0.001 | ± 0.295 | ± 0.025 | ± 0.015 |
| $K_2O$ | $TiO_2$ | $P_2O_5$ | $Cr_2O_3$ | $V_2O_5$ | NiO | LOI |
| 1.505 | 0.565 | 0.200 | 0.020 | 0.015 | 0.004 | 0.185 |
| ± 0.035 | ± 0.015 | ± 0.020 | ± 0.005 | ± 0.005 | ± 0.001 | ± 0.215 |

**Table 2: Physical property data and deformation conditions for each sample. Initial connected porosity, unconnected porosity, and permeability are given as $\phi$, $\phi_u$, and $k_0$, respectively. $\delta\phi_i$ = inelastic porosity change; $\phi + \delta\phi_i$ = post-deformation porosity; $k_e$ = post-deformation permeability. Samples were deformed under an effective pressure $p_{eff}$ until a total target strain $\varepsilon_t$, then**
10  **unloaded to determine the inelastic strain $\varepsilon_i$. Similarly, $\delta\phi_i$ indicates total porosity change after unloading. $p_{eff}$ is determined by subtracting a constant $p_p$ value of 10 MPa from the confining pressure $p_c$.**

| Sample | $\phi$ | $\phi_u$ | $p_{eff}$ [MPa] | $\varepsilon_t$ | $\varepsilon_i$ | $\delta\phi_i$ | $\phi + \delta\phi_i$ | $k_0$ [m$^2$] | $k_e$ [m$^2$] |
|---|---|---|---|---|---|---|---|---|---|
| LLB-6 | 0.21 | 0.01 | 10 | 0.010 | 0.006 | -0.005 | 0.21 | $3.80 \times 10^{-13}$ | $4.38 \times 10^{-13}$ |
| LLB-7 | 0.23 | 0.01 | 10 | 0.030 | 0.026 | -0.017 | 0.21 | $4.83 \times 10^{-13}$ | $3.71 \times 10^{-12}$ |
| LLB-1 | 0.21 | <0.01 | 10 | 0.060 | 0.055 | -0.020 | 0.19 | $5.01 \times 10^{-13}$ | $3.37 \times 10^{-13}$ |
| LLB-10 | 0.22 | 0.01 | 30 | 0.030 | 0.025 | -0.021 | 0.20 | $5.24 \times 10^{-13}$ | $3.56 \times 10^{-12}$ |
| LLB-9 | 0.22 | 0.01 | 30 | 0.060 | 0.054 | -0.040 | 0.18 | $4.97 \times 10^{-13}$ | $1.62 \times 10^{-13}$ |
| LLB-13 | 0.22 | 0.01 | 30 | 0.240 | 0.233 | -0.055 | 0.16 | $4.84 \times 10^{-12}$ | $5.51 \times 10^{-14}$ |
| LLB-3 | 0.19 | 0.01 | 50 | 0.030 | 0.024 | -0.019 | 0.18 | $5.31 \times 10^{-13}$ | $5.20 \times 10^{-13}$ |
| LLB-11 | 0.22 | 0.01 | 50 | 0.060 | 0.054 | -0.045 | 0.18 | $5.03 \times 10^{-13}$ | $1.29 \times 10^{-12}$ |
| LLB-5 | 0.21 | 0.01 | 50 | 0.180 | 0.174 | -0.074 | 0.13 | $4.32 \times 10^{-13}$ | $3.44 \times 10^{-14}$ |
| LLB-14 | 0.23 | 0.01 | 70 | 0.060 | 0.054 | -0.048 | 0.18 | $4.70 \times 10^{-13}$ | $1.27 \times 10^{-12}$ |



**Figure captions**

**Figure 1: Schematic of triaxial deformation apparatus, including confining pressure ($p_c$), pore pressure ($p_p$), and axial pressure ($p_{ax}$) circuits. Detail of sample assemblage is shown inset. (a) axial piston; (b) "blank" endcap; (c) sample; (d) copper foil jacket;**
**(e) nitrile jacket; (f) pore fluid distributor endcap. Directions of major $\sigma_1$ and minor $\sigma_3$ principal stresses are as shown, such that $\sigma_1 > \sigma_2 = \sigma_3$. Not to scale. Numbered valves allow various parts of each circuit to be used at any given time.**

**Figure 2: (a) Differential stress versus axial strain for all experiments. _Inset_: close-up of an experiment (sample LLB-7) highlighting different stages of deformation (refer to text for explanation). (b) Porosity change as a function of axial strain for all**
**tests. Effective pressure at which deformation was performed is indicated by the line colour.**

**Figure 3: Change in physical properties after accumulating inelastic axial strain. (a) Pre- and post-deformation porosity (the latter calculated by $\phi + \delta\phi_i$) versus inelastic strain. Effective pressure is indicated by the symbol colour. (b) Pre- and post-deformation gas permeability as a function of the inelastic strain accumulated by each sample. Effective pressure is indicated by colour as in**
**panel (a).**

**Figure 4: Backscattered scanning electron microscope images of LLB andesite, showing as-collected (a) and post-deformation (b - c) microstructure. Void space appears as black. Dense (metal-rich) phenocrysts appear as white or light grey within a darker grey groundmass. Both (b) and (c) are images of LLB-13, which was taken to beyond both $C^*$ and $C^{*\prime}$. Cataclastic pore collapse**
**associated with shear-enhanced compaction is shown in (b), while (c) shows part of the dilatant shear zone marking the transition from shear-enhanced compaction to dilation.**

**Figure 5: Compiled porosity and permeability data for triaxially deformed volcanic rocks. † Heap _et al._ (2015a); ‡ Farquharson _et al._ (2016b). (a) Porosity during deformation, calculated by summing the porosity change $\delta\phi$ and the initial sample porosity $\phi$.**
**Samples distinguished by line, effective pressure by colour. (b) Permeability measured post-deformation as a function of inelastic strain. Samples are distinguished by symbol, effective pressure by colour.**





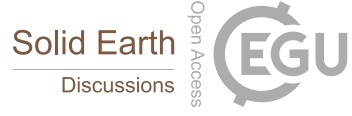

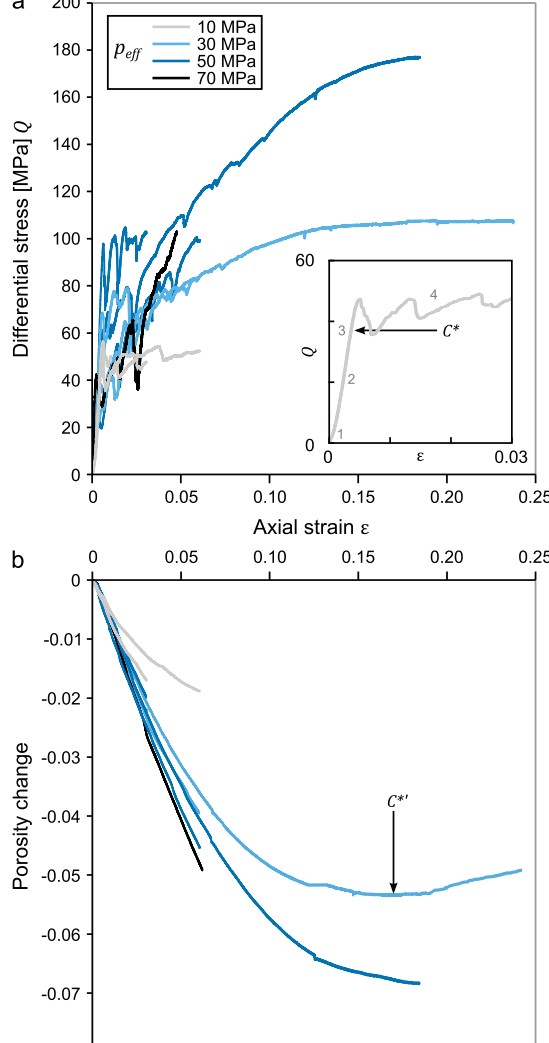



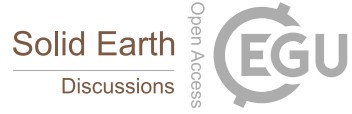

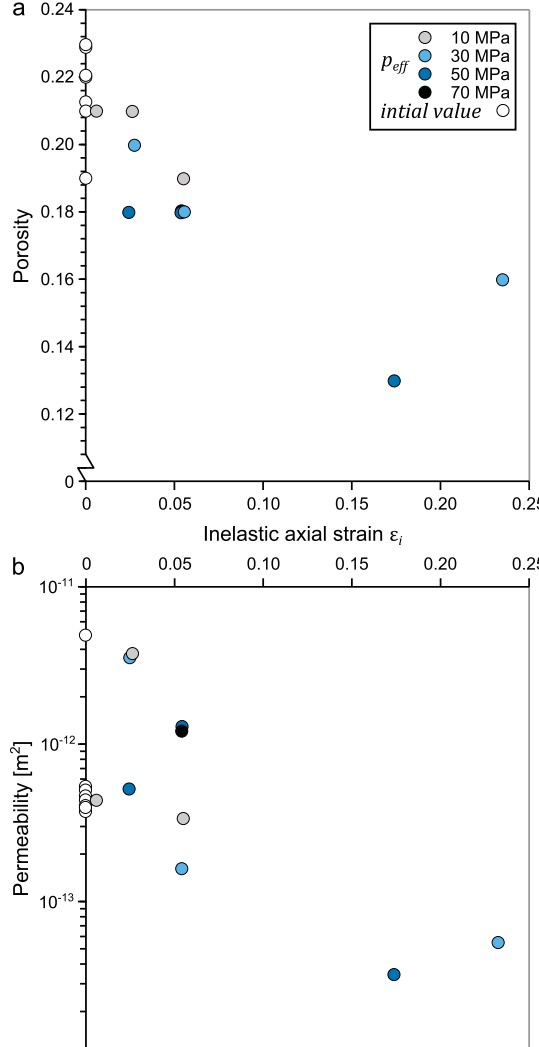

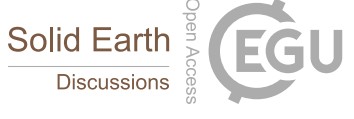



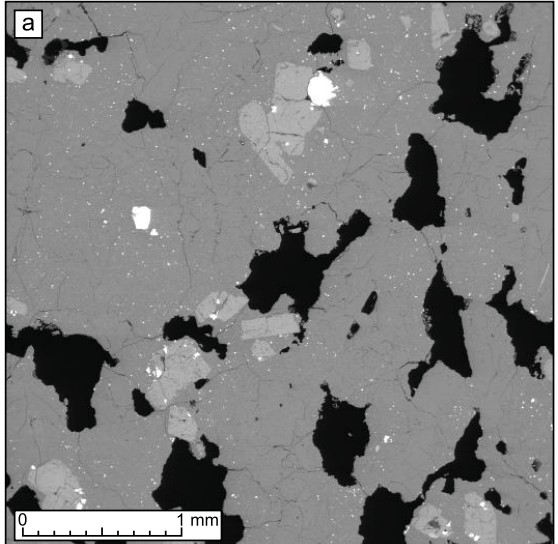

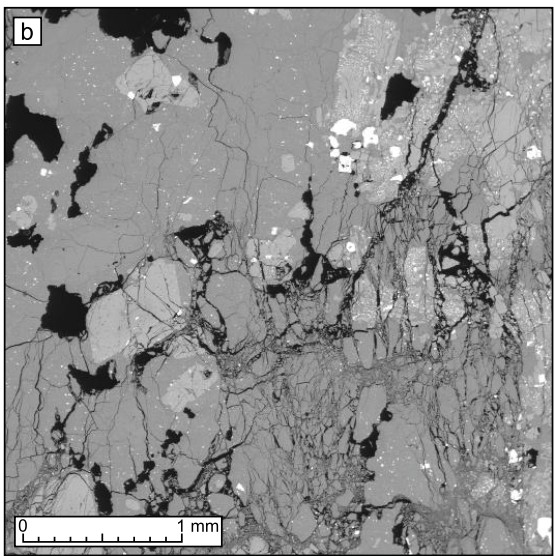

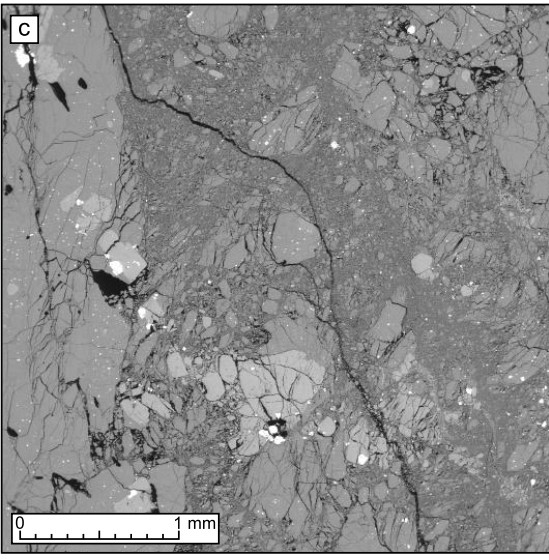



