# Peer review of "Inelastic compaction and permeability evolution in volcanic rock"

_Solid Earth, 2016_

## Referee Comment (RC1) · F. Wadsworth (Referee) · 9 Feb 2017

This manuscript deals with a central question in Earth science related to fluid migration in our volcanically and tectonically active lithosphere. Namely, the evolution of fluid permeability during deformation. The hypothesis that is here tested experimentally is that the evolution of the fluid permeability in volcanic rocks is highly dependent on the inelastic strain accumulated when the deformation occurs in the compaction regime. There is little or no analysis of the data presented beyond computing the inelastic component of the bulk (or axial?) strain and presented measured values of properties that evolve with strain. The is the only shortcoming of this work. Otherwise, the manuscript is well illustrated, well written and meets the standards of novelty and quality for publication. Below I submit some constructive comments which I hope will guide the authors in their revision of this article.

(1) In the abstract line 19, the term "high strains" is used. I think that throughout the manuscript the authors should be explicit and clear about what strain they are talking. Axial strain, pore strain, and inelastic strain are all introduced at some point and so referring to strain more generally is not clear. (2) Page 2 line 10; do the authors mean to say "siphoned"? Or do they simply mean that the volatiles are able to escape? Throughout the manuscript there is some flamboyant language that perhaps precludes a clear understanding of what is meant in places. Another example is perhaps the use of "amoeboid" as an adjective to describe a pore shape on line 26 of the same page. (3) Page 2 line 14; not all of the papers cited here are explicitly in support of the statement made. (4) Page 2 line 31; the authors state that porosity is the reciprocal of bulk density, which is not true. It's clear what the authors mean but, as stated above, they should be very clear in the language they use and check throughout the manuscript that they all agree on each sentence. (5) Page 2 line 40; by this point in the manuscript I thought it would be illustrative to have a diagram that schematically explained the concepts of this paragraph. If the authors felt inclined to provide one I think it would strongly improve the understanding of the regimes to which they refer. (6) Page 3 line 2; the reference to Farquharson et al., 2016b should occur at the end of the sentence – in a few places the authors put a reference in the middle of a sentence without it always being necessary. (7) Page 3 line 7; do they authors mean "monotonically" when they say "monotonously"? (8) Page 4 line 1; on all measurements of a value, the authors should give an experimental uncertainty. The density should have an uncertainty quoted with it, please. (9) Page 4 line 3; if the authors use a correction for raw permeability data, could they please show it explicitly. As far as I know they are far from any length limit associated with this journal and so they could take the space to be as thorough as possible. An equation for the correction and a plot to which the reader could refer to see how the correction works would be valuable. (10) Page 4, line 9; what does a poroelastic constant of unity mean in physical terms? It seems to

imply that the pressures are isotropically distributed in the solid matrix. Does this imply something about the compressibility? Could the authors unpack this useful equation a little in the text that follows? (11) Page 4, line 11; the authors state that they apply a pore pressure. Presumably they do not apply this to any isolated pores. This would need stating clearly here. (12) Page 4, line 18; I am familiar with the justification of a strain rate of $10^{-5}$ $s^{-1}$ in rock mechanics studies on the grounds of "standard practice". However, do the authors perhaps have some justification for using this rate on more physical grounds? For example, Heap & Wadsworth (2016) among many other studies showed that a dominant control of strain rate in permeable rock is to shift regime from effectively permeable on the timescale of deformation, to effectively impermeable. It's ok to say that this rate was used for experimental expediency but it would more satisfying to see if it could be justified. (13) Page 4, line 20; do the authors mean $d\phi$ and not $\delta\phi$? I suspect they do. In which case please change throughout. (14) Page 4, line 38; if $\alpha$ is different for different properties when everything else is the same, then it is not the same coefficient – i.e. it is not $\alpha$. In which case don't use the same symbol. This took me a few reads to understand and would be clearer if the authors simply referred to, for example, $\beta$, and named it another poroelastic constant that is different from $\alpha$. This does beg the question of why it's different and what physics $\beta$ is tracking that $\alpha$ is not. (15) Page 5, line 6; why would you need the mean radius? (16) Page 5, line 10; this equation begs an explanation. Why is this the answer to an effective cross-sectional area in a deformed sample? This is not explained at all. (17) Page 5, line 32; it's clear here that the authors mean an axial strain. Please can they go carefully through the manuscript and be specific about strain definitions. Otherwise they could say up front that throughout this work "strain" is used to mean "axial strain". This is important as they talk about other forms of strain. (18) The discussion is robust and I have no commentary. (19) The figures are well presented. However Figure 5, which is the take-home message of the work, is not clearly presented. I would recommend some analysis be explored, such as a "change in permeability" as a function of inelastic strain or some metric that scales with the effective

pressure of deformation? The reader might be looking for some take home conclusion which they don't find here. I do not see clearly, for example, that permeability increases and then decreases, as the authors state. (20) A central conclusion of the work seems to be this: that in the brittle regime permeability increases because strain is accommodated by discrete fractures. And that this is different from the compaction regime in which pore crushing produces diffuse areas (compaction bands) of granular material. The novel idea here is that the effective pressure controls the particle sizes in these zones which in turn controls the permeability limit during progressive compaction. However, this idea is left hanging without much additional support or exploration. There are many permeability models for granular media, including polydisperse examples (e.g. Wadsworth et al., 2016), which could be used to confirm this idea. It's exciting and therefore a little bit of a shame to leave it unexplored with simply a statement that it would make a good future study. If the authors felt inclined to unpack this idea a little, it would be worthwhile.

---

## Editor Comment (EC1) · A. Longo (Editor) · 28 Feb 2017

The paper deals with a basic problem of volcanology, in particular the evolution of porosity and permeability of volcanic rock under inelastic compaction. Implications on magmatic degassing and occurence of an explosive eruption are discussed. The study draws conclusions from laboratory experiments.
The paper is worth publishing after minor revision.

**Minor Observations**

**P2L32** The authors could provide other references on volcanic cases of detection of inelastic compaction to enhance the importance of the their study.

**P4L1**  Which is the standard deviation or the experimental error for the estimated value of the mean of $\rho_p$? The same for all experimental data that are reported further in the paper.

**P4L11-30**  I expect that the temperature plays a major role in determining the inelastic behaviour of rock. Which is the temperature of the samples during the experiments? Is it compatible with the depths corresponding to the different confining pressures applied?

**P5L4**  The verbs "are be" should be corrected, maybe only "are".

**P5L10**  Equation (2) should be explained in more details, or a reference for it must be given.

---

## Author Comment (AC1) · 28 Mar 2017

**Response to editor comment:**

**"Inelastic compaction and permeability evolution in volcanic rock"**

Dear editor,

Please find attached the revised version of our manuscript Inelastic compaction and permeability evolution in volcanic rock. We are pleased with the constructive suggestions from the reviewers which, although minor, have helped to further improve the paper. Below we outline the changes made to the manuscript (in **green**). Our responses to the editor comments are given in **bold** below.

We believe that we have addressed all of the comments in a satisfactory manner, and hope that you will consider our improved article for publication in *Solid Earth*. The revised manuscript and supplementary files are attached after the comment responses.

Yours sincerely,

Jamie Farquharson, on behalf of P. Baud, and M. J. Heap.

**"Inelastic compaction and permeability evolution in volcanic rock"**

*Editor comment* on "Inelastic compaction and permeability evolution in volcanic rock" *by* **Jamie I. Farquharson et al.**

A. Longo

antonella.longo@ingv.it

The paper deals with the basic problem of volcanology, in particular the evolution of porosity and permeability of volcanic rock under inelastic compaction. Implications on magmatic degassing and occurrence of an explosive eruption are discussed. The study draws conclusions from laboratory experiments.

The paper is worth publishing after minor revisions.

Minor observations

P2L32  The authors could provide other references on volcanic cases of detection of inelastic compaction to enhance the importance of the their study.

**We agree that further references would be useful here. We now include the following paragraph:**

**"To date, physical property data for volcanic materials at depth has been obtained predominantly by researchers assessing the suitability of volcanic deposits for hydrocarbon or geothermal energy exploitation. For example Chen et al. (2016, 2017) investigated volcanic sequences in the Junggar Basin in western China, in order to determine their suitability as gas and petroleum reservoirs. At both sites, the authors note a general decrease in porosity with depth; for example, Chen et al. (2016) report a decrease of porosity from ~0.30 to <0.10 between the surface and 1000 m depth. Often, the interpretation of logging data from volcanic materials (e.g. Millett et al., 2016) is non-trivial due to the variable quality of density logs, and variations in the relationship between density and porosity with depth, alteration, or the intersection of distinct facies (Li et al., 2009), as well as gaps in the stratigraphic record due to incomplete core recovery. An example of a study with an excellent degree of core recovery—99.7 %—is reported by Jónsson and Stefánsson (1982): these authors calibrate porosity and density data obtained by the Iceland Research Drilling Project from a borehole cored continuously to a depth of 1919 m near Reyðarfjörður in Iceland. To combat small-scale variability arising from the intersection of discrete geological units, Jónsson and Stefánsson (1982) calculate a running average of porosity and density against depth. Notably, the average porosity decreases from 0.13 at 400 m depth to 0.06 at 1200 m depth, corresponding to an increase in bulk rock density of approximately 200 kg m$^{-3}$."**

P4L1  Which is the standard deviation or the experimental error for the estimated value of the mean of $\rho_p$? The same for all experimental data that are reported further in the paper.

**Response to editor comment:**

**"Inelastic compaction and permeability evolution in volcanic rock"**

**Powder density $\rho_p$ (a.k.a. skeletal density or particle density) is determined by measuring the volume of a quantity of powdered rock of known mass. Using helium pycnometry, we measure the volume ten times or until the results of five successive runs are within a given tolerance of each other (whichever occurs first). In the case of the LLB powder presented herein, the standard deviation of our measurements was 0.0003 g cm$^{-3}$. Ultimately this gives a value of 2653 ± 0.17 kg m$^{-3}$. We have now updated this accordingly in the text. Similarly, errors in porosity measurements are a function of the iterated volume measurements as well as potential error in the measurement of sample length and diameter. We now address this explicitly in the text:**

**"In detail, helium pycnometry is used to determine the solid volume $Vs$ of each sample, which subsequently allows the calculation of the connected gas porosity $\phi$ as described. Automated measurements of $Vs$ were performed iteratively until five consecutive measurements yielded results within a range of 0.01 % of the sample volume, so precision of the pycnometer measurements is high (< 0.005 cm$^3$). A greater degree of error arises when manually measuring the sample dimensions, required in order to determine $\rho v$. Repeat measurements allowed an estimation of error in the length and diameter, which typically amount to < 0.05 cm$^3$ in terms of volume. Adopting the notation that $\epsilon_\phi$ is the error on the porosity calculation and that $\epsilon_x$ and $\epsilon_y$ are the independently calculated errors for measurements of $Vs$ and $V$, then the propagated error can be approximated by:**

$$\epsilon_\phi = \left[ \left( \frac{\epsilon_z}{V - V_s} \right)^2 + \left( \frac{\epsilon_y}{V} \right)^2 \right]^{0.5} ; \quad \epsilon_z = \left( \epsilon_x^2 + \epsilon_x^2 \right)^{0.5}$$

**Values for $\epsilon_\phi$ are generally < 0.005 for the samples described herein. As such, probable error on connected gas porosity measurements is low, and always contained within the symbol size when plotted graphically."**

**We now also include an appendix (after comments by Reviewer 1), which details the methods used to measure permeability, the corrections applied, and the sources of error in the permeability data. The text now reads: "The appendix contains further details on the determination of permeability, the application of corrections for inertial effects, and the sources and sizes of potential error in the measurements."**

P4L11-31 I expect that temperature plays a major role in determining the inelastic behaviour of rock .Which is the temperature of the samples during the experiments? Is it compatible with the depths corresponding to the different confining pressures applied?

**Samples were deformed at room temperature. Certainly in a natural volcanic setting there will be influences of temperature on rock strength that are not accounted for here (for instance, the closure of cracks driven by thermal expansion). Nevertheless, recent work (Heap et al., 2017) shows that the influence of temperature during compaction is negligible as long as the rock is below its glass transition temperature (i.e. subsolidus). In the same paper, the authors show that host rock material must be in extreme proximity to the conduit in order to be transiently molten (Heap et al., 2017 show that the $T$g**

isotherm only migrates on the centimetric scale for heating and cooling of the magma column over timescales of days to months).

We now acknowledge this point in the text:

"Samples were all deformed at room temperature. We note that in natural volcanic environments there may be some influence of temperature on rock strength (due, for instance, to the closure of cracks driven by thermal expansion). Nevertheless, a recent study by Heap et al. (2017) shows that the influence of temperature on the physical and mechanical properties of andesite may not be significant at temperatures below the glass transition $T$g. Importantly, this study showed that the failure mode and underlying microstructural mechanism driving ductile behaviour (cataclastic pore collapse) did not change below $T$g, which is itself largely restricted to the magma conduit and rock in the immediate vicinity. This is in agreement with previous studies by Vinciguerra et al. (2005) and Heap et al. (2014), both of which noted only negligible changes in microcrack density and porosity after thermally stressing volcanic materials. These authors attribute this phenomenon to the high initial crack density resulting from the complex thermal histories of volcanic rocks. "

P5L4 The verbs "are be" should be corrected, maybe only "are".

Agreed. This typo has now been corrected.

P5L10  Equation (2) should be explained in more details, or a reference for it must be given.

Essentially, Equation (2) is an extension of the relation between area A, volume V and length L (i.e. $A = V/L$), whereby we account for the change in pore volume after deformation. For clarity we now present the equation as follows (the accompanying text has also been updated accordingly):

$$A_{post} = \underbrace{\left[\frac{V_s}{1 - (\phi - \delta\phi_i)}\right]}_{V_{post}} \times \left[\frac{1}{l_{post}}\right]$$

where $V_{post}$ is the post-deformation volume and $l_{post}$ is the mean sample length after deformation.

This formulation of $V_{post}$ may be equivalently shown as

$$V_{post} = V_s + (\phi \times V_b) + (\delta\phi_i \times V_b)$$

 where $V_b$ is the "bulk" volume of the sample after deformation (i.e. both the solid and void parts) and $\phi$ and $\delta\phi_i$ are void fractions. We choose to present it in the fashion above as we do not have to include another additional variable ($V_b$).

**Response to editor comment:**

**"Inelastic compaction and permeability evolution in volcanic rock"**

**Revised manuscript and supplementary files**

In the following revised manuscript, new text is blue and underlined, and deleted text is red and struck-through.

[revised manuscript text omitted]
.  To date, physical property data at depth in volcanic materials has been obtained predominantly by researchers assessing the suitability of volcanic deposits for hydrocarbon or geothermal energy exploitation. For example Chen et al. (2016, 2017) investigated volcanic sequences in the Junggar Basin in western China, in order to determine their suitability as gas and petroleum reservoirs. At both sites, the authors note a general decrease in porosity with depth; for example, Chen et al. (2016) report a decrease of porosity from ∼0.30 to <0.10 between the surface and 1000 m depth. Often, the interpretation of logging data from volcanic materials (e.g. Millett et al., 2016) is  nontrivial due to the variable quality of density logs and variations in the relationship between density and  facies (Li et al., 2009), as well as gaps in the stratigraphic record due to incomplete core recovery. An example of a study with an excellent degree of core recovery—99.7 %—is reported by Jónsson and Stefánsson (1982): these authors calibrate porosity and density data obtained by the Iceland Research Drilling Project from a borehole cored continuously to a depth of 1919 m near Reyðarfjörður in Iceland. To combat small-scale variability arising from the intersection of discrete geological units, Jónsson and Stefánsson (1982) calculate a running average of porosity and density against depth. Notably, the average porosity decreases from 0.13 at 400 m depth to 0.06 at 1200 m depth, corresponding to an increase in bulk rock density of approximately 200 kg m$^{-3}$.

[revised manuscript text omitted]

$$\phi = \left( \frac{\rho_b - \rho_v}{\rho_v} \right) \tag{1a}$$

$$\phi_t = 1 - \left( \frac{\rho_b}{\rho_p} \right) \tag{1b}$$

$$\phi_u = \phi_t - \phi \tag{1c}$$

where a  value of 2653 $\pm$ 0.17 kg m$^{-3}$ is used for  $\rho_p$. In detail, helium pycnometry is used to determine the solid volume $V_s$ of each sample, which subsequently allows the calculation of the connected gas porosity $\phi$ as described. Automated measurements of $V_s$ were performed iteratively until five consecutive measurements yielded results within a range of 0.01% of the sample volume, so precision of the pycnometer measurements is high ($< 0.005$ cm$^3$). A greater degree of error arises when manually measuring the sample dimensions, required in order to determine $\rho_v$. Repeat measurements allowed an estimation of error in the length and diameter, which typically amount to $< 0.05$ cm$^3$ in terms of volume. Adopting the notation that $\epsilon_\phi$ is the error on the porosity calculation and that $\epsilon_x$ and $\epsilon_y$ are the independently calculated errors for measurements of $V_s$ and $V$, then the propagated error can be approximated by:

$$\epsilon_\phi = \phi \times \left[ \left( \frac{\epsilon_z}{V - V_s} \right)^2 + \left( \frac{\epsilon_y}{V} \right)^2 \right]^{\frac{1}{2}} ; \ \epsilon_z = \left( \epsilon_x^2 + \epsilon_y^2 \right)^{\frac{1}{2}} \tag{2}$$

Values for $\epsilon_\phi$ are generally $< 0.005$ for the samples described herein. As such, probable error on connected gas porosity measurements is low, and always contained within the symbol size when plotted graphically.

Gas permeability was measured under steady-state conditions with a confining pressure of 1 MPa using the setup

**Table 1.** Major element (oxide) composition, determined via X-ray fluorescence analysis. All values are given in weight percent (wt. %).

| $SiO_2$ | $Al_2O_3$ | $Fe_2O_3$ | $MnO$ | $MgO$ | $CaO$ | $Na_2O$ |
|---|---|---|---|---|---|---|
| 61.260 | 17.330 | 5.745 | 0.100 | 3.725 | 5.505 | 4.455 |
| $\pm$ 0.270 | $\pm$ 0.070 | $\pm$ 0.085 | $\pm$ 0.001 | $\pm$ 0.295 | $\pm$ 0.025 | $\pm$ 0.015 |
| $K_2O$ | $TiO_2$ | $P_2O5$ | $Cr_2O_3$ | $V_2O_5$ | $NiO$ | LOI |
| 1.505 | 0.565 | 0.200 | 0.020 | 0.015 | 0.004 | 0.185 |
| $\pm$ 0.035 | $\pm$ 0.015 | $\pm$ 0.020 | $\pm$ 0.005 | $\pm$ 0.005 | $\pm$ 0.001 | $\pm$ 0.215 |

$SiO_2$ = Silicon dioxide; $Al_2O_3$ = Aluminium oxide; $Fe_2O_3$ = Iron oxide; $MnO$ = Manganese(II) oxide;
$MgO$ = Magnesium oxide; $CaO$ =Calcium oxide; $Na_2O$ = Sodium oxide; $K_2O$ = Potassium oxide;
$TiO_2$ = Titanium oxide; $P_2O_5$ = Phosphorus pentoxide; $Cr_2O_3$ = Chromium(III) oxide; $V_2O_5$ =
Vanadium(V) oxide; $NiO$ = Nickel(II) oxide; LOI = loss on ignition.

described in ?Farquharson et al. (2016c). Where necessary, a correction was applied to the measured permeability values to account for turbulent flow (see Forchheimer, 1901), the effects of which can become non-negligible when measuring the permeability of high-porosity media. Appendix A contains further details on the determination of permeability, the application of corrections for inertial effects, and the sources and sizes of potential error in the measurements.

Samples were then encased in a copper foil jacket (which serves to retain bulk sample cohesion after deformation), saturated with distilled water, and loaded into the triaxial deformation rig at Université de Strasbourg (see –Figure 1). Throughout deformation we assume a simple effective stress law, whereby the effective confining pressure $p_{eff}$ experienced by a sample is a function of the confining pressure $p_c$ around the sample and the pressure of pore fluid $p_p$ within the sample, such that $p_{eff} = p_c - \alpha \cdot p_p$. Recent experimental work (Farquharson et al., 2016a) shows that $\alpha = 1$ is a reasonable assumption for porous andesite.

For each test, the confining and pore pressures were increased slowly until a targeted effective pressure (i.e. hydrostatic pressurisation). Note that pressure is only controlled within the connected porous network (i.e. $\phi$). However, we assume that any influence of incomplete sample saturation is negligible, due to the relatively small volume of isolated porosity $\phi_u$ in these andesites (see Table 2). Assuming a pycnometry-derived value for bulk density $\rho_b$ of approximately 2100 kg m$^{-3}$, the imposed effective pressures of 10, 30, 50, and 70 MPa are analogous to depths ranging from the upper 500 m of the edifice to greater than 3 km in depth (given that $p_c \propto \rho_b \cdot g\hat{z}$, where $g$ and $\hat{z}$ are surface gravitational acceleration and depth, respectively). The sample would then be left overnight to allow microstructural equilibrium. During the deformation experiments, a differential stress was introduced in the direction of the sample axis by advancing an axial piston (see –Figure 1) under servo-control, such that the sample is subjected to a constant strain rate of 10$^{-5}$ s$^{-1}$ .–(note that hereafter "strain" refers to axial strain unless otherwise specified). We note that – on the edifice-scale – absolute strain rates resulting from magma migration and edifice displacement are generally of the order of 10$^{-7}$ s$^{-1}$ or lower, for example as estimated from borehole strainmeters: Linde et al. (1993), or from spaceborne interferometry: Massonnet et al. (1995). However, strain and strain rates are undoubtedly highly variable throughout active volcanic systems. The chosen strain rate for these experiments (the international standard in rock mechanics Kovari et al., 1983; Ulusay comparable to shear rates inferred to occur along conduit margins by Rust et al. (2003). Similarly, Cashman et al. (2008) estimate strain rates of 3 - 8 $\times$ 10$^{-5}$ s$^{-1}$ for the formation of fault gouge at Mount St Helens (USA). Most importantly, a strain rate of 10$^{-5}$ s$^{-1}$ ensures that our samples are drained (i.e. the product of the Darcy timescale and the strain rate is $\lll$1; Heap

Confining pressure and pore pressure were servo-controlled throughout the experiments. During hydro-static and nonhydrostatic loading, the response of the pore fluid pump reflects variations in pore volume (see Read et al., 1995), which – normalised to the initial sample volume – corresponds to the porosity change $\delta\phi\Delta\phi$. For a porous material, $\delta\phi$–$\Delta\phi$ can be considered equal to the volumetric strain. Indeed, the response of the confining pressure pump provides an independent estimation of the volumetric strain, found to be in perfect agreement with the inferred $\delta\phi$ (see Baud et al. , 2014 for details).–$\Delta\phi$ (see Baud et al., 2014, for details). When this differential $\delta\phi$ $\Delta\phi$ is positive it signifies dilation (an increase in porosity) and when it is negative, it indicates compaction (a decrease in porosity). Deformation was allowed to continue for different amounts of axial strain accumulation – sample shortening relative to its original length ($\varepsilon_t$) – then unloaded. The strain recovered during the unloading phase is subtracted from the total axial strain $\varepsilon_t$ to give the inelastic (non-recoverable) strain accrued by the sample ($\varepsilon_i$). Similarly, the elastic porosity change recovered during unloading is subtracted from the porosity change at $\varepsilon_t$, to give the inelastic porosity change $\delta\phi_i\Delta\phi_i$. Samples were subsequently vacuum-dried once again and gas permeability re-measured.

[Figure]

**Figure 1.** Schematic of triaxial deformation apparatus, including confining pressure $p_c$, pore pressure $p_p$, and axial pressure $p_{ax}$ circuits. Detail of sample assemblage is shown inset. (a) axial piston; (b) "blank" endcap; (c) sample; (d) copper foil jacket; (e) nitrile jacket; (f) pore fluid distributor endcap. Directions of major $\sigma_1$ and minor $\sigma_3$ principal stresses are as shown, such that $\sigma_1 > \sigma_2 = \sigma_3$. Not to scale. Numbered valves allow various parts of each circuit to be used at any given time.

Samples were all deformed at room temperature. We note that in natural volcanic environments there may be some influence of temperature on rock strength (due, for instance, to the closure of cracks driven by thermal expansion). Nevertheless, a recent study by Heap et al. (2017) shows that the influence of temperature on the physical and mechanical properties of andesite may not be significant at temperatures below the glass transition $T_g$. Importantly, this study showed that the failure mode and underlying microstructural mechanism driving ductile behaviour (cataclastic pore collapse) did not change below $T_g$, which is itself largely restricted to the magma conduit and rock in the immediate vicinity. This is in agreement with previous studies by Vinciguerra et al. (2005) and Heap et al. (2014), both of which noted only negligible changes in microcrack density and porosity after thermally stressing volcanic materials. These authors attribute this phenomenon to the high initial crack density resulting from the complex thermal histories of volcanic rocks.

**2.2 Post-deformation permeability**

It has been shown in recent studies  (e.g. Vinciguerra et al., 2005; Nara et al., 2011) that the permeability of fractured volcanic materials is influenced by the effective pressure under which it is measured: permeability tends to decrease with increasing effective pressure. As such, we acknowledge the limitation that post-deformation measurements do not represent the permeability under the deformation conditions *sensu stricto*. Nevertheless, we choose to measure permeability under the conditions described above for a host of reasons. Investigations towards determining  properties (including permeability) other than rock strength indicate that  their evolution with pressure may differ as a function of porosity, pore geometry, and other factors; which is to say that  the effective pressure coefficient for a given rock property may not be the same as the Biot-Willis coefficient $\alpha$  (Bernabé et al., 1986). Given the lack of constraint on  the effective pressure effect for the permeability of volcanic rocks, permeability is measured at the lowest possible confining pressure (1 MPa, rather than at "*in situ*" pressures) and without imposing a differential stress, in order to allow comparison within and between sample sets (indeed, we compare our data with compiled literature data in Sect. 4.1). This procedure also avoids the potential for creep – a mechanism of time-dependent deformation whereby sub-critical crack growth induces damage and possibly even failure at stresses below the short-term strength of the rock  (e.g. Heap et al., 2017) as well as precluding other phenomena such as stress relaxation that may arise when measuring permeability under  a differential stress.

Measuring permeability requires that the sample dimensions, specifically length and cross-sectional area, are  well constrained. Prior to initial measurements of permeability, sample dimensions are measured accurately using digital callipers. However,  samples are often barrelled and thus non-cylindrical after mechanical deformation, making their mean radii nontrivial to determine. Assuming that the solid volume $V_s$ remains constant throughout deformation, then the post-deformation volume is equal to the sum of solid volume, the initial  pore volume, and the pore volume change after deformation. The post-deformation cross-sectional area $A_{post}$ can therefore be determined such that

$$A_{post} = \frac{V_s}{1-[\phi+\delta\phi_i]} \cdot l_{post}^{-1} \underbrace{\left[\frac{V_s}{1-(\phi+\Delta\phi_i)}\right]}_{V_{post}} \times \left[\frac{1}{l_{post}}\right]$$

(3)

where $V_{post}$ is the post-deformation volume and $l_{post}$ is the mean sample length after deformation.

**3 Results**

[revised manuscript text omitted]

$\phi$ = connected porosity; $\phi_u$ = unconnected porosity; $p_{eff}$ = effective pressure; $\varepsilon_t$ = target (total) axial strain; $\varepsilon_i$ = inelastic axial strain; $\Delta\phi_i$ = inelastic porosity change; $\phi + \Delta\phi_i$ = post-deformation porosity; $k_0$ = initial permeability; $k_e$ = post-deformation permeability.

within an "intact" host with the initial rock properties (which is to say, a sample of porosity $\phi$). The intact material must have a pore volume $V_1^\phi$ of $\phi \times (l_{intact} \times A_{post})$, where $l_{intact}$ is the overall length of the sample that is undamaged $(l_{post} - w_b)$, where $w_b$ is the width of the compaction band. The deformed sample contains a pore volume $V_2^\phi$ of $\phi + \Delta\phi_i \times (l_{post} \times A_{post})$. From this, we can relate the compaction band porosity $\phi_b$ to the width of the compaction band:

$$\phi_b = \left[\frac{V_2^\phi - V_1^\phi}{A_{post}}\right] \times \left[\frac{1}{w_b}\right]. \tag{4}$$

Solutions for $\phi_b$ and $w_b$ are non-unique (moreover, a greater value of $w_b$ could be a function of one wide band or a number of discrete, relatively thinner bands), but we can impose a lower bound on $\phi_b$ of zero, and an upper bound equal to the post-deformation porosity of the sample: $0 \leq \phi_b < (\phi + \Delta\phi_i)$. Assuming the compaction band porosity noted by Heap et al. (2015a) ($\sim$0.10) is typical for compacted andesite, then Equation 4 yields compaction band widths of between 1.63 and 23.57 mm (i.e. between 4 and 70 % of the overall sample length). We note that the lower end of this range is in line with the observations of Heap et al. (2015a).

We note that porosity loss is seemingly tied to the effective pressure under which compaction occurs (as observed in previous studies concerned with triaxial rock deformation, e.g. Wong et al., 1997; Baud et al., 2006; Heap et al., 2015a): for a given  increment of inelastic strain, the porosity lost by a sample is  greater at a higher effective pressure. This phenomenon is true both for total porosity change (Figure 2b) and for inelastic porosity loss (Figure 3a), which is to say that the inelastic compaction factor  $\Delta\phi_i/\varepsilon_i$

always decreases as effective pressure increases (Baud et al., 2006).

While the mechanism of cataclastic pore collapse is governed by the pore sizes (e.g. Zhu et al., 2010, 2011), it has also been demonstrated that the local stress field around a pore increases as a function of the incumbent confining pressure (Zhu et al., 2010). A  study of fault gouge formation in sandstone (Engelder, 1974) shows that fault-zone fragments are smaller when generated at higher confining pressures, and a similar effect (albeit less pronounced) was noted by Kennedy et al. (2012), who investigated fault gouge formation in dacitic dome rock.  It is reasonable to assume that cataclasis may become more efficient as the local stress field increases in line with the confining pressure; in turn, a finer distribution of fragments will more readily occlude the pores around which they develop. Whether a change in the mean fragment size generated during cataclastic pore collapse underlies the observed evolution of  $\Delta\phi_i/\varepsilon_i$ remains open to a targeted microstructural study. Nevertheless, we can interrogate Equation 4 to glean an idea of the effect of $\phi_b$ and $w_b$ at different effective pressures and axial strains.

Figure 5 shows the calculated $w_b$ for our experimental data as a function of inelastic axial strain. Values of $w_b$ are calculated using Equation 4, using values of $\phi_b$ of 0.20, 0.15, 0.10, and 0.05. At relatively higher imposed values of $\phi_b$ (0.20 and 0.15: Figure 5a, 5b), many of the resulting values of $w_b/l$ are non-physical (i.e. $w_b/l \not> 0$ or $w_b/l \not< 1$). However, at lower imposed values of $\phi_b$ (0.10 and 0.05: Figure 5c, 5d), values fall between 0 and 1. Moreover, there appears to be a systematic effect of $p_{eff}$, with deformation under relatively higher effective pressure yielding a higher ratio of $w_b/l$ – hence, a thicker compaction band – for any given amount of inelastic axial strain accumulation.

[Figure]

**Figure 3.** Change in physical properties after accumulating inelastic axial strain. (a) Pre- and post-deformation porosity (the latter calculated by $\phi + \delta\phi_i$ $\phi + \Delta\phi_i$) versus inelastic strain. Effective pressure is indicated by the symbol colour. (b) Pre- and post-deformation gas permeability as a function of the inelastic strain accumulated by each sample. Effective pressure is indicated by colour as in panel (a): note, however, that the value of $p_{eff}$ corresponds to deformation, whereas permeability was measured in each case at $p_{eff} = 1$ MPa.

As would be expected (e.g. Zhu and Wong, 1997), permeability reduction follows the same general trend

[Figure]

**Figure 4.** Backscattered scanning electron microscope images of LLB andesite, showing as-collected (a) and post-deformation (b - c) microstructure. Void space appears as black. Dense (metal-rich) phenocrysts appear as white or light grey within a darker grey groundmass. Both (b) and (c) are images of LLB-13, which was taken to beyond both $C^*$ and $C^{*\prime}$. Cataclastic pore collapse associated with shear-enhanced compaction is shown in (b), while (c) shows part of the dilatant shear zone marking the transition from shear-enhanced compaction to dilation.

[Figure]

**Figure 5.** Calculated sample length ratio $w_b/l$ as a function of inelastic axial strain accumulation $\varepsilon_i$. Values of $w_b$ calculated from Equation 4 using values of compaction band porosity of $\phi_b = 0.20$ (a), $\phi_b = 0.15$ (b), $\phi_b = 0.10$ (c), and $\phi_b = 0.05$ (d). Shaded area indicates range of physical values of $w_b/l$.

(—Figure 3b) as porosity reduction (—Figure 3a), with samples accumulating high strains showing a correspondingly large reduction in permeability.  Notably, there appears to be an influence of the effective pressure under which the sample was deformed and the change in permeability for a given increment of axial strain. A difference in measured post-deformation permeability $k_e$ may be due to (1) a variation in characteristic grain size, or (2) a variation in the thickness of compaction localisation features with respect to the sample length, as described above. Moreover, one may imagine that these two factors (characteristic grain size, band thickness) operate in tandem to reduce permeability as effective pressure increases. To test this theory, we again model the deformed samples as a layered medium, such that discrete bands of uniform permeability $k_b$ and thickness $w_b$ are embedded in a medium of permeability $k_0$ (see Figure 6a). A similar approach has previously been adopted by Vajdova et al. (2004) and Baud et al. (2012) to model the permeability of sandstones containing experimentally-induced compaction bands. Fluid flow through this simplified geometry may then be modelled by assuming conservation of mass (e.g. Freeze and Cherry, 1979), such that:

$$k_e = k_0 \times \left[ \left( \frac{w_b}{l} \right) \left( \frac{k_0}{k_b} - 1 \right) + 1 \right]^{-1} \qquad (5)$$

Variables $k_e$, $k_0$, and $l$ are already constrained, allowing us to solve for combinations of $w_b$ and $k_b$. If we assume that a compaction band comprises a granular bed (Figure 6a *inset*), we can relate its permeability $k_b$ to surface area $s$ in the following manner after Martys et al. (1994), who determined a universal scaling of permeability of a system of packed spheres:

$$k_b = \frac{2(1 - \phi^*)}{s^2} \phi^{*f} \qquad (6)$$

where $\phi^* = \phi_b - \phi_c$, and $f = 4.2$ (the value of $f$ is thought to be related to the initial grain geometry: Wad $\phi_c$ represents the percolation threshold, taken here as 0.03. Note that the characteristic porosity is taken as the porosity within a compaction band $\phi_b$ (i.e. the porosity of the granular layer with permeability $k_b$ and width $w_b$). In turn, we can relate $s$ to a characteristic grain size (e.g. Wadsworth et al., 2016):

$$s(r) = \frac{3(1 - \phi_b)}{r} \qquad (7)$$

where $r$ is the monodisperse particle radius. Note that in reality, the porosities assumed within the compaction bands here are not compatible with a monodisperse packing of spheres. Nevertheless, this greatly simplified approach gives an indication of the relative influence of the difference constituent parameters $r$, $\phi_b$, and $w_b$.

The assumed geometry is illustrated in Figure 6a, including the corresponding values of permeability and porosity for each layer. Figure 6b highlights the effects of changing either the characterisitic particle (i.e. grain) radius or the compaction band porosity. Notably – for a given porosity – a change in particle radius of 1 order of magnitude results in a change in compaction band permeability of 2 orders of magnitude. At relatively high initial porosities, a reduction in porosity by a given volume (for example, from 0.20 to 0.15) has little influence on $k_b$. On the other hand, when the porosity is low, a change in porosity of the same absolute volume (for example, from 0.10 to 0.05) exerts a much greater influence over $k_b$ (in this case, a reduction

by over 2 orders of magnitude). However, the bulk sample permeability (the equivalent permeability) depends not only on the porosity of the compaction band but also its width. Figures 6c – 6f show the equivalent permeability for different values of $\phi_b$ for changing values of $w_b/l$: the ratio of the band width to the overall length of the sample. Curves are modelled by combining Equations 5, 6, and 7, using the particle radii $r$ (noted on each figure panel) and a value of $k_0 = 5.0 \times 10^{-13}$ m$^2$. As there are multiple non-unique solutions for $\phi_b$ and $w_b$, we show model results for a range of potential $\phi_b$ values in panels 6c to 6f.

Evidently, the variation in our experimental data (for example, the difference between $k_e$ of samples deformed under different effective pressures) is not explained by a systematic evolution of $r$. This suggests that while $k_b$ is very sensitive to the characteristic grain radius, the tradeoff between $\phi_b$ and $w_b$ is more important in controlling the bulk sample permeability ($k_e$). Importantly, however, idealising the geometry of a compaction band in terms of a monodisperse particle size distribution cannot accurately represent its complex porous network. More accurate values of $\phi_b$ and $s$ (and hence, a better prediction of $k_b$ and $k_e$) may be acheived by adopting a polydisperse particle size distribution or by imposing a non-spherical characteristic particle shape, for example.

An evident weakness of employing the simple layered medium model outlined above is that we assume that the only operative mechanism is porosity- and permeability-reducing. However at low strains, there is no one-to-one relationship between permeability and porosity after deformation. Rather, permeability tends to increase moderately at inelastic axial strains less than around 0.05 (i.e. a 5% shortening in sample length); while initial values of permeability tended to be around $5 \times 10^{-13}$ m$^2$, the measured post-deformation permeability was often greater than $10^{-12}$ m$^2$ after accumulating a small amount of axial strain (Table 2). A similar phenomenon was also observed by Loaiza et al. (2012), who noted that permeability of Açores trachyandesite increased beyond a critical stress state during hydrostatic pressurisation. This critical stress – known as $P^*$: Zhang et al. (1990) – signals the onset of lithostatic inelastic compaction; Loaiza et al. (2012) show that stress-induced cracks coalesce between collapsed pores during hydrostatic compaction, improving connectivity and, in turn, increasing permeability. Prior to deformation, the samples of porous andesite used in our experiments – LLB – contained an isolated porosity of 0.01, on average (Table 2).

Similar to the mechanism posited by Loaiza et al. (2012), we suggest that distributed microcracking during the initial stages of ductile deformation serves to interconnect this isolated porosity, creating efficient pathways for fluid flow. The mechanical data of all the experiments (Figure 1a) exhibit intermittent stress drops – even in the instances where permeability was observed to increase relative to the initial value – which suggests

that compaction localisation in these andesites does not necessarily equate to the formation of an effective barrier to fluid flow. Loaiza et al. (2012), Adelinet et al. (2013), Heap et al. (2015a), and Heap et al. (2016) each examine microstructure of compaction bands formed in porous volcanic rocks (trachyandesite, basalt, andesite, and dacite, respectively). In all cases, the bands are irregular in shape and thickness, but do not necessarily constitute a contiguous surface of collapse pores (i.e. a layer of reduced porosity). This is supported by the results from a recent study by Baud et al. (2015), which examined compaction band-bearing sandstones using X-ray computed tomography. The authors show that when compaction bands are formed in a rock with porosity clusters, their path is more tortuous than in material with homogeneous porosity; consequently the bands do not comprise efficient permeability barriers in three dimensions. Notably, the porosity of the andesites deformed in our study exhibit marked heterogeneity in terms of porosity, pore shape, and pore size distribution (Figure 4a). The characteristic tortuosity of compaction bands formed in heterogeneous volcanic material – and the fact that the reduction in porosity relative to the host sample is remarkably less than observed in sandstones (Baud et al., 2012) – may explain the lack of an obvious observed influence on sample permeability in this study. Thus, counter-intuitively, small amounts of stress-induced compaction may actually increase permeability in volcanic materials. At higher strains however, this effect is overtaken by the global reduction in sample porosity, which serves to decrease the mean flow path aperture and forces fluids to travel through more tortuous routes.

**4.2 A limit to compaction and permeability reduction**

Porosity exerts a first order control on the brittle-ductile transition of porous rocks (e.g. Wong and Baud, 2012): high porosity fosters ductile behaviour in response to an applied differential stress whereas dilatant brittle behaviour is anticipated in low-porosity materials. If a high-porosity volcanic rock undergoes progressive compaction, it will eventually achieve a porosity low enough to respond in a dilatant fashion. The critical stress state at this transition is known as $C^{*\prime}$ has been previously described in sedimentary materials (e.g. Schock et al., 1973; Baud et al., 2000; Vajdova et al., 2004; Baud et al., 2006; Regnet et al., 2015), and recently in andesite from Volcán de Colima (Heap et al., 2015a). As anticipated, beyond a threshold stress-strain accumulation, sample LLB-13 exhibits a transition from compactant deformation to dilatant behaviour (highlighted by the arrow in Figure 1b). In volcanic rock – at the sample-scale – this is characterised by significant shortening and barrelling of the sample (a consequence of the high strains required to achieve and exceed $C^{*\prime}$) and the generation of a dilatant shear zone (Heap et al., 2015a), characteristically similar to

[Figure]

**Figure 6.** Model geometry and results. (a) Deformed sample may be thought of as a granular bed (with permeability $k_b$ and porosity $\phi_b$) within a sample matrix with permeability $k_0$ and porosity $\phi + \Delta\phi_i$. Inset highlights the characterisitc grain radius that governs surface area (Equation 7). (b) The permeability of a compaction band composed of packed spheres as a function of sphere radius and bed porosity. Note that in reality, porosities $< 0.26$ would require a polydisperse packing of spheres, or a granular bed composed of non-spherical grains. (c) – (f) Modelled equivalent permeability $k_e$ of a compaction band-bearing sample, plotted against the ratio of compaction band width relative to the overall sample length ($w_b/l$), for compaction band porosities of $\phi_b = 0.20$ (c), $\phi_b = 0.15$ (d), $\phi_b = 0.10$ (e), and $\phi_b = 0.05$ (f). Also plotted are data from this study (Table 2), where values of $w_b$ are calculated after Equation 4. Note that of the ten $k_e, w_b/l$ data pairs from this study, not all are plotted on each of the panels (c) to (f). This is because certain combinations of $\phi_b$ and $w_b$ yield non-physical values (Equation 4). Refer to text for further discussion.

[revised manuscript text omitted]
, possibly due to enhanced communution. By modelling a simple sample geometry where a compaction band is represented by a packed granular bed, we show that the permeability reduction within a discrete compaction band is sensitive to the characteristic grain size. The effect of the porosity of a compaction band has a variable influence on its permeability, with changes in band porosity becoming ever more important as porosity decreases. However, the overall trade-off between the width and porosity of a compaction band are far more important that grain size in controlling fluid flow throughout the bulk of a sample. At low strains ($< 0.05$), compaction tends to result in a moderate increase in permeability (not accounted for in our model), which we suggest is a result of increased pore connectivity due to distributed microcracking. This effect is outweighed by progressive compaction at higher strains, resulting in a general trend of decreasing permeability with ongoing inelastic compaction. There exists a physical limit to compaction, which we suggest is echoed in a limit to the potential for permeability reduction in a deforming sample. Compiled data show that at high strain, porosity and permeability tend to converge towards intermediate values (i.e. $0.10 \leq \phi \leq 0.20$; $10^{-14} \leq k \leq 10^{-13}$ m$^2$). Field evidence from the literature emphasises the importance of understanding the physical and mechanical properties of rock in active volcanic environments, in particular the evolution in a rock's capacity to effectively transmit magmatic volatiles.

**6   Data availability**

Data are presented in Table 1 and  2 in the text, or are available on request to the corresponding author.

**Appendix A:  Measuring permeability**

The constitutive equation governing fluid transport in porous and granular media was originally derived from experiments performed by Henry Darcy in the 1850s on the flow of water at different levels through sand. Since Darcy's work (1856), the theoretical framework of fluid transport—which is based on Newton's second law—has been well established and expanded, such that flow of gas through a porous medium may be given thus:

$$Q_v = \frac{-kA}{\mu} \frac{(p_b - p_a)}{l} \tag{A1}$$

where $\mu$ is the fluid viscosity, $Q_v$ is the volumetric flow rate, $A$ is the cross-sectional area available for flow and $l$ is the distance over which fluid flow occurs (i.e. the sample length). In a fluid transport system, flow is driven towards the region of lowest potential energy: in the special case of horizontal flow, this may be described by a differential between a region of relatively high pressure $p_b$ to one of relatively lower pressure $p_a$: a pressure differential or pressure drop $\nabla p$. Equation A1 is valid for all porous media as long as flow is laminar (two cases of non-laminar flow are discussed hereafter). While this expression is sufficient for the case of laminar (or "streamline") flow, when considering an ideal compressible gas measured under atmospheric conditions, it becomes convenient to present gas permeability $k_{gas}$ in the following manner (Klinkenberg, 1941; McPhee and Arthur, 1991):

$$k_{gas} = \frac{Q_v \mu l \cdot p_{atm}}{A \cdot \nabla p \bar{p}} \tag{A2}$$

where $p_{atm}$ is the atmospheric pressure at which $Q_v$ is measured, and the driving pressure is given as a product of the differential pressure $\nabla p$ and the mean pressure over the sample $\bar{p}$. The mean pressure $\bar{p}$ is determined

by the upstream and downstream pressures $p_b$ and $p_a$ such that $\bar{p} = (p_b + p_a)/2$. Under ambient conditions, $p_a$ is equal to the atmospheric pressure $p_{atm}$, and $p_b$ is equal to $\nabla p + p_{atm}$. The mean pressure therefore simplifies to $\bar{p} = (\nabla p + 2p_{atm})/2$.

Herein, gas permeability was measured using a steady-state permeameter using the setup described in Farquharson et al. (2016c). The apparatus is a commercial benchtop permeameter from Vinci Technologies, modified by incorporating interchangeable El-Flow volumetric mass flowmeters (from Bronkhorst) to measure the volumetric flow rate of gas at the downstream end of the experimental samples. Gas permeability was measured using nitrogen as the permeant (pore fluid). A confining pressure of 1 MPa was applied radially to the sample, in order to ensure that no leakage occurred along its margins during measurement. The sample was then left under this confining pressure for 1 hour, to allow for any necessary microstructural equilibrium. Gas would then be flowed through the sample, whilst the volumetric flow rate $Q_v$ and the pressure differential $\nabla p$ across the sample were continuously monitored by means of customised data acquisition system and a LabVIEW program written for this purpose.

The pressure of gas entering the sample could be adjusted using a regulator attached to the permeant gas bottle. By altering the flow of gas, a range of different values of $\nabla p$ was imposed across the samples (typically between 0.001 and 0.2 MPa). Once steady-state flow was achieved, the volumetric flow rate was noted. Thus with knowledge of the gas viscosity and sample dimensions, permeability could be calculated using Equation A2. However, two scenarios can necessitate that post-measurement corrections need to be applied to the calculated values due to inertial effects: flow turbulence or gas slippage.

**A1    Non-laminar flow 1: Turbulence**

Forchheimer (1901) conducted fluid flow experiments through porous media, noting that the relationship between the pressure differential $\nabla p$ and the volumetric flow rate $Q_v$ becomes nonlinear at high fluid velocities due to flow no longer being laminar. To account for this turbulence, an inertial term, here denoted $\iota$, must be introduced, such that:

$$\frac{1}{k_{fo}} = \frac{1}{k_{gas}} - \iota \cdot Q_v \tag{A3}$$

where $k_{fo}$ is the Forchheimer-corrected permeability value, and $k_{gas}$ is the as-measured value. In this scenario, the measured gas permeability would be lower than the true (corrected) permeability, as turbulence induces resistance to fluid flow.

**A2    Non-laminar flow 2: Gas slippage**

In his seminal 1941 paper, Klinkenberg showed that as the characteristic pore size or aperture approaches the mean free path of the permeant gas—the distance travelled between consecutive molecular collisions—interactions between the gas molecules and the pore (or crack) walls serve to reduce resistance to flow. Simply put, during liquid laminar flow, the layer of molecules adjacent to the pore (or crack) walls is static. However, for gases this molecule layer has a nonzero velocity due to molecular diffusion ("slip"). This slippage results in a higher flow rate at any given pressure differential for a gas than a liquid. Accordingly, the permeability measured using a gas would be artificially higher than if determined using a liquid.

The relationship of Klinkenberg (1941) is incorporated thus:

$$k_{gas} = k_{kl}\left(1 + \frac{b}{\bar{p}}\right) \tag{A4}$$

where $k_{gas}$ is the as-measured permeability calculated from gas flow experiments (note that in cases where a Forchheimer correction has been applied as in Equation A3, $k_{gas}$ is substituted by $k_{fo}$ in Equation A4), $\bar{p}$ is the mean flow pressure of gas in the system, $b$ is the Klinkenberg parameter (which depends on both the gas used but also the pore structure), and $k_{kl}$ is the Klinkenberg-corrected permeability value.

In the absence of inertial effects, plotting $Q_v$ against the driving pressure (i.e. $\nabla p \bar{p}$) yields a linear relationship. Deviations from linear behaviour indicate that one or both of the inertial phenomena described above are influencing the calculated permeability. In practice, the corrected permeability can be calculated using the slope and intercept of graphs of $Q_v$ against $k_{gas}^{-1}$, and $\bar{p}^{-1}$ against $k_{fo}$. In the main body of the text measured permeability, corrected for turbulence and/or gas slippage when necessary, is always presented as $k$. Figure A1 shows example data from volcanic rocks which exhibit laminar flow, turbulence, and gas slippage, respectively. The effects of inertial flow, especially in the case of gas slippage tends to be slight, although non-negligible.

**A3    Permeability and experimental error**

Sources of error in the permeability measurements include the sample dimensions, and the resolution of the pressure transducer and flowmeters. As mentioned, permeability is determined as a function of the $Q_v - \nabla p \bar{p}$ curve, which is a series of points fit by a simple linear regression (when flow is laminar). The respective precision of the transducer and flowmeter is thus encompassed by the coefficient of determination of the regression line (i.e. its $r^2$ value). If the data are unaffected by turbulence or gas slippage, then $r^2$ is generally greater than 0.99. If flow is nonlaminar, $r^2$ tends to be appreciably lower, and the permeability is

determined using Equation A3 or A4 as appropriate. Repeat measurements suggest that experimental error is always engirdled by the symbol size when plotted graphically.

Figure A1 shows flowrate and pressure data obtained during steady-state permeability measurements on three volcanic samples. For the first example, Figure A1a - c, flow is laminar, as evident from the linear relation between the volumetric flowrate ($Q_v$) and the driving pressure ($\nabla p\bar{p}$). Accordingly, the reciprocal permeability ($k_{gas}^{-1}$) versus $Q_v$ is negative and nonlinear, as is the measured permeability $k_{gas}$ against the reciprocal mean pressure $\bar{p}^{-1}$. In the second example, flow is turbulent, and the data in Figure A1d are nonlinear. Applying the correction derived from Figure A1e (Equation A3) yields Figure A1f, where the data are randomly distributed about the mean (no Klinkenberg correction is necessary). Finally, the data shown in Figure A1g - h highlight that a Klinkenberg correction is necessary (Equation A4). However, the correction is very slight, as indeed often tends to be the case in volcanic rocks (as opposed to "tight" materials such as granite). For the LLB data presented in Table 2, Forchheimer corrections were applied to the raw values where appropriate. Permeability values were affected only slightly, increasing by a factor of between 1.03 to 1.41.

*Author contributions.* J.F., P.B., and M.H. designed the experiments, which were performed by J.F. J.F. prepared the manuscript with contributions from all authors.

*Competing interests.* The authors declare that they have no conflict of interest.

*Acknowledgements.* Thierry Reuschlé, Alex Schubnel, Luke Griffiths, and Alexandra Kushnir are thanked for inspiring discussions. Fieldwork was funded in part by the framework of the LABEX ANR-11-LABX-0050_G-EAU-THERMIE-PROFONDE and therefore benefits from a funding from the state managed by the French National Research Agency as part of the Investments for the future program. JIF acknowledges an Initiative d'Excellence (IDEX) "Contrats doctoraux" grant from the French State. MJH acknowledges Initiative d'Excellence (IDEX) Attractivité grant "VOLPERM" and a CNRS INSU grant. We are grateful to Nick Varley and Oliver Lamb for field assistance at Volcán de Colima. Fabian Wadsworth and Antonella Longo are thanked for their constructive comments on the manuscript.


[Figure]

**Figure A1.** Data obtained during permeability measurements on three rocks. a) - c) Laminar flow in a sample from Whakaari, New Zealand. d) - f) Turbulent flow, in a sample of Ruapehu andesite (New Zealand). g) - i) A small gas slippage effect in a sample of Açores trachyandesite.

Commer, M., Helwig, S.L., Hördt, A. and Tezkan, B., 2005. Interpretation of long-offset transient electromagnetic data from Mount Merapi, Indonesia, using a three-dimensional optimization approach. Journal of Geophysical Research: Solid Earth, 110(B3).

Cubellis, E., Ferri, M. and Luongo, G., 1995. Internal structures of the Campi Flegrei caldera by gravimetric data. Journal of Volcanology and Geothermal Research, 65(1), pp.147-156.

Darcy, H.P.G., 1856. Les Fontaines publiques de la ville de Dijon. Exposition et application des principes à suivre et des formules à employer dans les questions de distribution d'eau. [Exhibition and implementation of the principles to follow and to formulae employ in the issue of water distribution.] Victor Dalmont, France. (In French).

Day, S.J., 1996. Hydrothermal pore fluid pressure and the stability of porous, permeable volcanoes. Geological Society, London, Special Publications, 110(1), pp.77-93.

Delcamp, A., Roberti, G. and de Vries, B.V.W., 2016. Water in volcanoes: evolution, storage and rapid release during landslides. Bulletin of Volcanology, 78(12), p.87

Denlinger, R.P. and Hoblitt, R.P., 1999. Cyclic eruptive behavior of silicic volcanoes. Geology, 27(5), pp.459-462.

Dieterich, J.H., 1988. Growth and persistence of Hawaiian volcanic rift zones. Journal of Geophysical Research: Solid Earth, 93(B5), pp.4258-4270.

Dzurisin, D., 2003. A comprehensive approach to monitoring volcano deformation as a window on the eruption cycle. Reviews of Geophysics,41(1).

Edmonds, M. and Herd, R.A., 2007. A volcanic degassing event at the explosive-effusive transition. Geophysical Research Letters, 34(21).

Eichelberger, J. C., Carrigan, C. R., Westrich, H. R., and Price, R. H., 1986. Non-explosive silicic volcanism. Nature, 323(6089), 598-602.

Engelder, J.T., 1974. Cataclasis and the generation of fault gouge. Geological Society of America Bulletin, 85(10), pp.1515-1522.

Faoro, I., Vinciguerra, S., Marone, C., Elsworth, D. and Schubnel, A., 2013. Linking permeability to crack density evolution in thermally stressed rocks under cyclic loading. Geophysical Research Letters, 40(11), pp.2590-2595.

Farquharson, J., Heap, M.J., Baud, P., Reuschlé, T. and Varley, N.R., 2016a. Pore pressure embrittlement in a volcanic edifice. Bulletin of Volcanology,78(1), pp.1-19.

Farquharson, J., Heap, M.J., Varley, N.R., Baud, P. and Reuschlé, T., 2015. Permeability and porosity relationships of edifice-forming andesites: a combined field and laboratory study. Journal of Volcanology and Geothermal Research, 297, pp.52-68.

Farquharson, J.I., Heap, M.J. and Baud, P., 2016b. Strain-induced permeability increase in volcanic rock. Geophysical Research Letters.

Farquharson, J.I., Heap, M.J., Lavallée, Y., Varley, N.R. and Baud, P., 2016c. Evidence for the development of permeability anisotropy in lava domes and volcanic conduits. Journal of Volcanology and Geothermal Research, 323, pp.163-185.

Finn, C.A., Deszcz-Pan, M., Anderson, E.D. and John, D.A., 2007. Three-dimensional geophysical mapping of rock alteration and water content at Mount Adams, Washington: Implications for lahar hazards. Journal of Geophysical Research: Solid Earth, 112(B10).

Freeze, R.A. and Cherry, J.A., 1979. Groundwater. Englewood.

Forchheimer, P., 1901. Wasserbewegung durch boden. [Water movement through soil.] Zeitschrift des Vereines Deutscher Ingenieure, 45(1782), p.1788. (In German).

Fortin, J., Stanchits, S., Vinciguerra, S. and Guéguen, Y., 2011. Influence of thermal and mechanical cracks on permeability and elastic wave velocities in a basalt from Mt. Etna volcano subjected to elevated pressure. Tectonophysics, 503(1), pp.60-74.

Gaunt, H. E., Sammonds, P. R., Meredith, P. G., Smith, R., and Pallister, J. S., 2014. Pathways for degassing during the lava dome eruption of Mount St. Helens 2004–2008. Geology, 42(11), 947-950.

Gonnermann, H.M. and Manga, M., 2013. Dynamics of magma ascent in the volcanic conduit. Modeling Volcanic Processes: The physics and mathematics of volcanism, p.55.

Gottsmann, J., Camacho, A.G., Martí, J., Wooller, L., Fernández, J., Garcia, A. and Rymer, H., 2008. Shallow structure beneath the Central Volcanic Complex of Tenerife from new gravity data: Implications for its evolution and recent reactivation. Physics of the Earth and Planetary Interiors, 168(3), pp.212-230.

Heap, M.J., Lavallée, Y., Petrakova, L., Baud, P., Reuschle, T., Varley, N.R. and Dingwell, D.B., 2014. Microstructural controls on the physical and mechanical properties of edifice-forming andesites at Volcán de Colima, Mexico. Journal of Geophysical Research: Solid Earth, 119(4), pp.2925-2963.

Heap, M.J., Baud, P., Meredith, P.G., Vinciguerra, S., Bell, A.F. and Main, I.G., 2011. Brittle creep in basalt and its application to time-dependent volcano deformation. Earth and Planetary Science Letters, 307(1), pp.71-82.

Heap, M.J., Brantut, N., Baud, P. and Meredith, P.G., 2015c. Time-dependent compaction band formation in sandstone. Journal of Geophysical Research: Solid Earth, 120(7), pp.4808-4830.

Heap, M.J., Farquharson, J.I., Baud, P., Lavallée, Y. and Reuschlé, T., 2015a. Fracture and compaction of andesite in a volcanic edifice. Bulletin of volcanology, 77(6), pp.1-19.

Heap, M.J., Kennedy, B.M., Pernin, N., Jacquemard, L., Baud, P., Farquharson, J.I., Scheu, B., Lavallée, Y., Gilg, H.A., Letham-Brake, M. and Mayer, K., 2015b. Mechanical behaviour and failure modes in the Whakaari (White Island volcano) hydrothermal system, New Zealand. Journal of Volcanology and Geothermal Research, 295, pp.26-42.

Heap, M.J., Russell, J.K. and Kennedy, L.A., 2016. Mechanical behaviour of dacite from Mount St. Helens (USA): A link between porosity and lava dome extrusion mechanism (dome or spine)?. Journal of Volcanology and Geothermal Research.

Heap, M.J. and Wadsworth, F.B., 2016. Closing an open system: pore pressure changes in permeable edifice rock at high strain rates. Journal of Volcanology and Geothermal Research, 315, pp.40-50.

Heap, M.J., Violay, M., Wadsworth, F.B. and Vasseur, J., 2017. From rock to magma and back again: The evolution of temperature and deformation mechanism in conduit margin zones. Earth and Planetary Science Letters, 463, pp.92-100.

Heimisson, E. R., Einarsson, P., Sigmundsson, F., and Brandsdóttir, B., 2015. Kilometer-scale Kaiser effect identified in Krafla volcano, Iceland. Geophysical Research Letters, 42(19), pp.7958-7965

Hurwitz, S., Kipp, K.L., Ingebritsen, S.E. and Reid, M.E., 2003. Groundwater flow, heat transport, and water table position within

volcanic edifices: Implications for volcanic processes in the Cascade Range. Journal of Geophysical Research: Solid Earth, 108(B12).

Jónsson, G. and Stefánsson, V., 1982. Density and porosity logging in the IRDP hole, Iceland. Journal of Geophysical Research: Solid Earth, 87(B8), pp.6619-6630.

Kempter, K.A., Benner, S.G. and Williams, S.N., 1996. Rincón de la Vieja volcano, Guanacaste province, Costa Rica: geology of the southwestern flank and hazards implications. Journal of volcanology and geothermal research, 71(2), pp.109-127.

Kennedy, L.A. and Russell, J.K., 2012. Cataclastic production of volcanic ash at Mount Saint Helens. Physics and Chemistry of the Earth, Parts A/B/C, 45, pp.40-49.

Kiyama, T., Kita, H., Ishijima, Y., Yanagidani, T., Aoki, K. and Sato, T., 1996, January. Permeability in anisotropic granite under hydrostatic compression and triaxial compression including post-failure region. In 2nd North American Rock Mechanics Symposium. American Rock Mechanics Association.

Klinkenberg, L.J., 1941. The permeability of porous media to liquids and gases. Drilling and production practice. American Petroleum Institute.

Kovari, K., Tisa, A., Einstein, H.H. and Franklin, J.A., 1983. Suggested methods for determining the strength of rock materials in triaxial compression: revised version. International journal of rock mechanics and mining sciences, 20(6), pp.283-290.

Kushnir, A.R., Martel, C., Bourdier, J.L., Heap, M.J., Reuschlé, T., Erdmann, S., Komorowski, J.C. and Cholik, N., 2016. Probing permeability and microstructure: Unravelling the role of a low-permeability dome on the explosivity of Merapi (Indonesia). Journal of Volcanology and Geothermal Research, 316, pp.56-71.

Lavallée, Y., Heap, M.J., Kueppers, U., Kendrick, J.E., and Dingwell, D.B., 2017. The fragility of Volcán de Colima—a material constraint. In: Varley, N., and Komorowski, J.C. (eds) Volcán de Colima: managing the threat. Springer, Berlin.

Le Cloarec, M.F. and Gauthier, P.J., 2003. Merapi Volcano, Central Java, Indonesia: A case study of radionuclide behavior in volcanic gases and its implications for magma dynamics at andesitic volcanoes. Journal of Geophysical Research: Solid Earth, 108(B5).

Le Pennec, J.L., Hermitte, D., Dana, I., Pezard, P., Coulon, C., Cochemé, J.J., Mulyadi, E., Ollagnier, F. and Revest, C., 2001. Electrical conductivity and pore-space topology of Merapi lavas: implications for the degassing of porphyritic andesite magmas. Geophysical Research Letters, 28(22), pp.4283-4286.

Li, N., Wu, H., Feng, Q., Wang, K., Shi, Y., Li, Q. and Luo, X., 2009. Matrix porosity calculation in volcanic and dolomite reservoirs and its application. Applied Geophysics, 6(3), p.287.

Linde, A.T., Agustsson, K., Sacks, I.S. and Stefansson, R., 1993. Mechanism of the 1991 eruption of Hekla from continuous borehole strain monitoring. Nature, 365(6448), p.737.

Loaiza, S., Fortin, J., Schubnel, A., Gueguen, Y., Vinciguerra, S. and Moreira, M., 2012. Mechanical behavior and localized failure modes in a porous basalt from the Azores. Geophysical Research Letters, 39(19).

Martys, N.S., Torquato, S. and Bentz, D.P., 1994. Universal scaling of fluid permeability for sphere packings. Physical Review E, 50(1), p.403.

Massonnet, D., Briole, P. and Arnaud, A., 1995. Deflation of Mount Etna monitored by spaceborne radar interferometry. Nature, 375(6532), p.567.

McPhee, C.A. and Arthur, K.G., 1991. Klinkenberg permeability measurements: problems and practical solutions. In Advances in Core Evaluation IL Reservoir Appraisal. Proceedings of the 2nd Society of Core Analysts European Core Analysis Symposium. Gordon & Breach Science Publishers, Philadelphia (pp. 371-391).

Melnik, O. and Sparks, R.S.J., 1999. Nonlinear dynamics of lava dome extrusion. Nature, 402(6757), pp.37-41.

Melnik, O. and Sparks, R.S.J., 2005. Controls on conduit magma flow dynamics during lava dome building eruptions. Journal of Geophysical Research: Solid Earth, 110(B2).

Millett, J.M., Hole, M.J., Jolley, D.W., Schofield, N. and Campbell, E., 2016. Frontier exploration and the North Atlantic Igneous Province: new insights from a 2.6 km offshore volcanic sequence in the NE Faroe–Shetland Basin. Journal of the Geological Society, 173(2), pp.320-336.

Mitchell, T.M. and Faulkner, D.R., 2008. Experimental measurements of permeability evolution during triaxial compression of initially intact crystalline rocks and implications for fluid flow in fault zones. Journal of Geophysical Research: Solid Earth, 113(B11).

Mogi, K. 1958. Relations between the eruptions of various volcanoes and the deformation of the ground surfaces around them. Bulletin of the Earthquake Research Institute, 36, pp. 99–114

Mordecai, M., Morris, L.H. and Eng, C., 1970, January. An investigation into the changes of permeability occurring in a sandstone when failed under triaxial stress conditions. In The 12th US Symposium on Rock Mechanics (USRMS). American Rock Mechanics Association.

Mueller, S., Scheu, B., Kueppers, U., Spieler, O., Richard, D. and Dingwell, D.B., 2011. The porosity of pyroclasts as an indicator of volcanic explosivity. Journal of Volcanology and Geothermal Research, 203(3), pp.168-174.

Mueller, S., Scheu, B., Spieler, O. and Dingwell, D.B., 2008. Permeability control on magma fragmentation. Geology, 36(5), pp.399-402.

Nara, Y., Meredith, P.G., Yoneda, T. and Kaneko, K., 2011. Influence of macro-fractures and micro-fractures on permeability and elastic wave velocities in basalt at elevated pressure. Tectonophysics, 503(1), pp.52-59.

Nguyen, C. T., Gonnermann, H. M., and Houghton, B. F., 2014. Explosive to effusive transition during the largest volcanic eruption of the 20th century (Novarupta 1912, Alaska). Geology, 42(8), 703-706.

Okumura, S., and Sasaki, O., 2014. Permeability reduction of fractured rhyolite in volcanic conduits and its control on eruption cyclicity. Geology, 42(10), 843-846.

Ōmori, F., 1920. Seismographical Observations of the Fore-shocks, After-shocks, and After-outbursts of the Great Sakura jima-Eruption of 1914. Bulletin of the Imperial Earthquake Investigation Committee, 8(5), pp.353-377.

Peach, C.J. and Spiers, C.J., 1996. Influence of crystal plastic deformation on dilatancy and permeability development in synthetic salt rock. Tectonophysics, 256(1), pp.101-128.

Read, M.D., Ayling, M.R., Meredith, P.G. and Murrell, S.A., 1995. Microcracking during triaxial deformation of porous

rocks monitored by changes in rock physical properties, II. Pore volumometry and acoustic emission measurements on water-saturated rocks. Tectonophysics, 245(3-4), pp.223-235.

Regnet, J.B., David, C., Fortin, J., Robion, P., Makhloufi, Y. and Collin, P.Y., 2015. Influence of microporosity distribution on the mechanical behavior of oolithic carbonate rocks. Geomechanics for Energy and the Environment, 3, pp.11-23.

Rust, A.C., Manga, M. and Cashman, K.V., 2003. Determining flow type, shear rate and shear stress in magmas from bubble shapes and orientations. Journal of Volcanology and Geothermal Research, 122(1), pp.111-132.

Rust, A. C., Cashman, K. V., and Wallace, P. J., 2004. Magma degassing buffered by vapor flow through brecciated conduit margins. Geology, 32(4), 349-352.

Sakuma, S., Kajiwara, T., Nakada, S., Uto, K. and Shimizu, H., 2008. Drilling and logging results of USDP-4—Penetration into the volcanic conduit of Unzen Volcano, Japan. Journal of Volcanology and Geothermal Research,175(1), pp.1-12.

Schock, R.N., Heard, H.C. and Stephens, D.R., 1973. Stress-strain behavior of a granodiorite and two graywackes on compression to 20 kilobars. Journal of Geophysical Research, 78(26), pp.5922-5941.

Sepúlveda, F., Lahsen, A., Bonvalot, S., Cembrano, J., Alvarado, A. and Letelier, P., 2005. Morpho-structural evolution of the Cordón Caulle geothermal region, Southern Volcanic Zone, Chile: Insights from gravity and 40 Ar/39 Ar dating. Journal of Volcanology and geothermal Research, 148(1), pp.165-189.

Setiawan, A., 2002. Modeling of Gravity Changes on Merapi Volcano: Observed Between 1997-2000. Ph.D. thesis, Darmstadt University of Technology, Darmstadt, Germany.

Shimada, M., Ito, K. and Cho, A., 1989. Ductile behavior of a fine-grained porous basalt at room temperature and pressures to 3 GPa. Physics of the Earth and Planetary Interiors, 55(3-4), pp.361-373.

Shteynberg, G.S. and Solov'yev, T., 1976. The shape of volcanoes and the position of subordinate vents. Izvestia Earth Phys, 5, pp.83-84.

Sigmundsson, F., Pinel, V., Lund, B., Albino, F., Pagli, C., Geirsson, H. and Sturkell, E., 2010. Climate effects on volcanism: influence on magmatic systems of loading and unloading from ice mass variations, with examples from Iceland. Philosophical Transactions of the Royal Society of London A: Mathematical, Physical and Engineering Sciences, 368(1919), pp.2519-2534.

Siratovich, P.A., Heap, M.J., Villeneuve, M.C., Cole, J.W., Kennedy, B.M., Davidson, J. and Reuschlé, T., 2016. Mechanical behaviour of the Rotokawa Andesites (New Zealand): Insight into permeability evolution and stress-induced behaviour in an actively utilised geothermal reservoir. Geothermics, 64, pp.163-179.

Sparks, R.S.J., 1978. The dynamics of bubble formation and growth in magmas: a review and analysis. Journal of Volcanology and Geothermal Research, 3(1-2), pp.1-37.

Sparks, R.S.J., 1997. Causes and consequences of pressurisation in lava dome eruptions. Earth and Planetary Science Letters, 150(3), pp.177-189.

Tiede, C., Camacho, A.G., Gerstenecker, C., Fernández, J. and Suyanto, I., 2005. Modeling the density at Merapi volcano area, Indonesia, via the inverse gravimetric problem. Geochemistry, Geophysics, Geosystems, 6(9).

Ulusay, R. and Hudson, J.A., 2007. The Complete ISRM Suggested Methods for Rock Characterization, Testing and Monitoring: 1974–2006. International Society for Rock Mechanics. ISBN 978-975-93675-4-1.

Vajdova, V., Baud, P. and Wong, T.-f., 2004. Permeability evolution during localized deformation in Bentheim sandstone. Journal of Geophysical Research: Solid Earth, 109(B10).

van Wyk de Vries, B., Kerle, N. and Petley, D., 2000. Sector collapse forming at Casita volcano, Nicaragua. Geology, 28(2), pp.167-170.

van Wyk de Vries, B.V.W. and Borgia, A., 1996. The role of basement in volcano deformation. Geological Society, London, Special Publications, 110(1), pp.95-110.

van Wyk de Vries, B.V.W. and Matela, R., 1998. Styles of volcano-induced deformation: numerical models of substratum flexure, spreading and extrusion. Journal of Volcanology and Geothermal Research, 81(1), pp.1-18.

Vinciguerra, S., Trovato, C., Meredith, P.G. and Benson, P.M., 2005. Relating seismic velocities, thermal cracking and permeability in Mt. Etna and Iceland basalts. International Journal of Rock Mechanics and Mining Sciences, 42(7), pp.900-910.

Violay, M., Gibert, B., Mainprice, D., Evans, B., Dautria, J.M., Azais, P. and Pezard, P., 2012. An experimental study of the brittle-ductile transition of basalt at oceanic crust pressure and temperature conditions. Journal of Geophysical Research: Solid Earth, 117(B3).

Violay, M., Gibert, B., Mainprice, D. and Burg, J.P., 2015. Brittle versus ductile deformation as the main control of the deep fluid circulation in oceanic crust. Geophysical Research Letters, 42(8), pp.2767-2773.

Voight, B., Hoblitt, R.P., Clarke, A.B., Lockhart, A.B., Miller, A., Lynch, L. and McMahon, J., 1998. Remarkable cyclic ground deformation monitored in real-time on Montserrat, and its use in eruption forecasting. Geophysical Research Letters, 25(18), pp.3405-3408.

Wadge, G., Mattioli, G.S. and Herd, R.A., 2006. Ground deformation at Soufrière Hills Volcano, Montserrat during 1998–2000 measured by radar interferometry and GPS. Journal of Volcanology and Geothermal Research,152(1), pp.157-173.

Wadsworth, F.B., Vasseur, J., Scheu, B., Kendrick, J.E., Lavallée, Y. and Dingwell, D.B., 2016. Universal scaling of fluid permeability during volcanic welding and sediment diagenesis. Geology, 44(3), pp.219-222.

Wong, T.-f., and Baud, P., 2012. The brittle-ductile transition in porous rock: A review. Journal of Structural Geology, 44, pp.25-53.

Wong, T.-f., David, C. and Zhu, W., 1997. The transition from brittle faulting to cataclastic flow in porous sandstones: Mechanical deformation. Journal of Geophysical Research: Solid Earth, 102(B2), pp.3009-3025

Woods, A. W., and Koyaguchi, T., 1994. Transitions between explosive and effusive eruptions of silicic magmas. Nature, 370(6491), 641-644.

Xiaochun, L., Manabu, T., Zhishen, W., Hitoshi, K. and Takashi, O., 2003. Faulting-induced permeability change in Shirahama sandstone and implication for $CO_2$ aquifer storage. Chinese Journal of Rock Mechanics and Engineering, 22(6), pp.995-1001.

[revised manuscript text omitted]
. To date, physical property data at depth in volcanic materials has been obtained predominantly by researchers assessing the suitability of volcanic deposits for hydrocarbon or geothermal energy exploitation. For example Chen et al. (2016, 2017a) investigated volcanic sequences in the Junggar Basin in western China, in order to determine their suitability as gas and petroleum reservoirs. At both sites, the authors note a general decrease in porosity with depth; for example, Chen et al. (2016) report a decrease of porosity from ∼0.30 to <0.10 between the surface and 1000 m depth. Often, the interpretation of logging data from volcanic materials (e.g. Millett et al., 2016) is non-trivial due to the variable quality of density logs, and variations in the relationship between density and porosity with depth, alteration, or the intersection of distinct facies (Li et al., 2009), as well as gaps in the stratigraphic record due to incomplete core recovery. An example of a study with an excellent degree of core recovery—99.7 %—is reported by Jónsson and Stefánsson (1982): these authors calibrate porosity and density data obtained by the Iceland Research Drilling Project from a borehole cored continuously to a depth of 1919 m near Reyðarfjörður in Iceland. To combat small-scale variability arising from the intersection of discrete geological units, Jónsson and Stefánsson (1982) calculate a running average of porosity and density against depth. Notably, the average porosity decreases from 0.13 at 400 m depth to 0.06 at 1200 m depth, corresponding to an increase in bulk rock density of approximately 200 kg m$^{-3}$.

The observation that porosity tends to decrease with increasing depth is borne out by experimental deformation studies, which show that the propensity for compactant – rather than dilatant – behaviour of volcanic rock is intrinsically linked to the confining pressure under which the sample is deformed (e.g. Shimada et al., 1989; Heap et al., 2015a; Zhu et al., 2016), as well as being heavily reliant on its initial porosity (Heap et al., 2015a; Zhu et al., 2016) and other factors such as temperature (Violay et al., 2012, 2015; Heap et al., 2017) and alteration (Siratovich et al., 2016). In detail, high effective pressures and/or high initial porosity promote ductile behaviour, whereas dilatant brittle failure is favoured in low-porosity volcanic rock deformed under a range of effective pressures. For an in-depth study regarding the influence of effective pressure and porosity on the failure mode of andesite, the reader is referred to Heap et al. (2015a).

While pre-failure permeability has been explored in plutonic (Zoback and Byerlee, 1975; Kiyama et al., 1996; Mitchell and Faulkner, 2008) and volcanic (Faoro et al., 2013) rocks, studies of post-failure permeability change have been generally limited to investigations into sedimentary and synthetic materials (e.g. Mordecai and Morris, 1970; Peach and Spiers, 1996; Zhu and Wong, 1997; Regnet et al., 2015). However, a recent study explored brittle failure in compression of low- to intermediate-porosity volcanic rock and the influence of progressive stress-induced dilation (Farquharson et al., 2016b). Permeability was found to increase with

ongoing strain accumulation under triaxial conditions. With regards to the influence of inelastic compaction, research has yielded both decreases (Zhu and Wong, 1997; Baud et al., 2012; Chen et al., 2017b) and increases (Xiaochun et al., 2003) in the permeability of porous sandstone. Alam et al. (2014) investigated the permeability evolution of welded tuff from Shikotsu (Hokkaidō Prefecture, Japan), finding that permeability decreased monotonically with triaxial compression (both in the dilation and compaction regimes) and that the rate of permeability decrease was tied to the effective pressure under which deformation was performed. Pilot experiments on porous andesite – described in Heap et al. (2015a) – also indicate permeability loss as a result of inelastic compaction. Building on the work of these studies, this contribution investigates the response of the physical properties of volcanic rock – i.e. porosity and permeability – as a function of inelastic compaction under conditions anticipated in volcanic environments. Using a simplified geometry, we use a layered flow model to discuss permeability reduction as a function of compaction localisation. We then expound these results in light of the potential influence of edifice rock compaction on volcanic activity.

**2 Materials and methods**

**2.1 Sample preparation and deformation**

To assess the influence of inelastic compaction on volcanic rock permeability, a porous andesite from Volcán de Colima (Mexico) was used. The construction history, geomorphology and eruptive style of Volcán de Colima make it a useful analogue for other active andesitic stratovolcanoes around the world, such as Gunung Merapi (Indonesia), Ruapehu (New Zealand), Volcán Rincon de la Vieja (Costa Rica), Santa María (Guatemala), Tungurahua (Ecuador) and many, many more. Core samples were prepared from a block of andesite approximately 1 m$^3$, collected in May 2014 from the La Lumbre debris-flow track (barranca) on the south-western flank of the volcano. The andesite – "LLB" – is a vesicular porphyritic andesite containing subhedral phenocrysts and microphenocrysts, of unknown age. Bulk geochemical analysis is given in Table 1.

This andesite was chosen because its relatively high initial connected porosity $\phi$ means that it can be deformed in the ductile regime under pressure conditions relevant to a volcanic edifice (Heap et al., 2015a). Ten sample cores were prepared with a diameter of 20 mm, and were ground flat and parallel to a nominal length of 40 mm. Samples were dried in a vacuum oven for at least 48 hours, and the following steps carried out (adopting the protocol of Farquharson et al., 2016b):

1. Physical properties (porosity, permeability) were measured,

2. Samples were saturated, then deformed triaxially in compression under a set effective pressure to a given degree of axial strain,

3. Samples were unloaded, dried for 48 hours, and their permeability was re-measured.

Each of these stages are described in more detail hereafter. Helium pycnometry was used to measure the bulk and powder densities of LLB samples ($\rho_b$ and $\rho_p$, respectively), whilst measurements of sample dimensions allow the calculation of the bulk volume $V$, and in turn the volumetric mass density $\rho_v$. In turn, porosity (connected $\phi$, total $\phi_t$, and unconnected $\phi_u$) can be calculated:

$$\phi = \left( \frac{\rho_b - \rho_v}{\rho_v} \right) \tag{1a}$$

$$\phi_t = 1 - \left( \frac{\rho_b}{\rho_p} \right) \tag{1b}$$

$$\phi_u = \phi_t - \phi \tag{1c}$$

where a value of $2653 \pm 0.17$ kg m$^{-3}$ is used for $\rho_p$. In detail, helium pycnometry is used to determine the solid volume $V_s$ of each sample, which subsequently allows the calculation of the connected gas porosity $\phi$ as described. Automated measurements of $V_s$ were performed iteratively until five consecutive measurements yielded results within a range of 0.01% of the sample volume, so precision of the pycnometer measurements is high ($< 0.005$ cm$^3$). A greater degree of error arises when manually measuring the sample dimensions, required in order to determine $\rho_v$. Repeat measurements allowed an estimation of error in the length and diameter, which typically amount to $< 0.05$ cm$^3$ in terms of volume. Adopting the notation that $\epsilon_\phi$ is the error on the porosity calculation and that $\epsilon_x$ and $\epsilon_y$ are the independently calculated errors for measurements of $V_s$ and $V$, then the propagated error can be approximated by:

$$\epsilon_\phi = \phi \times \left[ \left( \frac{\epsilon_z}{V - V_s} \right)^2 + \left( \frac{\epsilon_y}{V} \right)^2 \right]^{\frac{1}{2}} ; \ \epsilon_z = \left( \epsilon_x^2 + \epsilon_y^2 \right)^{\frac{1}{2}} \tag{2}$$

Values for $\epsilon_\phi$ are generally $< 0.005$ for the samples described herein. As such, probable error on connected gas porosity measurements is low, and always contained within the symbol size when plotted graphically.

Gas permeability was measured under steady-state conditions with a confining pressure of 1 MPa using the setup described in Farquharson et al. (2016c). Where necessary, a correction was applied to the measured permeability values to account for turbulent flow (see Forchheimer, 1901), the effects of which can become non-negligible when measuring the permeability of high-porosity media. Appendix A contains further details on the determination of permeability, the application of corrections for inertial effects, and the sources and sizes of potential error in the measurements.

**Table 1.** Major element (oxide) composition, determined via X-ray fluorescence analysis. All values are given in weight percent (wt. %).

| $SiO_2$ | $Al_2O_3$ | $Fe_2O_3$ | MnO | MgO | CaO | $Na_2O$ |
|---|---|---|---|---|---|---|
| 61.260 | 17.330 | 5.745 | 0.100 | 3.725 | 5.505 | 4.455 |
| $\pm$ 0.270 | $\pm$ 0.070 | $\pm$ 0.085 | $\pm$ 0.001 | $\pm$ 0.295 | $\pm$ 0.025 | $\pm$ 0.015 |

| $K_2O$ | $TiO_2$ | $P_2O5$ | $Cr_2O_3$ | $V_2O_5$ | NiO | LOI |
|---|---|---|---|---|---|---|
| 1.505 | 0.565 | 0.200 | 0.020 | 0.015 | 0.004 | 0.185 |
| $\pm$ 0.035 | $\pm$ 0.015 | $\pm$ 0.020 | $\pm$ 0.005 | $\pm$ 0.005 | $\pm$ 0.001 | $\pm$ 0.215 |

$SiO_2$ = Silicon dioxide; $Al_2O_3$ = Aluminium oxide; $Fe_2O_3$ = Iron oxide; MnO = Manganese(II) oxide;
MgO = Magnesium oxide; CaO =Calcium oxide; $Na_2O$ = Sodium oxide; $K_2O$ = Potassium oxide;
$TiO_2$ = Titanium oxide; $P_2O_5$ = Phosphorus pentoxide; $Cr_2O_3$ = Chromium(III) oxide; $V_2O_5$ =
Vanadium(V) oxide; NiO = Nickel(II) oxide; LOI = loss on ignition.

Samples were then encased in a copper foil jacket (which serves to retain bulk sample cohesion after deformation), saturated with distilled water, and loaded into the triaxial deformation rig at Université de Strasbourg (see Figure 1). Throughout deformation we assume a simple effective stress law, whereby the effective confining pressure $p_{eff}$ experienced by a sample is a function of the confining pressure $p_c$ around the sample and the pressure of pore fluid $p_p$ within the sample, such that $p_{eff} = p_c - \alpha \cdot p_p$. Recent experimental work (Farquharson et al., 2016a) shows that $\alpha = 1$ is a reasonable assumption for porous andesite.

For each test, the confining and pore pressures were increased slowly until a targeted effective pressure (i.e. hydrostatic pressurisation). Note that pressure is only controlled within the connected porous network (i.e. $\phi$). However, we assume that any influence of incomplete sample saturation is negligible, due to the relatively small volume of isolated porosity $\phi_u$ in these andesites (see Table 2). Assuming a pycnometry-derived value for bulk density $\rho_b$ of approximately 2100 kg m$^{-3}$, the imposed effective pressures of 10, 30, 50, and 70 MPa are analogous to depths ranging from the upper 500 m of the edifice to greater than 3 km in depth (given that $p_c \propto \rho_b \cdot g\hat{z}$, where $g$ and $\hat{z}$ are surface gravitational acceleration and depth, respectively). The sample would then be left overnight to allow microstructural equilibrium. During the deformation experiments, a differential stress was introduced in the direction of the sample axis by advancing an axial piston (see Figure 1) under servo-control, such that the sample is subjected to a constant strain rate of 10$^{-5}$ s$^{-1}$ (note that hereafter "strain" refers to axial strain unless otherwise specified). We note that – on the edifice-scale – absolute strain rates resulting from magma migration and edifice displacement are generally of the order of 10$^{-7}$ s$^{-1}$ or lower, for example as estimated from borehole strain-meters: Linde et al. (1993), or from spaceborne interferometry: Massonnet et al. (1995). However, strain and strain rates are undoubtedly highly variable throughout active volcanic systems. The chosen strain rate for these experiments (the international standard in rock mechanics Kovari et al., 1983; Ulusay and Hudson, 2007) is comparable to shear rates inferred to occur along conduit margins by Rust et al. (2003). Similarly, Cashman et al. (2008) estimate strain rates of 3 - 8 $\times$ 10$^{-5}$ s$^{-1}$ for the formation of fault gouge at Mount St Helens (USA). Most importantly, a strain rate of 10$^{-5}$ s$^{-1}$ ensures that our samples are drained (i.e. the product of the Darcy timescale and the strain rate is $\ll$ 1: Heap and Wadsworth, 2016).

Confining pressure and pore pressure were servo-controlled throughout the experiments. During hydrostatic and nonhydrostatic loading, the response of the pore fluid pump reflects variations in pore volume (see Read et al., 1995), which – normalised to the initial sample volume – corresponds to the porosity change $\Delta\phi$. For a porous material, $\Delta\phi$ can be considered equal to the volumetric strain. Indeed, the response of the confining pressure pump provides an independent estimation of the volumetric strain, found to be in perfect agreement with the inferred $\Delta\phi$ (see Baud et al., 2014, for details). When this differential $\Delta\phi$ is positive it signifies dilation (an increase in porosity) and when it is negative, it indicates compaction (a decrease in porosity). Deformation was allowed to continue for different amounts of axial strain accumulation – sample shortening relative to its original length ($\varepsilon_t$) – then unloaded. The strain recovered during the unloading phase is subtracted from the total axial strain $\varepsilon_t$ to give the inelastic (non-recoverable) strain accrued by the sample ($\varepsilon_i$). Similarly, the elastic porosity change recovered during unloading is subtracted from the porosity change at $\varepsilon_t$, to give the inelastic porosity change $\Delta\phi_i$. Samples were vacuum-dried once again and gas permeability re-measured. Samples were all deformed at room temperature. We note that in natural volcanic environments there may be some influence of temperature on rock strength (due, for instance, to the closure of cracks driven by thermal expansion). Nevertheless, a recent study by Heap et al. (2017) shows that the influence of temperature on the physical and mechanical properties of andesite may not be significant at temperatures below the glass transition $T_g$. Importantly, this study showed that the failure mode and underlying microstructural mechanism driving ductile behaviour (cataclastic pore collapse) did not change below $T_g$, which is itself largely restricted to the

[Figure]

**Figure 1.** Schematic of triaxial deformation apparatus, including confining pressure $p_c$, pore pressure $p_p$, and axial pressure $p_{ax}$ circuits. Detail of sample assemblage is shown inset. (a) axial piston; (b) "blank" endcap; (c) sample; (d) copper foil jacket; (e) nitrile jacket; (f) pore fluid distributor endcap. Directions of major $\sigma_1$ and minor $\sigma_3$ principal stresses are as shown, such that $\sigma_1 > \sigma_2 = \sigma_3$. Not to scale. Numbered valves allow various parts of each circuit to be used at any given time.

magma conduit and rock in the immediate vicinity. This is in agreement with previous studies by Vinciguerra et al. (2005) and Heap et al. (2014), both of which noted only negligible changes in microcrack density and porosity after thermally stressing volcanic materials. These authors attribute this phenomenon to the high initial crack density resulting from the complex thermal histories of volcanic rocks.

**2.2 Post-deformation permeability**

It has been shown in recent studies (e.g. Vinciguerra et al., 2005; Nara et al., 2011) that the permeability of fractured volcanic materials is influenced by the effective pressure under which it is measured: permeability tends to decrease with increasing effective pressure. As such, we acknowledge the limitation that post-deformation measurements do not represent the permeability under the deformation conditions *sensu stricto*. Nevertheless, we choose to measure permeability under the conditions described above for a host of reasons. Investigations towards determining the influence of effective pressure on properties (including permeability) other than rock strength indicate that their evolution with pressure may differ as a function of porosity, pore geometry, and other factors; which is to say that the effective pressure coefficient for a given rock property may not be the same as the Biot-Willis coefficient $\alpha$ (Bernabé et al., 1986). Given the lack of constraint on the effective pressure effect for the permeability of volcanic rocks, permeability is measured at the lowest possible confining pressure (1 MPa, rather than at "*in situ*" pressures) and without imposing a differential stress, in order to allow comparison within and between sample sets (indeed, we compare our data with compiled literature data in Sect. 4.1). This procedure also avoids the potential for creep – a mechanism of time-dependent deformation whereby subcritical crack growth induces damage and possibly even failure at stresses below the short-term strength of the rock (e.g. Heap et al., 2011; Brantut et al., 2013) – as well as precluding other phenomena such as stress relaxation that may arise when measuring permeability under a differential stress.

Measuring permeability requires that the sample dimensions, specifically length and cross-sectional area, are well constrained. Prior to initial measurements of permeability, sample dimensions are measured accurately using digital callipers. However, samples are often barrelled and thus noncylindrical after mechanical deformation, making their mean radii nontrivial to determine. Assuming that the solid volume $V_s$ remains constant throughout deformation, then the post-deformation volume is equal to the sum of solid volume, the initial pore volume, and the pore volume change after deformation. The post-deformation cross-sectional area $A_{post}$ can therefore be determined such that

$$A_{post} = \underbrace{\left[\frac{V_s}{1-(\phi+\Delta\phi_i)}\right]}_{V_{post}} \times \left[\frac{1}{l_{post}}\right] \quad (3)$$

where $V_{post}$ is the post-deformation volume and $l_{post}$ is the mean sample length after deformation.

**3 Results**

[revised manuscript text omitted]

$\phi$ = connected porosity; $\phi_u$ = unconnected porosity; $p_{eff}$ = effective pressure; $\varepsilon_t$ = target (total) axial strain; $\varepsilon_i$ = inelastic axial strain; $\Delta\phi_i$ = inelastic porosity change; $\phi + \Delta\phi_i$ = post-deformation porosity; $k_0$ = initial permeability; $k_e$ = post-deformation permeability.

**4   Discussion**

**4.1   Microstructural controls on permeability evolution**

The underlying micromechanical mechanism driving inelastic compaction in volcanic rocks has been shown to be cataclastic pore collapse (Zhu et al., 2011; Heap et al., 2015a; Zhu et al., 2016). Figure 4 illustrates this process by showing images of an intact and a deformed sample. Figure 4a is a backscattered scanning electron microscope image of an as-collected sample of LLB andesite, whereas the images in Figure 4b and 4c (from the same sample suite) are of a samples that has accumulated high strain (> 0.20) under an effective pressure of 30 MPa. The undeformed sample (Figure 4a) is pervasively microcracked, with highly amœboid pores ranging from < 10 μm to around 80 μm in diameter. Cataclastic pore collapse involves intense microcracking, which develops in a concentric damage zone around a pore. As the process of cataclasis – progressive fracturing and comminution – continues, fragments can spall into the void space, thus reducing porosity (Zhu et al., 2010). Figure 4b clearly shows abundant fractures created during triaxial deformation, both within the groundmass and crystals. In many areas, fragments have been comminuted to the micron-scale. In these samples, as observed in previous experimental studies of volcanic rock (Loaiza et al., 2012; Adelinet et al., 2013; Heap et al., 2015a), cataclastic pore collapse is localised in the form of bands traversing the sample.

The occurrence of compaction bands in andesite has been shown to correspond to periodic stress drops (Heap et al., 2015a), which are abundant in the mechanical data of Figure 2a. Our experimental data (Figure 2b and Figure 3a) show that cataclastic pore collapse progressively reduces the porosity of these andesites. If we assume that compaction is perfectly localised in our samples, we can consider a compaction band-bearing sample as a layered medium where the band of porosity $\phi_b$ is embedded within an "intact" host with the initial rock properties (which is to say, a sample of porosity $\phi$). The intact material must have a pore volume $V_1^\phi$ of $\phi \times (l_{intact} \times A_{post})$, where $l_{intact}$ is the overall length of the sample that is undamaged ($l_{post} - w_b$), where $w_b$ is the width of the compaction band. The deformed sample contains a pore volume $V_2^\phi$ of $\phi + \Delta\phi_i \times (l_{post} \times A_{post})$. From this, we can relate the compaction band porosity $\phi_b$ to the width of the compaction band:

$$\phi_b = \left[\frac{V_2^\phi - V_1^\phi}{A_{post}}\right] \times \left[\frac{1}{w_b}\right]. \tag{4}$$

Solutions for $\phi_b$ and $w_b$ are non-unique (moreover, a greater value of $w_b$ could be a function of one wide band or a number of discrete, relatively thinner bands), but we can impose a lower bound on $\phi_b$ of zero, and an upper bound equal to the post-deformation porosity of the sample: $0 \leq \phi_b < (\phi + \Delta\phi_i)$. Assuming the compaction band porosity noted by Heap et al. (2015a) (~0.10) is typical for compacted andesite, then Equation 4 yields compaction band widths of between 1.63 and 23.57 mm (i.e. between 4 and 70 % of the overall sample length). We note that the lower end of this range is in line with the observations of Heap et al. (2015a).

We note that porosity loss is seemingly tied to the effective pressure under which compaction occurs (as observed in previous studies concerned with triaxial rock deformation, e.g. Wong et al., 1997; Baud et al., 2006; Heap et al., 2015a): for a given increment of inelastic strain, the porosity lost by a sample is greater at a higher effective pressure. This phenomenon is true both for total porosity change (Figure 2b) and for inelastic porosity loss (Figure 3a), which is to say that the inelastic compaction factor $\Delta\phi_i/\varepsilon_i$ always decreases as effective pressure increases (Baud et al., 2006).

While the mechanism of cataclastic pore collapse is governed by the pore sizes (e.g. Zhu et al., 2010, 2011), it has also been demonstrated that the local stress field around a

[Figure]

[Figure]

[Figure]

**Figure 4.** Backscattered scanning electron microscope images of LLB andesite, showing as-collected (a) and post-deformation (b - c) microstructure. Void space appears as black. Dense (metal-rich) phenocrysts appear as white or light grey within a darker grey groundmass. Both (b) and (c) are images of LLB-13, which was taken to beyond both $C^*$ and $C^{*\prime}$. Cataclastic pore collapse associated with shear-enhanced compaction is shown in (b), while (c) shows part of the dilatant shear zone marking the transition from shear-enhanced compaction to dilation.

pore increases as a function of the incumbent confining pressure (Zhu et al., 2010). A study of fault gouge formation in sandstone (Engelder, 1974) shows that fault-zone fragments are smaller when generated at higher confining pressures, and a similar effect (albeit less pronounced) was noted by Kennedy et al. (2012), who investigated fault gouge formation in dacitic dome rock. It is reasonable to assume that cataclasis may become more efficient as the local stress field increases in line with the confining pressure; in turn, a finer distribution of fragments will more readily occlude the pores around which they develop. Whether a change in the mean fragment size generated during cataclastic pore collapse underlies the observed evolution of $\Delta\phi_i/\varepsilon_i$ remains open to a targeted microstructural study. Nevertheless, we can interrogate Equation 4 to glean an idea of the effect of $\phi_b$ and $w_b$ at different effective pressures and axial strains.

Figure 5 shows the calculated $w_b$ for our experimental data as a function of inelastic axial strain. Values of $w_b$ are calculated using Equation 4, using values of $\phi_b$ of 0.20, 0.15, 0.10, and 0.05. At relatively higher imposed values of $\phi_b$ (0.20 and 0.15: Figure 5a, 5b), many of the resulting values of $w_b/l$ are non-physical (i.e. $w_b/l \not> 0$ or $w_b/l \not< 1$). However, at lower imposed values of $\phi_b$ (0.10 and 0.05: Figure 5c, 5d), values fall between 0 and 1. Moreover, there appears to be a systematic effect of $p_{eff}$, with deformation under relatively higher effective pressure yielding a higher ratio of $w_b/l$ – hence, a thicker compaction band – for any given amount of inelastic axial strain accumulation.

As would be expected (e.g. Zhu and Wong, 1997), permeability reduction follows the same general trend (Figure 3b) as porosity reduction (Figure 3a), with samples accumulating high strains showing a correspondingly large reduction in permeability. Notably, there appears to be an influence of the effective pressure under which the sample was deformed and the change in permeability for a given increment of axial strain. A difference in measured post-deformation permeability $k_e$ may be due to (1) a variation in characteristic grain size, or (2) a variation in the thickness of compaction localisation features with respect to the sample length, as described above. Moreover, one may imagine that these two factors (characteristic grain size, band thickness) operate in tandem to reduce permeability as effective pressure increases. To test this theory, we again model the deformed samples as a layered medium, such that discrete bands of uniform permeability $k_b$ and thickness $w_b$ are embedded in a medium of permeability $k_0$ (see Figure 6a). A similar approach has previously been adopted by Vajdova et al. (2004) and Baud et al. (2012) to model the permeability of sandstones containing experimentally-induced compaction bands. Fluid flow through this simplified geometry may then be modelled by assuming conservation of mass (e.g. Freeze and Cherry, 1979), such that:

$$k_e = k_0 \times \left[ \left( \frac{w_b}{l} \right) \left( \frac{k_0}{k_b} - 1 \right) + 1 \right]^{-1} \qquad (5)$$

[Figure]

**Figure 5.** Calculated sample length ratio $w_b/l$ as a function of inelastic axial strain accumulation $\varepsilon_i$. Values of $w_b$ calculated from Equation 4 using values of compaction band porosity of $\phi_b = 0.20$ (a), $\phi_b = 0.15$ (b), $\phi_b = 0.10$ (c), and $\phi_b = 0.05$ (d). Shaded area indicates range of physical values of $w_b/l$.

Variables $k_e$, $k_0$, and $l$ are already constrained, allowing us to solve for combinations of $w_b$ and $k_b$. If we assume that a compaction band comprises a granular bed (Figure 6a *inset*), we can relate its permeability $k_b$ to surface area $s$ in the following manner after Martys et al. (1994), who determined a universal scaling of permeability of a system of packed spheres:

$$k_b = \frac{2(1-\phi^*)}{s^2}\phi^{*f} \tag{6}$$

where $\phi^* = \phi_b - \phi_c$, and $f = 4.2$ (the value of $f$ is thought to be related to the initial grain geometry: Wadsworth et al., 2016). $\phi_c$ represents the percolation threshold, taken here as 0.03. Note that the characteristic porosity is taken as the porosity within a compaction band $\phi_b$ (i.e. the porosity of the granular layer with permeability $k_b$ and width $w_b$). In turn, we can relate $s$ to a characteristic grain size (e.g. Wadsworth et al., 2016):

$$s(r) = \frac{3(1-\phi_b)}{r} \tag{7}$$

where $r$ is the monodisperse particle radius. Note that in reality, the porosities assumed within the compaction bands here are not compatible with a monodisperse packing of spheres. Nevertheless, this greatly simplified approach gives an indication of the relative influence of the difference constituent parameters $r$, $\phi_b$, and $w_b$.

The assumed geometry is illustrated in Figure 6a, including the corresponding values of permeability and porosity for each layer. Figure 6b highlights the effects of changing either the characterisitic particle (i.e. grain) radius or the compaction band porosity. Notably – for a given porosity – a change in particle radius of 1 order of magnitude results in a change in compaction band permeability of 2 orders of magnitude. At relatively high initial porosities, a reduction in porosity by a given volume (for example, from 0.20

to 0.15) has little influence on $k_b$. On the other hand, when the porosity is low, a change in porosity of the same absolute volume (for example, from 0.10 to 0.05) exerts a much greater influence over $k_b$ (in this case, a reduction by over 2 orders of magnitude). However, the bulk sample permeability (the equivalent permeability) depends not only on the porosity of the compaction band but also its width. Figures 6c – 6f show the equivalent permeability for different values of $\phi_b$ for changing values of $w_b/l$: the ratio of the band width to the overall length of the sample. Curves are modelled by combining Equations 5, 6, and 7, using the particle radii $r$ (noted on each figure panel) and a value of $k_0 = 5.0 \times 10^{-13}$ m$^2$. As there are multiple non-unique solutions for $\phi_b$ and $w_b$, we show model results for a range of potential $\phi_b$ values in panels 6c to 6f.

Evidently, the variation in our experimental data (for example, the difference between $k_e$ of samples deformed under different effective pressures) is not explained by a systematic evolution of $r$. This suggests that while $k_b$ is very sensitive to the characteristic grain radius, the tradeoff between $\phi_b$ and $w_b$ is more important in controlling the bulk sample permeability ($k_e$). Importantly, however, idealising the geometry of a compaction band in terms of a monodisperse particle size distribution cannot accurately represent its complex porous network. More accurate values of $\phi_b$ and $s$ (and hence, a better prediction of $k_b$ and $k_e$) may be acheived by adopting a polydisperse particle size distribution or by imposing a non-spherical characteristic particle shape, for example.

An evident weakness of employing the simple layered medium model outlined above is that we assume that the only operative mechanism is porosity- and permeability-reducing. However at low strains, there is no one-to-one relationship between permeability and porosity after deformation. Rather, permeability tends to increase moderately at inelastic axial strains less than around 0.05 (i.e. a 5% shortening in sample length): while initial values of permeability tended to be

[Figure]

**Figure 6.** Model geometry and results. (a) Deformed sample may be thought of as a granular bed (with permeability $k_b$ and porosity $\phi_b$) within a sample matrix with permeability $k_0$ and porosity $\phi + \Delta\phi_i$. Inset highlights the characterisitc grain radius that governs surface area (Equation 7). (b) The permeability of a compaction band composed of packed spheres as a function of sphere radius and bed porosity. Note that in reality, porosities $< 0.26$ would require a polydisperse packing of spheres, or a granular bed composed of non-spherical grains. (c) – (f) Modelled equivalent permeability $k_e$ of a compaction band-bearing sample, plotted against the ratio of compaction band width relative to the overall sample length ($w_b/l$), for compaction band porosities of $\phi_b = 0.20$ (c), $\phi_b = 0.15$ (d), $\phi_b = 0.10$ (e), and $\phi_b = 0.05$ (f). Also plotted are data from this study (Table 2), where values of $w_b$ are calculated after Equation 4. Note that of the ten $k_e, w_b/l$ data pairs from this study, not all are plotted on each of the panels (c) to (f). This is because certain combinations of $\phi_b$ and $w_b$ yield non-physical values (Equation 4). Refer to text for further discussion.

around $5 \times 10^{-13}$ m$^2$, the measured post-deformation permeability was often greater than $10^{-12}$ m$^2$ after accumulating a small amount of axial strain (Table 2). A similar phenomenon was also observed by Loaiza et al. (2012), who noted that permeability of Açores trachyandesite increased beyond a critical stress state during hydrostatic pressurisation. This critical stress – known as $P^*$: Zhang et al. (1990) – signals the onset of lithostatic inelastic compaction; Loaiza et al. (2012) show that stress-induced cracks coalesce between collapsed pores during hydrostatic compaction, improving connectivity and, in turn, increasing permeability. Prior to deformation, the samples of porous andesite used in our experiments – LLB – contained an isolated porosity of 0.01, on average (Table 2).

Similar to the mechanism posited by Loaiza et al. (2012), we suggest that distributed microcracking during the initial stages of ductile deformation serves to interconnect this isolated porosity, creating efficient pathways for fluid flow. The mechanical data of all the experiments (Figure 1a) exhibit intermittent stress drops – even in the instances where permeability was observed to increase relative to the initial value – which suggests that compaction localisation in these andesites does not necessarily equate to the formation of an effective barrier to fluid flow. Loaiza et al. (2012), Adelinet et al. (2013), Heap et al. (2015a), and Heap et al. (2016) each examine microstructure of compaction bands formed in porous volcanic rocks (trachyandesite, basalt, andesite, and dacite, respectively). In all cases, the bands are irregular in shape and thickness, but do not necessarily constitute a contiguous surface of collapse pores (i.e. a layer of reduced porosity). This is supported by the results from a recent study by Baud et al. (2015), which examined compaction band-bearing sandstones using X-ray computed tomography. The authors show that when compaction bands are formed in a rock with porosity clusters, their path is more tortuous than in material with homogeneous porosity; consequently the bands do not comprise efficient permeability barriers in three dimensions. Notably, the porosity of the andesites deformed in our study exhibit marked heterogeneity in terms of porosity, pore shape, and pore size distribution (Figure 4a). The characteristic tortuosity of compaction bands formed in heterogeneous volcanic material – 
[revised manuscript text omitted]
, possibly due to enhanced communion. By modelling a simple sample geometry where a compaction band is represented by a packed granular bed, we show that the permeability reduction within a discrete compaction band is sensitive to the characteristic grain size. The effect of the porosity of a compaction band has a variable influence on its permeability, with changes in band porosity becoming ever more important as porosity decreases. However, the overall trade-off between the width and porosity of a compaction band are far more important that grain size in controlling fluid flow throughout the bulk of a sample. At low strains ($< 0.05$), compaction tends to result in a moderate increase in permeability (not accounted for in our model), which we suggest is a result of increased pore connectivity due to distributed microcracking. This effect is outweighed by progressive compaction at higher strains, resulting in a general trend of decreasing permeability with ongoing inelastic compaction. There exists a physical limit to compaction, which we suggest is echoed in a limit to the potential for permeability reduction in a deforming sample. Compiled data show that at high strain, porosity and permeability tend to converge towards intermediate values (i.e. $0.10 \leq \phi \leq 0.20$; $10^{-14} \leq k \leq 10^{-13}$ m$^2$). Field evidence from the literature emphasises the importance of understanding the physical and mechanical properties of rock in active volcanic environments, in particular the evolution in a rock's capacity to effectively transmit magmatic volatiles.

**6  Data availability**

Data are presented in Table 1 and 2 in the text, or are available on request to the corresponding author.

**Appendix A:  Measuring permeability**

The constitutive equation governing fluid transport in porous and granular media was originally derived from experiments performed by Henry Darcy in the 1850s on the flow of water at different levels through sand. Since Darcy's work (1856), the theoretical framework of fluid transport—which is based on Newton's second law—has been well established and expanded, such that flow of gas through a porous medium may be given thus:

$$Q_v = \frac{-kA}{\mu} \frac{(p_b - p_a)}{l} \tag{A1}$$

where $\mu$ is the fluid viscosity, $Q_v$ is the volumetric flow rate, $A$ is the cross-sectional area available for flow and $l$ is the distance over which fluid flow occurs (i.e. the sample length). In a fluid transport system, flow is driven towards the region of lowest potential energy: in the special case of horizontal flow, this may be described by a differential between a region of relatively high pressure $p_b$ to one of relatively lower pressure $p_a$: a pressure differential or pressure drop $\nabla p$. Equation A1 is valid for all porous media as long as flow is laminar (two cases of non-laminar flow are discussed hereafter). While this expression is sufficient for the case of laminar (or "streamline") flow, when considering an ideal compressible gas measured under atmospheric conditions, it becomes convenient to present gas permeability $k_{gas}$ in the following manner (Klinkenberg, 1941; McPhee and Arthur, 1991):

$$k_{gas} = \frac{Q_v \mu l \cdot p_{atm}}{A \cdot \nabla p \bar{p}} \tag{A2}$$

where $p_{atm}$ is the atmospheric pressure at which $Q_v$ is measured, and the driving pressure is given as a product of the differential pressure $\nabla p$ and the mean pressure over the sample $\bar{p}$. The mean pressure $\bar{p}$ is determined by the upstream and downstream pressures $p_b$ and $p_a$ such that $\bar{p} = (p_b + p_a)/2$. Under ambient conditions, $p_a$ is equal to the atmospheric pressure $p_{atm}$, and $p_b$ is equal to $\nabla p + p_{atm}$. The mean pressure therefore simplifies to $\bar{p} = (\nabla p + 2p_{atm})/2$.

Herein, gas permeability was measured using a steady-state permeameter using the setup described in Farquharson et al. (2016c). The apparatus is a commercial benchtop permeameter from Vinci Technologies, modified by incorporating interchangeable El-Flow volumetric mass flowmeters (from Bronkhorst) to measure the volumetric flow rate of gas at the downstream end of the experimental samples. Gas permeability was measured using nitrogen as the permeant (pore fluid). A confining pressure of 1 MPa was applied radially to the sample, in order to ensure that no leakage occurred along its margins during measurement. The sample was then left under this confining pressure for 1 hour, to allow for any necessary microstructural equilibrium. Gas would then be flowed through the sample, whilst the volumetric flow rate $Q_v$ and the pressure differential $\nabla p$ across the sample were continuously monitored by means of customised data acquisition system and a LabVIEW program written for this purpose.

The pressure of gas entering the sample could be adjusted using a regulator attached to the permeant gas bottle. By altering the flow of gas, a range of different values of $\nabla p$ was imposed across the samples (typically between 0.001 and 0.2 MPa). Once steady-state flow was achieved, the volumetric flow rate was noted. Thus with knowledge of the gas viscosity and sample dimensions, permeability could be calculated using Equation A2. However, two scenarios can necessitate that post-measurement corrections need to be applied to the calculated values due to inertial effects: flow turbulence or gas slippage.

**A1  Non-laminar flow 1: Turbulence**

Forchheimer (1901) conducted fluid flow experiments through porous media, noting that the relationship between

the pressure differential $\nabla p$ and the volumetric flow rate $Q_v$ becomes nonlinear at high fluid velocities due to flow no longer being laminar. To account for this turbulence, an inertial term, here denoted $\iota$, must be introduced, such that:

$$\frac{1}{k_{fo}} = \frac{1}{k_{gas}} - \iota \cdot Q_v \tag{A3}$$

where $k_{fo}$ is the Forchheimer-corrected permeability value, and $k_{gas}$ is the as-measured value. In this scenario, the measured gas permeability would be lower than the true (corrected) permeability, as turbulence induces resistance to fluid flow.

**A2   Non-laminar flow 2: Gas slippage**

In his seminal 1941 paper, Klinkenberg showed that as the characteristic pore size or aperture approaches the mean free path of the permeant gas—the distance travelled between consecutive molecular collisions—interactions between the gas molecules and the pore (or crack) walls serve to reduce resistance to flow. Simply put, during liquid laminar flow, the layer of molecules adjacent to the pore (or crack) walls is static. However, for gases this molecule layer has a nonzero velocity due to molecular diffusion ("slip"). This slippage results in a higher flow rate at any given pressure differential for a gas than a liquid. Accordingly, the permeability measured using a gas would be artificially higher than if determined using a liquid.

The relationship of Klinkenberg (1941) is incorporated thus:

$$k_{gas} = k_{kl}\left(1 + \frac{b}{\bar{p}}\right) \tag{A4}$$

where $k_{gas}$ is the as-measured permeability calculated from gas flow experiments (note that in cases where a Forchheimer correction has been applied as in Equation A3, $k_{gas}$ is substituted by $k_{fo}$ in Equation A4), $\bar{p}$ is the mean flow pressure of gas in the system, $b$ is the Klinkenberg parameter (which depends on both the gas used but also the pore structure), and $k_{kl}$ is the Klinkenberg-corrected permeability value.

In the absence of inertial effects, plotting $Q_v$ against the driving pressure (i.e. $\nabla p \bar{p}$) yields a linear relationship. Deviations from linear behaviour indicate that one or both of the inertial phenomena described above are influencing the calculated permeability. In practice, the corrected permeability can be calculated using the slope and intercept of graphs of $Q_v$ against $k_{gas}^{-1}$, and $\bar{p}^{-1}$ against $k_{fo}$. In the main body of the text measured permeability, corrected for turbulence and/or gas slippage when necessary, is always presented as $k$. Figure A1 shows example data from volcanic rocks which exhibit laminar flow, turbulence, and gas slippage, respectively. The effects of inertial flow, especially in the case of gas slippage tends to be slight, although non-negligible.

**A3   Permeability and experimental error**

Sources of error in the permeability measurements include the sample dimensions, and the resolution of the pressure transducer and flowmeters. As mentioned, permeability is determined as a function of the $Q_v - \nabla p \bar{p}$ curve, which is a series of points fit by a simple linear regression (when flow is laminar). The respective precision of the transducer and flowmeter is thus encompassed by the coefficient of determination of the regression line (i.e. its $r^2$ value). If the data are unaffected by turbulence or gas slippage, then $r^2$ is generally greater than 0.99. If flow is nonlaminar, $r^2$ tends to be appreciably lower, and the permeability is determined using Equation A3 or  A4 as appropriate. Repeat measurements suggest that experimental error is always engirdled by the symbol size when plotted graphically.

Figure A1 shows flowrate and pressure data obtained during steady-state permeability measurements on three volcanic samples. For the first example, Figure A1a - c, flow is laminar, as evident from the linear relation between the volumetric flowrate ($Q_v$) and the driving pressure ($\nabla p \bar{p}$). Accordingly, the reciprocal permeability ($k_{gas}^{-1}$) versus $Q_v$ is negative and nonlinear, as is the measured permeability $k_{gas}$ against the reciprocal mean pressure $\bar{p}^{-1}$. In the second example, flow is turbulent, and the data in Figure A1d are nonlinear. Applying the correction derived from Figure A1e (Equation A3) yields Figure A1f, where the data are randomly distributed about the mean (no Klinkenberg correction is necessary). Finally, the data shown in Figure A1g - h highlight that a Klinkenberg correction is necessary (Equation A4). However, the correction is very slight, as indeed often tends to be the case in volcanic rocks (as opposed to "tight" materials such as granite). For the LLB data presented in Table 2, Forchheimer corrections were applied to the raw values where appropriate. Permeability values were affected only slightly, increasing by a factor of between 1.03 to 1.41.

*Author contributions.* J.F., P.B., and M.H. designed the experiments, which were performed by J.F. J.F. prepared the manuscript with contributions from all authors.

*Competing interests.* The authors declare that they have no conflict of interest.

*Acknowledgements.* Thierry Reuschlé, Alex Schubnel, Luke Griffiths, and Alexandra Kushnir are thanked for inspiring discussions. Fieldwork was funded in part by the framework of the LABEX  ANR-11-LABX-0050_G-EAU-THERMIE-PROFONDE and therefore benefits from a funding from the state managed by the French National Research Agency as part of the Investments for the future program. JIF acknowledges an Initiative d'Excellence (IDEX) "Contrats doctoraux" grant from the French State. MJH acknowledges Initiative d'Excellence (IDEX) Attractivité grant

[Figure]

**Figure A1.** Data obtained during permeability measurements on three rocks. a) - c) Laminar flow in a sample from Whakaari, New Zealand. d) - f) Turbulent flow, in a sample of Ruapehu andesite (New Zealand). g) - i) A small gas slippage effect in a sample of Açores trachyandesite.

"VOLPERM" and a CNRS INSU grant. We are grateful to Nick Varley and Oliver Lamb for field assistance at Volcán de Colima. Fabian Wadsworth and Antonella Longo are thanked for their constructive comments on the manuscript.

---

## Author Comment (AC2) · 28 Mar 2017

**Response to reviewer:**

**"Inelastic compaction and permeability evolution in volcanic rock"**

Please find attached the revised version of our manuscript *Inelastic compaction and permeability evolution in volcanic rock*. We are pleased with the constructive suggestions from the reviewer which, although minor, have helped to further improve the paper. Below we outline the changes made to the manuscript (in **green**). Our responses to the reviewer comments are given in **bold** below.

The revised manuscript and supplementary files are attached after the comment responses.

Yours sincerely,

Jamie Farquharson, on behalf of P. Baud, and M. J. Heap.

**Response to reviewer:**

**"Inelastic compaction and permeability evolution in volcanic rock"**

*Reviewer comment* **on "Inelastic compaction and permeability evolution in volcanic rock"** *by* **Jamie I. Farquharson et al.**

This manuscript deals with a central question in Earth science related to fluid migration in our volcanically and tectonically active lithosphere. Namely, the evolution of fluid permeability during deformation. The hypothesis that is here tested experimentally is that the evolution of the fluid permeability in volcanic rocks is highly dependent on the inelastic strain accumulated when the deformation occurs in the compaction regime. There is little or no analysis of the data presented beyond computing the inelastic component of the bulk (or axial?) strain and presented measured values of properties that evolve with strain. The is the only shortcoming of this work. Otherwise, the manuscript is well illustrated, well written and meets the standards of novelty and quality for publication.

Below I submit some constructive comments which I hope will guide the authors in their revision of this article.

**We thank the reviewer for their assessment of the article, which we found very useful. Amendments and additions to the text are outlined below.**

(1) In the abstract line 19, the term "high strains" is used. I think that throughout the manuscript the authors should be explicit and clear about what strain they are talking. Axial strain, pore strain, and inelastic strain are all introduced at some point and so referring to strain more generally is not clear.

**We agree with the reviewer that this is perhaps not always clear. We have changed "strain" to "axial strain" where suggested in the abstract. In the main body of the text, we now state: "(note that hereafter "strain" refers to axial strain unless otherwise specified)".**

(2) Page 2 line 10; do the authors mean to say "siphoned"? Or do they simply mean that the volatiles are able to escape? Throughout the manuscript there is some flamboyant language that perhaps precludes a clear understanding of what is meant in places. Another example is perhaps the use of "amoeboid" as an adjective to describe a pore shape on line 26 of the same page.

**We have now amended the text, such that it now reads:**

**"If the permeability of a volcanic system (including the edifice) is high, volatiles may escape from the magma and the propensity for explosive behaviour reduced".**

**Further on in the text, we now state:**

**"... along pre-existing planes of weakness such as zones of amœboid pores (pores that are rounded but highly irregular in shape) or microfractures..."**

(3) Page 2 line 14; not all of the papers cited here are explicitly in support of the statement made.

Inelastic compaction and permeability evolution in volcanic rock
journal article response
en

**Response to reviewer:**

**"Inelastic compaction and permeability evolution in volcanic rock"**

**A fair point. Nevertheless, while not all of these articles explicitly state this, the idea of outgassing having a direct influence on eruptive behaviour is an important foundation for each of them. For clarity, we have now changed the text to reflect this:**

**"If the permeability of a volcanic system (including the edifice) is high, volatiles may escape from the magma and the propensity for explosive behaviour reduced; on the other hand, low system permeability could promote pressure build-up and violent eruptive activity, a concept underlying numerous studies in volcanology (for example, Eichelberger et al., 1986; Woods and Koyaguchi, 1994; Edmonds and Herd, 2007; Mueller et al., 2008; Mueller et al., 2011; Nguyen et al., 2014; Okumura and Sasaki, 2014; Gaunt et al. 2014)."**

(4) Page 2 line 31; the authors state that porosity is the reciprocal of bulk density, which is not true. It's clear what the authors mean but, as stated above, they should be very clear in the language they use and check throughout the manuscript that they all agree on each sentence.

**We have now amended this sentence for clarity: "Bulk rock density and porosity have been estimated during..."**

(5) Page 2 line 40; by this point in the manuscript I thought it would be illustrative to have a diagram that schematically explained the concepts of this paragraph. If the authors felt inclined to provide one I think it would strongly improve the understanding of the regimes to which they refer.

**While it is a fair point that these concepts may be somewhat foreign to some readers, there are other articles that aptly demonstrate the influence of effective pressure and porosity on the transition between brittle and ductile behaviour in different rock types. We would highlight also that this is not the primary topic of this contribution. As such, we contend that referring the reader to another paper is preferable at this point. The text now reads:**

**"In detail, high effective pressures and/ or high initial porosity promote ductile behaviour, whereas dilatant brittle failure is favoured in low-porosity volcanic rock deformed under a range of effective pressures. For a comprehensive (though not exhaustive) study regarding the influence of effective pressure and porosity on the failure mode of volcanic rock, the reader is referred to Heap et al. (2015)."**

(6) Page 3 line 2; the reference to Farquharson et al., 2016b should occur at the end of the sentence – in a few places the authors put a reference in the middle of a sentence without it always being necessary.

**This citation now appears at the end of the sentence as suggested.**

(7) Page 3 line 7; do they authors mean "monotonically" when they say "monotonously"?

**Yes. Well spotted.**

(8) Page 4 line 1; on all measurements of a value, the authors should give an experimental uncertainty. The density should have an uncertainty quoted with it, please.

**Response to reviewer:**

**"Inelastic compaction and permeability evolution in volcanic rock"**

**Agreed. This has now been addressed in the manuscript (please refer to the responses to the Editor comments, above).**

(9) Page 4 line 3; if the authors use a correction for raw permeability data, could they please show it explicitly. As far as I know they are far from any length limit associated with this journal and so they could take the space to be as thorough as possible. An equation for the correction and a plot to which the reader could refer to see how the correction works would be valuable.

**This is now addressed explicitly by adding an appendix to the paper. The appendix "Measuring permeability" describes the method used to measure permeability in greater detail than the main body of the text, and outlines the requirement and application of both the Forchheimer and Klinkenberg corrections. An additional figure has been included to provide examples of laminar and turbulent flow in volcanic materials, as well as the gas slippage effect. We also now state that "[f]or the LLB data presented in Table 2 Forchheimer corrections were applied to the raw values where appropriate. Permeability values were affected only slightly, increasing by a factor of between 1.03 to 1.41."**

(10) Page 4, line 9; what does a poroelastic constant of unity mean in physical terms? It seems to imply that the pressures are isotropically distributed in the solid matrix. Does this imply something about the compressibility? Could the authors unpack this useful equation a little in the text that follows?

14) Page 4, line 38; if \alpha is different for different properties when everything else is the same, then it is not the same coefficient – i.e. it is not \alpha. In which case don't use the same symbol. This took me a few reads to understand and would be clearer if the authors simply referred to, for example, \beta, and named it another poroelastic constant that is different from \alpha. This does beg the question of why it's different and what physics \beta is tracking that \alpha is not.

**The reviewer is correct in their statement that α is related to compressibility. The value characterises the poroelastic response of a material to volumetric deformation, comprising the ratio between $K$, the drained bulk modulus of a material (the reciprocal of the compressibility of a medium under drained conditions; i.e. where the material is fully saturated throughout deformation), and $H$, which is the reciprocal of the poroelastic expansion coefficient ($1/H$). $1/H$ describes the bulk volume change resulting from a change in pore pressure alone (i.e. when the applied stress is a constant).**

**The Biot-Willis coefficient α is thus of the form $\alpha = K/H$, and is the ratio of volume of fluid that is added to storage divided by the change in bulk volume under the constraint that pore pressure remains constant, i.e. the volumetric responses of the fluid and solid components of a deforming sample.**

**In conventional triaxial deformation experiments, $\alpha$ is assumed to equal 1. In practice, this means that the failure strength of a sample deformed under $p_c$ and $p_p$ of 50 and 10 MPa, respectively, would be identical to the failure strength of the same sample deformed**

at pressures of 150 and 110 MPa (in either case, the nominal $p_{eff}$ is equal to 40 MPa. By contrast, if the volumetric response of the fluid and solid constituents are unequal and $\alpha$ < 1, then the stress regimes in the two scenarios will differ. In essence, this means that the measured failure stress of a sample would be different in each scenario, all other parameters being equal.

While the effective pressure effect on failure strength is well-known to be a function, in part, of the Biot-Willis coefficient, this is not as well constrained for permeability (i.e., the permeability of a sample may well differ when measured under different confining pressures, but this is not necessarily a sole function of compressibility and poroelastic expansion).

To avoid confusion, we have altered the text:
"Investigations towards determining the influence of effective pressure on properties (including permeability) other than rock strength indicate that their evolution with pressure may differ as a function of porosity, pore geometry, and other factors, which is to say that the effective pressure coefficient for one property may not be the same as the Biot-Willis coefficient α (Bernabe et al., 1986). Given the lack of constraint on the effective pressure effect for the permeability of volcanic rocks, permeability is measured at the lowest possible confining pressure..."

However, as we do not explicitly investigate the value or influence of α in this contribution, we feel that expounding on its derivation may be distracting and unnecessary in the context of the manuscript.

(11) Page 4, line 11; the authors state that they apply a pore pressure. Presumably they do not apply this to any isolated pores. This would need stating clearly here.

The reviewer is correct to assume that pore pressure is not controlled within any isolated porosity. We now state in the text:

"Note that pressure is only controlled within the connected porous network (i.e. $\phi$). However, we assume that any influence of incomplete sample saturation is negligible, due to the relatively small volume of isolated porosity in these andesites (see Table 2)."

(12) Page 4, line 18; I am familiar with the justification of a strain rate of 10^-5 s^-1 in rock mechanics studies on the grounds of "standard practice". However, do the authors perhaps have some justification for using this rate on more physical grounds? For example, Heap & Wadsworth (2016) among many other studies showed that a dominant control of strain rate in permeable rock is to shift regime from effectively permeable on the timescale of deformation, to effectively impermeable. It's ok to say that this rate was used for experimental expediency but it would more satisfying to see if it could be justified.

The reviewer is correct in noting that the rate of $10^{-5}$ $s^{-1}$ is typically employed as it is the standard practice in rock deformation fields. However, evidence suggests that this value is encompassed by the range of strain rates encountered in natural systems. For completeness, we now include the following paragraph in the manuscript where suggested:

Inelastic compaction and permeability evolution in volcanic rock
en

**Response to reviewer:**

**"Inelastic compaction and permeability evolution in volcanic rock"**

**"We note that—on the edifice-scale—absolute strain rates resulting from magma migration and edifice displacement are generally of the order of $10^{-7}$ s$^{-1}$ or lower (for example as estimated from borehole strainmeters: Linde et al. 1993, or from spaceborne interferometry: Massonnet et al., 1995). However, strain and strain rates are undoubtedly highly variable throughout active volcanic systems. The chosen strain rate for these experiments—the international standard in rock mechanics (Kovari et al. 1983; see also Ulusay and Hudson 2007)—is comparable to shear rates inferred to occur along conduit margins by Rust et al (2003). Similarly, Cashman et al. (2008) estimate strain rates of 3-8 ×$10^{-5}$ s$^{-1}$ for the formation of fault gouge at Mount St Helens (USA). Most importantly, a strain rate of $10^{-5}$ s$^{-1}$ ensures that our samples are drained (i.e. the product of the Darcy timescale and the strain rate is << 1: Heap and Wadsworth, 2016). "**

(13) Page 4, line 20; do the authors mean d\phi and not \delta\phi? I suspect they do. In which case please change throughout.

**Typically $d$ is used to note a full derivative or differential (as opposed to the symbol $\partial$ for a partial derivative). Delta $\Delta$ or $\delta$ are more commonly employed to represent a quantifiable change in some property, which is its purpose here (specifically, the inelastic change in porosity). To avoid confusion, we have now changed all instances of $\delta$ to $\Delta$ throughout the text.**

(15) Page 5, line 6; why would you need the mean radius?

**We agree that this is perhaps unclear. The key property here is not the sample radius, but $A$, the cross-sectional area of the sample for fluid flow.**

**The permeability of a medium, measured using an ideal compressible fluid under atmospheric conditions can be described in the following manner:**

$$k = \frac{Q_v \nu L}{A} \frac{p_{atm}}{\nabla p \cdot \overline{p}}.$$

**An important parameter is $A$, the cross-sectional area of the sample for fluid flow. While this is easily constrained for a perfect cylinder, it is less trivial for a deformed cylinder. We employ the equation shown in the text to solve for the mean value of $A$.**

(16) Page 5, line 10; this equation begs an explanation. Why is this the answer to an effective cross-sectional area in a deformed sample? This is not explained at all.

**As addressed in response to the Editor's comments, this equation is essentially just the relation between area, volume, and length. The necessary values required for determining permeability  are $A$ and $L$. While we can measure the latter quantity, it is difficult to practically measure $A$ when the sample is very deformed. However, we can determined $V$ as a function of the solid rock volume, and the pore volume in the post-deformation sample. For clarity we have now presented the equation in a different form and expanded the text (see above in response to Editor's comments).**

(17) Page 5, line 32; it's clear here that the authors mean an axial strain. Please can they go carefully through the manuscript and be specific about strain definitions. Otherwise they could

**Response to reviewer:**

**"Inelastic compaction and permeability evolution in volcanic rock"**

say up front that throughout this work "strain" is used to mean "axial strain". This is important as they talk about other forms of strain.

**As suggested previously by the reviewer, we now state that "strain" refers to axial strain unless otherwise specified".**

(18) The discussion is robust and I have no commentary.

**We thank the reviewer for their assessment of the discussion.**

(19) The figures are well presented. However Figure 5, which is the take-home message of the work, is not clearly presented. I would recommend some analysis be explored, such as a "change in permeability" as a function of inelastic strain or some metric that scales with the effective pressure of deformation? The reader might be looking for some take home conclusion which they don't find here. I do not see clearly, for example, that permeability increases and then decreases, as the authors state.

**Figure 3b shows this increase and subsequent decrease. This would be clearer no doubt if the data were connected by lines, we choose not to as they are not the same samples (i.e. they are independent data). While we appreciate the reviewer's stance that it is important to have a "take-home" figure, we are wary of over-analysing the compiled data of Figure 5, given that there are few comparable data available at high axial strains. However, in response to another comment by the reviewer, we have included another section in the manuscript that applies a simple flow model to our data, and address the effective pressure component in this way.**

(20) A central conclusion of the work seems to be this: that in the brittle regime permeability increases because strain is accommodated by discrete fractures. And that this is different from the compaction regime in which pore crushing produces diffuse areas (compaction bands) of granular material. The novel idea here is that the effective pressure controls the particle sizes in these zones which in turn controls the permeability limit during progressive compaction. However, this idea is left hanging without much additional support or exploration. There are many permeability models for granular media, including polydisperse examples (e.g. Wadsworth et al., 2016), which could be used to confirm this idea. It's exciting and therefore a little bit of a shame to leave it unexplored with simply a statement that it would make a good future study. If the authors felt inclined to unpack this idea a little, it would be worthwhile.

**We appreciate the suggestion from the reviewer that a simple model could be employed in order to explore the dependence of sample permeability on effective pressure. We now include an additional segment that addresses this concept specifically. This section, highlighted in the accompanying revised manuscript, models a compaction band-bearing sample as a layered medium, wherein the compaction band itself is envisaged as a granular bed. This new section is highlighted in the accompanying manuscript. Indeed, this analysis shows us that the permeability reduction within a discrete compaction band is sensitive to the characteristic grain size. However, the overall trade-off between the width and porosity of a compaction band are far more important that grain size in controlling fluid flow throughout the bulk of a sample. The introduction and conclusion have been updated to include our modelling section.**

**Response to reviewer:**

**"Inelastic compaction and permeability evolution in volcanic rock"**

**Response to reviewer:**

**"Inelastic compaction and permeability evolution in volcanic rock"**

**Revised manuscript and supplementary files**

In the following revised manuscript, new text is blue and underlined, and deleted text is red and crossed-through.

[revised manuscript text omitted]
.  To date, physical property data at depth in volcanic materials has been obtained predominantly by researchers assessing the suitability of volcanic deposits for hydrocarbon or geothermal energy exploitation. For example Chen et al. (2016, 2017) investigated volcanic sequences in the Junggar Basin in western China, in order to determine their suitability as gas and petroleum reservoirs. At both sites, the authors note a general decrease in porosity with depth; for example, Chen et al. (2016) report a decrease of porosity from ∼0.30 to <0.10 between the surface and 1000 m depth. Often, the interpretation of logging data from volcanic materials (e.g. Millett et al., 2016) is non-trivial due to the variable quality of density logs and variations in the relationship between density and facies (Li et al., 2009), as well as gaps in the stratigraphic record due to incomplete core recovery. An example of a study with an excellent degree of core recovery—99.7 %—is reported by Jónsson and Stefánsson (1982): these authors calibrate porosity and density data obtained by the Iceland Research Drilling Project from a borehole cored continuously to a depth of 1919 m near Reyðarfjörður in Iceland. To combat small-scale variability arising from the intersection of discrete geological units, Jónsson and Stefánsson (1982) calculate a running average of porosity and density against depth. Notably, the average porosity decreases from 0.13 at 400 m depth to 0.06 at 1200 m depth, corresponding to an increase in bulk rock density of approximately 200 kg m$^{-3}$.

[revised manuscript text omitted]

$$\phi = \left( \frac{\rho_b - \rho_v}{\rho_v} \right) \tag{1a}$$

$$\phi_t = 1 - \left( \frac{\rho_b}{\rho_p} \right) \tag{1b}$$

$$\phi_u = \phi_t - \phi \tag{1c}$$

where a  value of 2653 $\pm$ 0.17 kg m$^{-3}$ is used for  $\rho_p$. In detail, helium pycnometry is used to determine the solid volume $V_s$ of each sample, which subsequently allows the calculation of the connected gas porosity $\phi$ as described. Automated measurements of $V_s$ were performed iteratively until five consecutive measurements yielded results within a range of 0.01% of the sample volume, so precision of the pycnometer measurements is high ($< 0.005$ cm$^3$). A greater degree of error arises when manually measuring the sample dimensions, required in order to determine $\rho_v$. Repeat measurements allowed an estimation of error in the length and diameter, which typically amount to $< 0.05$ cm$^3$ in terms of volume. Adopting the notation that $\epsilon_\phi$ is the error on the porosity calculation and that $\epsilon_x$ and $\epsilon_y$ are the independently calculated errors for measurements of $V_s$ and $V$, then the propagated error can be approximated by:

$$\epsilon_\phi = \phi \times \left[ \left( \frac{\epsilon_z}{V - V_s} \right)^2 + \left( \frac{\epsilon_y}{V} \right)^2 \right]^{\frac{1}{2}} ; \; \epsilon_z = \left( \epsilon_x^2 + \epsilon_y^2 \right)^{\frac{1}{2}} \tag{2}$$

Values for $\epsilon_\phi$ are generally $< 0.005$ for the samples described herein. As such, probable error on connected gas porosity measurements is low, and always contained within the symbol size when plotted graphically.

Gas permeability was measured under steady-state conditions with a confining pressure of 1 MPa using the setup

**Table 1.** Major element (oxide) composition, determined via X-ray fluorescence analysis. All values are given in weight percent (wt. %).

| $SiO_2$ | $Al_2O_3$ | $Fe_2O_3$ | $MnO$ | $MgO$ | $CaO$ | $Na_2O$ |
|---|---|---|---|---|---|---|
| 61.260 | 17.330 | 5.745 | 0.100 | 3.725 | 5.505 | 4.455 |
| $\pm$ 0.270 | $\pm$ 0.070 | $\pm$ 0.085 | $\pm$ 0.001 | $\pm$ 0.295 | $\pm$ 0.025 | $\pm$ 0.015 |

| $K_2O$ | $TiO_2$ | $P_2O5$ | $Cr_2O_3$ | $V_2O_5$ | $NiO$ | LOI |
|---|---|---|---|---|---|---|
| 1.505 | 0.565 | 0.200 | 0.020 | 0.015 | 0.004 | 0.185 |
| $\pm$ 0.035 | $\pm$ 0.015 | $\pm$ 0.020 | $\pm$ 0.005 | $\pm$ 0.005 | $\pm$ 0.001 | $\pm$ 0.215 |

$SiO_2$ = Silicon dioxide; $Al_2O_3$ = Aluminium oxide; $Fe_2O_3$ = Iron oxide; $MnO$ = Manganese(II) oxide;
$MgO$ = Magnesium oxide; $CaO$ =Calcium oxide; $Na_2O$ = Sodium oxide; $K_2O$ = Potassium oxide;
$TiO_2$ = Titanium oxide; $P_2O_5$ = Phosphorus pentoxide; $Cr_2O_3$ = Chromium(III) oxide; $V_2O_5$ =
Vanadium(V) oxide; $NiO$ = Nickel(II) oxide; LOI = loss on ignition.

described in ?Farquharson et al. (2016c). Where necessary, a correction was applied to the measured permeability values to account for turbulent flow (see Forchheimer, 1901), the effects of which can become non-negligible when mea-
5 suring the permeability of high-porosity media. Appendix A contains further details on the determination of permeability, the application of corrections for inertial effects, and the sources and sizes of potential error in the measurements.

Samples were then encased in a copper foil jacket (which
10 serves to retain bulk sample cohesion after deformation), saturated with distilled water, and loaded into the triaxial deformation rig at Université de Strasbourg (see –Figure 1). Throughout deformation we assume a simple effective stress law, whereby the effective confining pressure $p_{eff}$ experi-
15 enced by a sample is a function of the confining pressure $p_c$ around the sample and the pressure of pore fluid $p_p$ within the sample, such that $p_{eff} = p_c - \alpha \cdot p_p$. Recent experimental work (Farquharson et al., 2016a) shows that $\alpha = 1$ is a reasonable assumption for porous andesite.
20 For each test, the confining and pore pressures were increased slowly until a targeted effective pressure (i.e. hydrostatic pressurisation). Note that pressure is only controlled within the connected porous network (i.e. $\phi$). However, we assume that any influence of incomplete
25 sample saturation is negligible, due to the relatively small volume of isolated porosity $\phi_u$ in these andesites (see Table 2). Assuming a pycnometry-derived value for bulk density $\rho_b$ of approximately 2100 kg m$^{-3}$, the imposed effective pressures of 10, 30, 50, and 70 MPa are analogous
30 to depths ranging from the upper 500 m of the edifice to greater than 3 km in depth (given that $p_c \propto \rho_b \cdot g\hat{z}$, where $g$ and $\hat{z}$ are surface gravitational acceleration and depth, respectively). The sample would then be left overnight to allow microstructural equilibrium. During the deformation
35 experiments, a differential stress was introduced in the direction of the sample axis by advancing an axial piston (see –Figure 1) under servo-control, such that the sample is subjected to a constant strain rate of 10$^{-5}$ s$^{-1}$ –(note that hereafter "strain" refers to axial strain unless otherwise
40 specified). We note that – on the edifice-scale – absolute

strain rates resulting from magma migration and edifice displacement are generally of the order of 10$^{-7}$ s$^{-1}$ or lower, for example as estimated from borehole strainmeters: Linde et al. (1993), or from spaceborne interferometry:
45 Massonnet et al. (1995). However, strain and strain rates are undoubtedly highly variable throughout active volcanic systems. The chosen strain rate for these experiments (the international standard in rock mechanics Kovari et al., 1983; Ulusay comparable to shear rates inferred to occur along
50 conduit margins by Rust et al. (2003). Similarly, Cashman et al. (2008) estimate strain rates of 3 - 8 $\times$ 10$^{-5}$ s$^{-1}$ for the formation of fault gouge at Mount St Helens (USA). Most importantly, a strain rate of 10$^{-5}$ s$^{-1}$ ensures that our samples are drained (i.e. the product of the Darcy timescale and the strain rate is $\lll$ 1; Heap

Confining pressure and pore pressure were servo-controlled throughout the experiments. During hydro-static and nonhydrostatic loading, the response of the pore fluid pump reflects variations in pore volume
60 (see Read et al., 1995), which – normalised to the initial sample volume – corresponds to the porosity change $\delta\phi\Delta\phi$. For a porous material, $\delta\phi$–$\Delta\phi$ can be considered equal to the volumetric strain. Indeed, the response of the confining pressure pump provides an independent estimation of
65 the volumetric strain, found to be in perfect agreement with the inferred $\delta\phi$ (see Baud et al. , 2014 for details). $\Delta\phi$ (see Baud et al., 2014, for details). When this differential $\delta\phi$ $\Delta\phi$ is positive it signifies dilation (an increase in porosity) and when it is negative, it indicates compaction (a de-
70 crease in porosity). Deformation was allowed to continue for different amounts of axial strain accumulation – sample shortening relative to its original length ($\varepsilon_t$) – then unloaded. The strain recovered during the unloading phase is subtracted from the total axial strain $\varepsilon_t$ to give the inelastic
75 (non-recoverable) strain accrued by the sample ($\varepsilon_i$). Similarly, the elastic porosity change recovered during unloading is subtracted from the porosity change at $\varepsilon_t$, to give the inelastic porosity change $\delta\phi_i\Delta\phi_i$. Samples were subsequently vacuum-dried once again and gas permeability re-measured.
80

[Figure]

**Figure 1.** Schematic of triaxial deformation apparatus, including confining pressure $p_c$, pore pressure $p_p$, and axial pressure $p_{ax}$ circuits. Detail of sample assemblage is shown inset. (a) axial piston; (b) "blank" endcap; (c) sample; (d) copper foil jacket; (e) nitrile jacket; (f) pore fluid distributor endcap. Directions of major $\sigma_1$ and minor $\sigma_3$ principal stresses are as shown, such that $\sigma_1 > \sigma_2 = \sigma_3$. Not to scale. Numbered valves allow various parts of each circuit to be used at any given time.

Samples were all deformed at room temperature. We note that in natural volcanic environments there may be some influence of temperature on rock strength (due, for instance, to the closure of cracks driven by thermal expansion). Nevertheless, a recent study by Heap et al. (2017) shows that the influence of temperature on the physical and mechanical properties of andesite may not be significant at temperatures below the glass transition $T_g$. Importantly, this study showed that the failure mode and underlying microstructural mechanism driving ductile behaviour (cataclastic pore collapse) did not change below $T_g$, which is itself largely restricted to the magma conduit and rock in the immediate vicinity. This is in agreement with previous studies by Vinciguerra et al. (2005) and Heap et al. (2014), both of which noted only negligible changes in microcrack density and porosity after thermally stressing volcanic materials. These authors attribute this phenomenon to the high initial crack density resulting from the complex thermal histories of volcanic rocks.

**2.2 Post-deformation permeability**

It has been shown in recent studies  (e.g. Vinciguerra et al., 2005; Nara et al., 2011) that the permeability of fractured volcanic materials is influenced by the effective pressure under which it is measured: permeability tends to decrease with increasing effective pressure. As such, we acknowledge the limitation that post-deformation measurements do not represent the permeability under the deformation conditions *sensu stricto*. Nevertheless, we choose to measure permeability under the conditions described above for a host of reasons. Investigations towards determining  properties (including permeability) other than rock strength indicate that  their evolution with pressure may differ as a function of porosity, pore geometry, and other factors; which is to say that  the effective pressure coefficient for a given rock property may not be the same as the Biot-Willis coefficient $\alpha$  (Bernabé et al., 1986). Given the lack of constraint on  the effective pressure effect for the permeability of volcanic rocks, permeability is measured at the lowest possible confining pressure (1 MPa, rather than at "*in situ*" pressures) and without imposing a differential stress, in order to allow comparison within and between sample sets (indeed, we compare our data with compiled literature data in Sect. 4.1). This procedure also avoids the potential for creep – a mechanism of time-dependent deformation whereby sub-critical crack growth induces damage and possibly even failure at stresses below the short-term strength of the rock  (e.g. Heap et al., 2017) as well as precluding other phenomena such as stress relaxation that may arise when measuring permeability under  a differential stress.

Measuring permeability requires that the sample dimensions, specifically length and cross-sectional area, are  well constrained. Prior to initial measurements of permeability, sample dimensions are measured accurately using digital callipers. However,  samples are often barrelled and thus non-cylindrical after mechanical deformation, making their mean radii nontrivial to determine. Assuming that the solid volume $V_s$ remains constant throughout deformation, then the post-deformation volume is equal to the sum of solid volume, the initial  pore volume, and the pore volume change after deformation. The post-deformation cross-sectional area $A_{post}$ can therefore be determined such that

$$A_{post} = \frac{V_s}{1 - [\phi + \delta\phi_i]} \cdot l_{post}^{-1} \underbrace{\left[ \frac{V_s}{1 - (\phi + \Delta\phi_i)} \right]}_{V_{post}} \times \left[ \frac{1}{l_{post}} \right]$$

(3)

where $V_{post}$ is the post-deformation volume and $l_{post}$ is the mean sample length after deformation.

**3 Results**

[revised manuscript text omitted]

$\phi$ = connected porosity; $\phi_u$ = unconnected porosity; $p_{eff}$ = effective pressure; $\varepsilon_t$ = target (total) axial strain; $\varepsilon_i$ = inelastic axial strain; $\Delta\phi_i$ = inelastic porosity change; $\phi + \Delta\phi_i$ = post-deformation porosity; $k_0$ = initial permeability; $k_e$ = post-deformation permeability.

within an "intact" host with the initial rock properties (which is to say, a sample of porosity $\phi$). The intact material must have a pore volume $V_1^{\phi}$ of $\phi \times (l_{intact} \times A_{post})$, where $l_{intact}$ is the overall length of the sample that is undamaged $(l_{post} - w_b)$, where $w_b$ is the width of the compaction band. The deformed sample contains a pore volume $V_2^{\phi}$ of $\phi + \Delta\phi_i \times (l_{post} \times A_{post})$. From this, we can relate the compaction band porosity $\phi_b$ to the width of the compaction band:

$$\phi_b = \left[\frac{V_2^{\phi} - V_1^{\phi}}{A_{post}}\right] \times \left[\frac{1}{w_b}\right]. \tag{4}$$

Solutions for $\phi_b$ and $w_b$ are non-unique (moreover, a greater value of $w_b$ could be a function of one wide band or a number of discrete, relatively thinner bands), but we can impose a lower bound on $\phi_b$ of zero, and an upper bound equal to the post-deformation porosity of the sample: $0 \leq \phi_b < (\phi + \Delta\phi_i)$. Assuming the compaction band porosity noted by Heap et al. (2015a) ($\sim$0.10) is typical for compacted andesite, then Equation 4 yields compaction band widths of between 1.63 and 23.57 mm (i.e. between 4 and 70 % of the overall sample length). We note that the lower end of this range is in line with the observations of Heap et al. (2015a).

We note that porosity loss is seemingly tied to the effective pressure under which compaction occurs (as observed in previous studies concerned with triaxial rock deformation, e.g. Wong et al., 1997; Baud et al., 2006; Heap et al., 2015a): for a given  increment of inelastic strain, the porosity lost by a sample is  greater at a higher effective pressure. This phenomenon is true both for total porosity change ( Figure 2b) and for inelastic porosity loss ( Figure 3a), which is to say that the inelastic compaction factor  $\Delta\phi_i/\varepsilon_i$

always decreases as effective pressure increases (Baud et al., 2006).

While the mechanism of cataclastic pore collapse is governed by the pore sizes (e.g. Zhu et al., 2010, 2011), it has also been demonstrated that the local stress field around a pore increases as a function of the incumbent confining pressure (Zhu et al., 2010). A  study of fault gouge formation in sandstone (Engelder, 1974) shows that fault-zone fragments are smaller when generated at higher confining pressures, and a similar effect (albeit less pronounced) was noted by Kennedy et al. (2012), who investigated fault gouge formation in dacitic dome rock.  It is reasonable to assume that cataclasis may become more efficient as the local stress field increases in line with the confining pressure; in turn, a finer distribution of fragments will more readily occlude the pores around which they develop. Whether a change in the mean fragment size generated during cataclastic pore collapse underlies the observed evolution of  $\Delta\phi_i/\varepsilon_i$ remains open to a targeted microstructural study. Nevertheless, we can interrogate Equation 4 to glean an idea of the effect of $\phi_b$ and $w_b$ at different effective pressures and axial strains.

Figure 5 shows the calculated $w_b$ for our experimental data as a function of inelastic axial strain. Values of $w_b$ are calculated using Equation 4, using values of $\phi_b$ of 0.20, 0.15, 0.10, and 0.05. At relatively higher imposed values of $\phi_b$ (0.20 and 0.15: Figure 5a, 5b), many of the resulting values of $w_b/l$ are non-physical (i.e. $w_b/l \not> 0$ or $w_b/l \not< 1$). However, at lower imposed values of $\phi_b$ (0.10 and 0.05: Figure 5c, 5d), values fall between 0 and 1. Moreover, there appears to be a systematic effect of $p_{eff}$, with deformation under relatively higher effective pressure yielding a higher ratio of $w_b/l$ – hence, a thicker compaction band – for any given amount of inelastic axial strain accumulation.

[Figure]

**Figure 3.** Change in physical properties after accumulating inelastic axial strain. (a) Pre- and post-deformation porosity (the latter calculated by $\phi + \delta\phi_i$ $\phi + \Delta\phi_i$) versus inelastic strain. Effective pressure is indicated by the symbol colour. (b) Pre- and post-deformation gas permeability as a function of the inelastic strain accumulated by each sample. Effective pressure is indicated by colour as in panel (a): note, however, that the value of $p_{eff}$ corresponds to deformation, whereas permeability was measured in each case at $p_{eff} = 1$ MPa.

As would be expected (e.g. Zhu and Wong, 1997), permeability reduction follows the same general trend

[Figure]

**Figure 4.** Backscattered scanning electron microscope images of LLB andesite, showing as-collected (a) and post-deformation (b - c) microstructure. Void space appears as black. Dense (metal-rich) phenocrysts appear as white or light grey within a darker grey groundmass. Both (b) and (c) are images of LLB-13, which was taken to beyond both $C^*$ and $C^{*\prime}$. Cataclastic pore collapse associated with shear-enhanced compaction is shown in (b), while (c) shows part of the dilatant shear zone marking the transition from shear-enhanced compaction to dilation.

[Figure]

**Figure 5.** Calculated sample length ratio $w_b/l$ as a function of inelastic axial strain accumulation $\varepsilon_i$. Values of $w_b$ calculated from Equation 4 using values of compaction band porosity of $\phi_b = 0.20$ (a), $\phi_b = 0.15$ (b), $\phi_b = 0.10$ (c), and $\phi_b = 0.05$ (d). Shaded area indicates range of physical values of $w_b/l$.

(—Figure 3b) as porosity reduction (—Figure 3a), with samples accumulating high strains showing a correspondingly large reduction in permeability.  Notably, there appears to be an influence of the effective pressure under which the sample was deformed and the change in permeability for a given increment of axial strain. A difference in measured post-deformation permeability $k_e$ may be due to (1) a variation in characteristic grain size, or (2) a variation in the thickness of compaction localisation features with respect to the sample length, as described above. Moreover, one may imagine that these two factors (characteristic grain size, band thickness) operate in tandem to reduce permeability as effective pressure increases. To test this theory, we again model the deformed samples as a layered medium, such that discrete bands of uniform permeability $k_b$ and thickness $w_b$ are embedded in a medium of permeability $k_0$ (see Figure 6a). A similar approach has previously been adopted by Vajdova et al. (2004) and Baud et al. (2012) to model the permeability of sandstones containing experimentally-induced compaction bands. Fluid flow through this simplified geometry may then be modelled by assuming conservation of mass (e.g. Freeze and Cherry, 1979), such that:

$$k_e = k_0 \times \left[ \left( \frac{w_b}{l} \right) \left( \frac{k_0}{k_b} - 1 \right) + 1 \right]^{-1} \qquad (5)$$

Variables $k_e$, $k_0$, and $l$ are already constrained, allowing us to solve for combinations of $w_b$ and $k_b$. If we assume that a compaction band comprises a granular bed (Figure 6a *inset*), we can relate its permeability $k_b$ to surface area $s$ in the following manner after Martys et al. (1994), who determined a universal scaling of permeability of a system of packed spheres:

$$k_b = \frac{2(1 - \phi^*)}{s^2} \phi^* f \qquad (6)$$

where $\phi^* = \phi_b - \phi_c$, and $f = 4.2$ (the value of $f$ is thought to be related to the initial grain geometry: Wad $\phi_c$ represents the percolation threshold, taken here as 0.03. Note that the characteristic porosity is taken as the porosity within a compaction band $\phi_b$ (i.e. the porosity of the granular layer with permeability $k_b$ and width $w_b$). In turn, we can relate $s$ to a characteristic grain size (e.g. Wadsworth et al., 2016):

$$s(r) = \frac{3(1 - \phi_b)}{r} \qquad (7)$$

where $r$ is the monodisperse particle radius. Note that in reality, the porosities assumed within the compaction bands here are not compatible with a monodisperse packing of spheres. Nevertheless, this greatly simplified approach gives an indication of the relative influence of the difference constituent parameters $r$, $\phi_b$, and $w_b$.

The assumed geometry is illustrated in Figure 6a, including the corresponding values of permeability and porosity for each layer. Figure 6b highlights the effects of changing either the characterisitic particle (i.e. grain) radius or the compaction band porosity. Notably – for a given porosity – a change in particle radius of 1 order of magnitude results in a change in compaction band permeability of 2 orders of magnitude. At relatively high initial porosities, a reduction in porosity by a given volume (for example, from 0.20 to 0.15) has little influence on $k_b$. On the other hand, when the porosity is low, a change in porosity of the same absolute volume (for example, from 0.10 to 0.05) exerts a much greater influence over $k_b$ (in this case, a reduction

by over 2 orders of magnitude). However, the bulk sample permeability (the equivalent permeability) depends not only on the porosity of the compaction band but also its width. Figures 6c–6f show the equivalent permeability for different values of $\phi_b$ for changing values of $w_b/l$: the ratio of the band width to the overall length of the sample. Curves are modelled by combining Equations 5, 6, and 7, using the particle radii $r$ (noted on each figure panel) and a value of $k_0 = 5.0 \times 10^{-13}$ m$^2$. As there are multiple non-unique solutions for $\phi_b$ and $w_b$, we show model results for a range of potential $\phi_b$ values in panels 6c to 6f.

Evidently, the variation in our experimental data (for example, the difference between $k_e$ of samples deformed under different effective pressures) is not explained by a systematic evolution of $r$. This suggests that while $k_b$ is very sensitive to the characteristic grain radius, the tradeoff between $\phi_b$ and $w_b$ is more important in controlling the bulk sample permeability ($k_e$). Importantly, however, idealising the geometry of a compaction band in terms of a monodisperse particle size distribution cannot accurately represent its complex porous network. More accurate values of $\phi_b$ and $s$ (and hence, a better prediction of $k_b$ and $k_e$) may be acheived by adopting a polydisperse particle size distribution or by imposing a non-spherical characteristic particle shape, for example.

An evident weakness of employing the simple layered medium model outlined above is that we assume that the only operative mechanism is porosity- and permeability-reducing. However at low strains, there is no one-to-one relationship between permeability and porosity after deformation. Rather, permeability tends to increase moderately at inelastic axial strains less than around 0.05 (i.e. a 5% shortening in sample length): while initial values of permeability tended to be around $5 \times 10^{-13}$ m$^2$, the measured post-deformation permeability was often greater than $10^{-12}$ m$^2$ after accumulating a small amount of axial strain (Table 2). A similar phenomenon was also observed by Loaiza et al. (2012), who noted that permeability of Açores trachyandesite increased beyond a critical stress state during hydrostatic pressurisation. This critical stress – known as $P^*$: Zhang et al. (1990) – signals the onset of lithostatic inelastic compaction; Loaiza et al. (2012) show that stress-induced cracks coalesce between collapsed pores during hydrostatic compaction, improving connectivity and, in turn, increasing permeability. Prior to deformation, the samples of porous andesite used in our experiments – LLB – contained an isolated porosity of 0.01, on average (Table 2).

Similar to the mechanism posited by Loaiza et al. (2012), we suggest that distributed microcracking during the initial stages of ductile deformation serves to interconnect this isolated porosity, creating efficient pathways for fluid flow. The mechanical data of all the experiments (Figure 1a) exhibit intermittent stress drops – even in the instances where permeability was observed to increase relative to the initial value – which suggests that compaction localisation in these andesites does not necessarily equate to the formation of an effective barrier to fluid flow. Loaiza et al. (2012), Adelinet et al. (2013), Heap et al. (2015a), and Heap et al. (2016) each examine microstructure of compaction bands formed in porous volcanic rocks (trachyandesite, basalt, andesite, and dacite, respectively). In all cases, the bands are irregular in shape and thickness, but do not necessarily constitute a contiguous surface of collapse pores (i.e. a layer of reduced porosity). This is supported by the results from a recent study by Baud et al. (2015), which examined compaction band-bearing sandstones using X-ray computed tomography. The authors show that when compaction bands are formed in a rock with porosity clusters, their path is more tortuous than in material with homogeneous porosity; consequently the bands do not comprise efficient permeability barriers in three dimensions. Notably, the porosity of the andesites deformed in our study exhibit marked heterogeneity in terms of porosity, pore shape, and pore size distribution (Figure 4a). The characteristic tortuosity of compaction bands formed in heterogeneous volcanic material – and the fact that the reduction in porosity relative to the host sample is remarkably less than observed in sandstones (Baud et al., 2012) – may explain the lack of an obvious observed influence on sample permeability in this study. Thus, counter-intuitively, small amounts of stress-induced compaction may actually increase permeability in volcanic materials. At higher strains however, this effect is overtaken by the global reduction in sample porosity, which serves to decrease the mean flow path aperture and forces fluids to travel through more tortuous routes.

**4.2 A limit to compaction and permeability reduction**

Porosity exerts a first order control on the brittle-ductile transition of porous rocks (e.g. Wong and Baud, 2012): high porosity fosters ductile behaviour in response to an applied differential stress whereas dilatant brittle behaviour is anticipated in low-porosity materials. If a high-porosity volcanic rock undergoes progressive compaction, it will eventually achieve a porosity low enough to respond in a dilatant fashion. The critical stress state at this transition is known as $C^{*\prime}$ has been previously described in sedimentary materials (e.g. Schock et al., 1973; Baud et al., 2000; Vajdova et al., 2004; Baud et al., 2006; Regnet et al., 2015), and recently in andesite from Volcán de Colima (Heap et al., 2015a). As anticipated, beyond a threshold stress-strain accumulation, sample LLB-13 exhibits a transition from compactant deformation to dilatant behaviour (highlighted by the arrow in Figure 1b). In volcanic rock – at the sample-scale – this is characterised by significant shortening and barrelling of the sample (a consequence of the high strains required to achieve and exceed $C^{*\prime}$) and the generation of a dilatant shear zone (Heap et al., 2015a), characteristically similar to

[Figure]

**Figure 6.** Model geometry and results. (a) Deformed sample may be thought of as a granular bed (with permeability $k_b$ and porosity $\phi_b$) within a sample matrix with permeability $k_0$ and porosity $\phi + \Delta\phi_i$. Inset highlights the characterisitc grain radius that governs surface area (Equation 7). (b) The permeability of a compaction band composed of packed spheres as a function of sphere radius and bed porosity. Note that in reality, porosities < 0.26 would require a polydisperse packing of spheres, or a granular bed composed of non-spherical grains. (c) – (f) Modelled equivalent permeability $k_e$ of a compaction band-bearing sample, plotted against the ratio of compaction band width relative to the overall sample length ($w_b/l$), for compaction band porosities of $\phi_b = 0.20$ (c), $\phi_b = 0.15$ (d), $\phi_b = 0.10$ (e), and $\phi_b = 0.05$ (f). Also plotted are data from this study (Table 2), where values of $w_b$ are calculated after Equation 4. Note that of the ten $k_e, w_b/l$ data pairs from this study, not all are plotted on each of the panels (c) to (f). This is because certain combinations of $\phi_b$ and $w_b$ yield non-physical values (Equation 4). Refer to text for further discussion.

[revised manuscript text omitted]
, possibly due to enhanced communution. By modelling a simple sample geometry where a compaction band is represented by a packed granular bed, we show that the permeability reduction within a discrete compaction band is sensitive to the characteristic grain size. The effect of the porosity of a compaction band has a variable influence on its permeability, with changes in band porosity becoming ever more important as porosity decreases. However, the overall trade-off between the width and porosity of a compaction band are far more important that grain size in controlling fluid flow throughout the bulk of a sample. At low strains ($< 0.05$), compaction tends to result in a moderate increase in permeability (not accounted for in our model), which we suggest is a result of increased pore connectivity due to distributed microcracking. This effect is outweighed by progressive compaction at higher strains, resulting in a general trend of decreasing permeability with ongoing inelastic compaction. There exists a physical limit to compaction, which we suggest is echoed in a limit to the potential for permeability reduction in a deforming sample. Compiled data show that at high strain, porosity and permeability tend to converge towards intermediate values (i.e. $0.10 \leq \phi \leq 0.20$; $10^{-14} \leq k \leq 10^{-13}$ m$^2$). Field evidence from the literature emphasises the importance of understanding the physical and mechanical properties of rock in active volcanic environments, in particular the evolution in a rock's capacity to effectively transmit magmatic volatiles.

**6    Data availability**

Data are presented in Table 1 and  2 in the text, or are available on request to the corresponding author.

**Appendix A:  Measuring permeability**

The constitutive equation governing fluid transport in porous and granular media was originally derived from experiments performed by Henry Darcy in the 1850s on the flow of water at different levels through sand. Since Darcy's work (1856), the theoretical framework of fluid transport—which is based on Newton's second law—has been well established and expanded, such that flow of gas through a porous medium may be given thus:

$$Q_v = \frac{-kA}{\mu} \frac{(p_b - p_a)}{l} \tag{A1}$$

where $\mu$ is the fluid viscosity, $Q_v$ is the volumetric flow rate, $A$ is the cross-sectional area available for flow and $l$ is the distance over which fluid flow occurs (i.e. the sample length). In a fluid transport system, flow is driven towards the region of lowest potential energy: in the special case of horizontal flow, this may be described by a differential between a region of relatively high pressure $p_b$ to one of relatively lower pressure $p_a$: a pressure differential or pressure drop $\nabla p$. Equation A1 is valid for all porous media as long as flow is laminar (two cases of non-laminar flow are discussed hereafter). While this expression is sufficient for the case of laminar (or "streamline") flow, when considering an ideal compressible gas measured under atmospheric conditions, it becomes convenient to present gas permeability $k_{gas}$ in the following manner (Klinkenberg, 1941; McPhee and Arthur, 1991):

$$k_{gas} = \frac{Q_v \mu l \cdot p_{atm}}{A \cdot \nabla p \bar{p}} \tag{A2}$$

where $p_{atm}$ is the atmospheric pressure at which $Q_v$ is measured, and the driving pressure is given as a product of the differential pressure $\nabla p$ and the mean pressure over the sample $\bar{p}$. The mean pressure $\bar{p}$ is determined

by the upstream and downstream pressures $p_b$ and $p_a$ such that $\bar{p} = (p_b + p_a)/2$. Under ambient conditions, $p_a$ is equal to the atmospheric pressure $p_{atm}$, and $p_b$ is equal to $\nabla p + p_{atm}$. The mean pressure therefore simplifies to $\bar{p} = (\nabla p + 2p_{atm})/2$.

Herein, gas permeability was measured using a steady-state permeameter using the setup described in Farquharson et al. (2016c). The apparatus is a commercial benchtop permeameter from Vinci Technologies, modified by incorporating interchangeable El-Flow volumetric mass flowmeters (from Bronkhorst) to measure the volumetric flow rate of gas at the downstream end of the experimental samples. Gas permeability was measured using nitrogen as the permeant (pore fluid). A confining pressure of 1 MPa was applied radially to the sample, in order to ensure that no leakage occurred along its margins during measurement. The sample was then left under this confining pressure for 1 hour, to allow for any necessary microstructural equilibrium. Gas would then be flowed through the sample, whilst the volumetric flow rate $Q_v$ and the pressure differential $\nabla p$ across the sample were continuously monitored by means of customised data acquisition system and a LabVIEW program written for this purpose.

The pressure of gas entering the sample could be adjusted using a regulator attached to the permeant gas bottle. By altering the flow of gas, a range of different values of $\nabla p$ was imposed across the samples (typically between 0.001 and 0.2 MPa). Once steady-state flow was achieved, the volumetric flow rate was noted. Thus with knowledge of the gas viscosity and sample dimensions, permeability could be calculated using Equation A2. However, two scenarios can necessitate that post-measurement corrections need to be applied to the calculated values due to inertial effects: flow turbulence or gas slippage.

**A1 Non-laminar flow 1: Turbulence**

Forchheimer (1901) conducted fluid flow experiments through porous media, noting that the relationship between the pressure differential $\nabla p$ and the volumetric flow rate $Q_v$ becomes nonlinear at high fluid velocities due to flow no longer being laminar. To account for this turbulence, an inertial term, here denoted $\iota$, must be introduced, such that:

$$\frac{1}{k_{fo}} = \frac{1}{k_{gas}} - \iota \cdot Q_v \qquad (A3)$$

where $k_{fo}$ is the Forchheimer-corrected permeability value, and $k_{gas}$ is the as-measured value. In this scenario, the measured gas permeability would be lower than the true (corrected) permeability, as turbulence induces resistance to fluid flow.

**A2 Non-laminar flow 2: Gas slippage**

In his seminal 1941 paper, Klinkenberg showed that as the characteristic pore size or aperture approaches the mean free path of the permeant gas—the distance travelled between consecutive molecular collisions—interactions between the gas molecules and the pore (or crack) walls serve to reduce resistance to flow. Simply put, during liquid laminar flow, the layer of molecules adjacent to the pore (or crack) walls is static. However, for gases this molecule layer has a nonzero velocity due to molecular diffusion ("slip"). This slippage results in a higher flow rate at any given pressure differential for a gas than a liquid. Accordingly, the permeability measured using a gas would be artificially higher than if determined using a liquid.

The relationship of Klinkenberg (1941) is incorporated thus:

$$k_{gas} = k_{kl}\left(1 + \frac{b}{\bar{p}}\right) \qquad (A4)$$

where $k_{gas}$ is the as-measured permeability calculated from gas flow experiments (note that in cases where a Forchheimer correction has been applied as in Equation A3, $k_{gas}$ is substituted by $k_{fo}$ in Equation A4), $\bar{p}$ is the mean flow pressure of gas in the system, $b$ is the Klinkenberg parameter (which depends on both the gas used but also the pore structure), and $k_{kl}$ is the Klinkenberg-corrected permeability value.

In the absence of inertial effects, plotting $Q_v$ against the driving pressure (i.e. $\nabla p\bar{p}$) yields a linear relationship. Deviations from linear behaviour indicate that one or both of the inertial phenomena described above are influencing the calculated permeability. In practice, the corrected permeability can be calculated using the slope and intercept of graphs of $Q_v$ against $k_{gas}^{-1}$, and $\bar{p}^{-1}$ against $k_{fo}$. In the main body of the text measured permeability, corrected for turbulence and/or gas slippage when necessary, is always presented as $k$. Figure A1 shows example data from volcanic rocks which exhibit laminar flow, turbulence, and gas slippage, respectively. The effects of inertial flow, especially in the case of gas slippage tends to be slight, although non-negligible.

**A3 Permeability and experimental error**

Sources of error in the permeability measurements include the sample dimensions, and the resolution of the pressure transducer and flowmeters. As mentioned, permeability is determined as a function of the $Q_v - \nabla p\bar{p}$ curve, which is a series of points fit by a simple linear regression (when flow is laminar). The respective precision of the transducer and flowmeter is thus encompassed by the coefficient of determination of the regression line (i.e. its $r^2$ value). If the data are unaffected by turbulence or gas slippage, then $r^2$ is generally greater than 0.99. If flow is nonlaminar, $r^2$ tends to be appreciably lower, and the permeability is

determined using Equation A3 or A4 as appropriate. Repeat measurements suggest that experimental error is always engirdled by the symbol size when plotted graphically.

Figure A1 shows flowrate and pressure data obtained during steady-state permeability measurements on three volcanic samples. For the first example, Figure A1a - c, flow is laminar, as evident from the linear relation between the volumetric flowrate ($Q_v$) and the driving pressure ($\nabla p\bar{p}$). Accordingly, the reciprocal permeability ($k_{gas}^{-1}$) versus $Q_v$ is negative and nonlinear, as is the measured permeability $k_{gas}$ against the reciprocal mean pressure $\bar{p}^{-1}$. In the second example, flow is turbulent, and the data in Figure A1d are nonlinear. Applying the correction derived from Figure A1e (Equation A3) yields Figure A1f, where the data are randomly distributed about the mean (no Klinkenberg correction is necessary). Finally, the data shown in Figure A1g - h highlight that a Klinkenberg correction is necessary (Equation A4). However, the correction is very slight, as indeed often tends to be the case in volcanic rocks (as opposed to "tight" materials such as granite). For the LLB data presented in Table 2, Forchheimer corrections were applied to the raw values where appropriate. Permeability values were affected only slightly, increasing by a factor of between 1.03 to 1.41.

*Author contributions.* J.F., P.B., and M.H. designed the experiments, which were performed by J.F. J.F. prepared the manuscript with contributions from all authors.

*Competing interests.* The authors declare that they have no conflict of interest.

*Acknowledgements.* Thierry Reuschlé, Alex Schubnel, Luke Griffiths, and Alexandra Kushnir are thanked for inspiring discussions. Fieldwork was funded in part by the framework of the LABEX ANR-11-LABX-0050_G-EAU-THERMIE-PROFONDE and therefore benefits from a funding from the state managed by the French National Research Agency as part of the Investments for the future program. JIF acknowledges an Initiative d'Excellence (IDEX) "Contrats doctoraux" grant from the French State. MJH acknowledges Initiative d'Excellence (IDEX) Attractivité grant "VOLPERM" and a CNRS INSU grant. We are grateful to Nick Varley and Oliver Lamb for field assistance at Volcán de Colima. Fabian Wadsworth and Antonella Longo are thanked for their constructive comments on the manuscript.


[Figure]

**Figure A1.** Data obtained during permeability measurements on three rocks. a) - c) Laminar flow in a sample from Whakaari, New Zealand. d) - f) Turbulent flow, in a sample of Ruapehu andesite (New Zealand). g) - i) A small gas slippage effect in a sample of Açores trachyandesite.

Commer, M., Helwig, S.L., Hördt, A. and Tezkan, B., 2005. Interpretation of long-offset transient electromagnetic data from Mount Merapi, Indonesia, using a three-dimensional optimization approach. Journal of Geophysical Research: Solid Earth, 110(B3).

Cubellis, E., Ferri, M. and Luongo, G., 1995. Internal structures of the Campi Flegrei caldera by gravimetric data. Journal of Volcanology and Geothermal Research, 65(1), pp.147-156.

Darcy, H.P.G., 1856. Les Fontaines publiques de la ville de Dijon. Exposition et application des principes à suivre et des formules à employer dans les questions de distribution d'eau. [Exhibition and implementation of the principles to follow and to formulae employ in the issue of water distribution.] Victor Dalmont, France. (In French).

Day, S.J., 1996. Hydrothermal pore fluid pressure and the stability of porous, permeable volcanoes. Geological Society, London, Special Publications, 110(1), pp.77-93.

Delcamp, A., Roberti, G. and de Vries, B.V.W., 2016. Water in volcanoes: evolution, storage and rapid release during landslides. Bulletin of Volcanology, 78(12), p.87

Denlinger, R.P. and Hoblitt, R.P., 1999. Cyclic eruptive behavior of silicic volcanoes. Geology, 27(5), pp.459-462.

Dieterich, J.H., 1988. Growth and persistence of Hawaiian volcanic rift zones. Journal of Geophysical Research: Solid Earth, 93(B5), pp.4258-4270.

Dzurisin, D., 2003. A comprehensive approach to monitoring volcano deformation as a window on the eruption cycle. Reviews of Geophysics,41(1).

Edmonds, M. and Herd, R.A., 2007. A volcanic degassing event at the explosive-effusive transition. Geophysical Research Letters, 34(21).

Eichelberger, J. C., Carrigan, C. R., Westrich, H. R., and Price, R. H., 1986. Non-explosive silicic volcanism. Nature, 323(6089), 598-602.

Engelder, J.T., 1974. Cataclasis and the generation of fault gouge. Geological Society of America Bulletin, 85(10), pp.1515-1522.

Faoro, I., Vinciguerra, S., Marone, C., Elsworth, D. and Schubnel, A., 2013. Linking permeability to crack density evolution in thermally stressed rocks under cyclic loading. Geophysical Research Letters, 40(11), pp.2590-2595.

Farquharson, J., Heap, M.J., Baud, P., Reuschlé, T. and Varley, N.R., 2016a. Pore pressure embrittlement in a volcanic edifice. Bulletin of Volcanology,78(1), pp.1-19.

Farquharson, J., Heap, M.J., Varley, N.R., Baud, P. and Reuschlé, T., 2015. Permeability and porosity relationships of edifice-forming andesites: a combined field and laboratory study. Journal of Volcanology and Geothermal Research, 297, pp.52-68.

Farquharson, J.I., Heap, M.J. and Baud, P., 2016b. Strain-induced permeability increase in volcanic rock. Geophysical Research Letters.

Farquharson, J.I., Heap, M.J., Lavallée, Y., Varley, N.R. and Baud, P., 2016c. Evidence for the development of permeability anisotropy in lava domes and volcanic conduits. Journal of Volcanology and Geothermal Research, 323, pp.163-185.

Finn, C.A., Deszcz-Pan, M., Anderson, E.D. and John, D.A., 2007. Three-dimensional geophysical mapping of rock alteration and water content at Mount Adams, Washington: Implications for lahar hazards. Journal of Geophysical Research: Solid Earth, 112(B10).

Freeze, R.A. and Cherry, J.A., 1979. Groundwater. Englewood.

Forchheimer, P., 1901. Wasserbewegung durch boden. [Water movement through soil.] Zeitschrift des Vereines Deutscher Ingenieure, 45(1782), p.1788. (In German).

Fortin, J., Stanchits, S., Vinciguerra, S. and Guéguen, Y., 2011. Influence of thermal and mechanical cracks on permeability and elastic wave velocities in a basalt from Mt. Etna volcano subjected to elevated pressure. Tectonophysics, 503(1), pp.60-74.

Gaunt, H. E., Sammonds, P. R., Meredith, P. G., Smith, R., and Pallister, J. S., 2014. Pathways for degassing during the lava dome eruption of Mount St. Helens 2004–2008. Geology, 42(11), 947-950.

Gonnermann, H.M. and Manga, M., 2013. Dynamics of magma ascent in the volcanic conduit. Modeling Volcanic Processes: The physics and mathematics of volcanism, p.55.

Gottsmann, J., Camacho, A.G., Martí, J., Wooller, L., Fernández, J., Garcia, A. and Rymer, H., 2008. Shallow structure beneath the Central Volcanic Complex of Tenerife from new gravity data: Implications for its evolution and recent reactivation. Physics of the Earth and Planetary Interiors, 168(3), pp.212-230.

Heap, M.J., Lavallée, Y., Petrakova, L., Baud, P., Reuschle, T., Varley, N.R. and Dingwell, D.B., 2014. Microstructural controls on the physical and mechanical properties of edifice-forming andesites at Volcán de Colima, Mexico. Journal of Geophysical Research: Solid Earth, 119(4), pp.2925-2963.

Heap, M.J., Baud, P., Meredith, P.G., Vinciguerra, S., Bell, A.F. and Main, I.G., 2011. Brittle creep in basalt and its application to time-dependent volcano deformation. Earth and Planetary Science Letters, 307(1), pp.71-82.

Heap, M.J., Brantut, N., Baud, P. and Meredith, P.G., 2015c. Time-dependent compaction band formation in sandstone. Journal of Geophysical Research: Solid Earth, 120(7), pp.4808-4830.

Heap, M.J., Farquharson, J.I., Baud, P., Lavallée, Y. and Reuschlé, T., 2015a. Fracture and compaction of andesite in a volcanic edifice. Bulletin of volcanology, 77(6), pp.1-19.

Heap, M.J., Kennedy, B.M., Pernin, N., Jacquemard, L., Baud, P., Farquharson, J.I., Scheu, B., Lavallée, Y., Gilg, H.A., Letham-Brake, M. and Mayer, K., 2015b. Mechanical behaviour and failure modes in the Whakaari (White Island volcano) hydrothermal system, New Zealand. Journal of Volcanology and Geothermal Research, 295, pp.26-42.

Heap, M.J., Russell, J.K. and Kennedy, L.A., 2016. Mechanical behaviour of dacite from Mount St. Helens (USA): A link between porosity and lava dome extrusion mechanism (dome or spine)?. Journal of Volcanology and Geothermal Research.

Heap, M.J. and Wadsworth, F.B., 2016. Closing an open system: pore pressure changes in permeable edifice rock at high strain rates. Journal of Volcanology and Geothermal Research, 315, pp.40-50.

Heap, M.J., Violay, M., Wadsworth, F.B. and Vasseur, J., 2017. From rock to magma and back again: The evolution of temperature and deformation mechanism in conduit margin zones. Earth and Planetary Science Letters, 463, pp.92-100.

Heimisson, E. R., Einarsson, P., Sigmundsson, F., and Brandsdóttir, B., 2015. Kilometer-scale Kaiser effect identified in Krafla volcano, Iceland. Geophysical Research Letters, 42(19), pp.7958-7965

Hurwitz, S., Kipp, K.L., Ingebritsen, S.E. and Reid, M.E., 2003. Groundwater flow, heat transport, and water table position within

volcanic edifices: Implications for volcanic processes in the Cascade Range. Journal of Geophysical Research: Solid Earth, 108(B12).

Jónsson, G. and Stefánsson, V., 1982. Density and porosity logging in the IRDP hole, Iceland. Journal of Geophysical Research: Solid Earth, 87(B8), pp.6619-6630.

Kempter, K.A., Benner, S.G. and Williams, S.N., 1996. Rincón de la Vieja volcano, Guanacaste province, Costa Rica: geology of the southwestern flank and hazards implications. Journal of volcanology and geothermal research, 71(2), pp.109-127.

Kennedy, L.A. and Russell, J.K., 2012. Cataclastic production of volcanic ash at Mount Saint Helens. Physics and Chemistry of the Earth, Parts A/B/C, 45, pp.40-49.

Kiyama, T., Kita, H., Ishijima, Y., Yanagidani, T., Aoki, K. and Sato, T., 1996, January. Permeability in anisotropic granite under hydrostatic compression and triaxial compression including post-failure region. In 2nd North American Rock Mechanics Symposium. American Rock Mechanics Association.

Klinkenberg, L.J., 1941. The permeability of porous media to liquids and gases. Drilling and production practice. American Petroleum Institute.

Kovari, K., Tisa, A., Einstein, H.H. and Franklin, J.A., 1983. Suggested methods for determining the strength of rock materials in triaxial compression: revised version. International journal of rock mechanics and mining sciences, 20(6), pp.283-290.

Kushnir, A.R., Martel, C., Bourdier, J.L., Heap, M.J., Reuschlé, T., Erdmann, S., Komorowski, J.C. and Cholik, N., 2016. Probing permeability and microstructure: Unravelling the role of a low-permeability dome on the explosivity of Merapi (Indonesia). Journal of Volcanology and Geothermal Research, 316, pp.56-71.

Lavallée, Y., Heap, M.J., Kueppers, U., Kendrick, J.E., and Dingwell, D.B., 2017. The fragility of Volcán de Colima—a material constraint. In: Varley, N., and Komorowski, J.C. (eds) Volcán de Colima: managing the threat. Springer, Berlin.

Le Cloarec, M.F. and Gauthier, P.J., 2003. Merapi Volcano, Central Java, Indonesia: A case study of radionuclide behavior in volcanic gases and its implications for magma dynamics at andesitic volcanoes. Journal of Geophysical Research: Solid Earth, 108(B5).

Le Pennec, J.L., Hermitte, D., Dana, I., Pezard, P., Coulon, C., Cochémé, J.J., Mulyadi, E., Ollagnier, F. and Revest, C., 2001. Electrical conductivity and pore-space topology of Merapi lavas: implications for the degassing of porphyritic andesite magmas. Geophysical Research Letters, 28(22), pp.4283-4286.

Li, N., Wu, H., Feng, Q., Wang, K., Shi, Y., Li, Q. and Luo, X., 2009. Matrix porosity calculation in volcanic and dolomite reservoirs and its application. Applied Geophysics, 6(3), p.287.

Linde, A.T., Agustsson, K., Sacks, I.S. and Stefansson, R., 1993. Mechanism of the 1991 eruption of Hekla from continuous borehole strain monitoring. Nature, 365(6448), p.737.

Loaiza, S., Fortin, J., Schubnel, A., Gueguen, Y., Vinciguerra, S. and Moreira, M., 2012. Mechanical behavior and localized failure modes in a porous basalt from the Azores. Geophysical Research Letters, 39(19).

Martys, N.S., Torquato, S. and Bentz, D.P., 1994. Universal scaling of fluid permeability for sphere packings. Physical Review E, 50(1), p.403.

Massonnet, D., Briole, P. and Arnaud, A., 1995. Deflation of Mount Etna monitored by spaceborne radar interferometry. Nature, 375(6532), p.567.

McPhee, C.A. and Arthur, K.G., 1991. Klinkenberg permeability measurements: problems and practical solutions. In Advances in Core Evaluation IL Reservoir Appraisal. Proceedings of the 2nd Society of Core Analysts European Core Analysis Symposium. Gordon & Breach Science Publishers, Philadelphia (pp. 371-391).

Melnik, O. and Sparks, R.S.J., 1999. Nonlinear dynamics of lava dome extrusion. Nature, 402(6757), pp.37-41.

Melnik, O. and Sparks, R.S.J., 2005. Controls on conduit magma flow dynamics during lava dome building eruptions. Journal of Geophysical Research: Solid Earth, 110(B2).

Millett, J.M., Hole, M.J., Jolley, D.W., Schofield, N. and Campbell, E., 2016. Frontier exploration and the North Atlantic Igneous Province: new insights from a 2.6 km offshore volcanic sequence in the NE Faroe–Shetland Basin. Journal of the Geological Society, 173(2), pp.320-336.

Mitchell, T.M. and Faulkner, D.R., 2008. Experimental measurements of permeability evolution during triaxial compression of initially intact crystalline rocks and implications for fluid flow in fault zones. Journal of Geophysical Research: Solid Earth, 113(B11).

Mogi, K. 1958. Relations between the eruptions of various volcanoes and the deformation of the ground surfaces around them. Bulletin of the Earthquake Research Institute, 36, pp. 99–114

Mordecai, M., Morris, L.H. and Eng, C., 1970, January. An investigation into the changes of permeability occurring in a sandstone when failed under triaxial stress conditions. In The 12th US Symposium on Rock Mechanics (USRMS). American Rock Mechanics Association.

Mueller, S., Scheu, B., Kueppers, U., Spieler, O., Richard, D. and Dingwell, D.B., 2011. The porosity of pyroclasts as an indicator of volcanic explosivity. Journal of Volcanology and Geothermal Research, 203(3), pp.168-174.

Mueller, S., Scheu, B., Spieler, O. and Dingwell, D.B., 2008. Permeability control on magma fragmentation. Geology, 36(5), pp.399-402.

Nara, Y., Meredith, P.G., Yoneda, T. and Kaneko, K., 2011. Influence of macro-fractures and micro-fractures on permeability and elastic wave velocities in basalt at elevated pressure. Tectonophysics, 503(1), pp.52-59.

Nguyen, C. T., Gonnermann, H. M., and Houghton, B. F., 2014. Explosive to effusive transition during the largest volcanic eruption of the 20th century (Novarupta 1912, Alaska). Geology, 42(8), 703-706.

Okumura, S., and Sasaki, O., 2014. Permeability reduction of fractured rhyolite in volcanic conduits and its control on eruption cyclicity. Geology, 42(10), 843-846.

Ōmori, F., 1920. Seismographical Observations of the Fore-shocks, After-shocks, and After-outbursts of the Great Sakura jima-Eruption of 1914. Bulletin of the Imperial Earthquake Investigation Committee, 8(5), pp.353-377.

Peach, C.J. and Spiers, C.J., 1996. Influence of crystal plastic deformation on dilatancy and permeability development in synthetic salt rock. Tectonophysics, 256(1), pp.101-128.

Read, M.D., Ayling, M.R., Meredith, P.G. and Murrell, S.A., 1995. Microcracking during triaxial deformation of porous

rocks monitored by changes in rock physical properties, II. Pore volumometry and acoustic emission measurements on water-saturated rocks. Tectonophysics, 245(3-4), pp.223-235.

Regnet, J.B., David, C., Fortin, J., Robion, P., Makhloufi, Y. and Collin, P.Y., 2015. Influence of microporosity distribution on the mechanical behavior of oolithic carbonate rocks. Geomechanics for Energy and the Environment, 3, pp.11-23.

Rust, A.C., Manga, M. and Cashman, K.V., 2003. Determining flow type, shear rate and shear stress in magmas from bubble shapes and orientations. Journal of Volcanology and Geothermal Research, 122(1), pp.111-132.

Rust, A. C., Cashman, K. V., and Wallace, P. J., 2004. Magma degassing buffered by vapor flow through brecciated conduit margins. Geology, 32(4), 349-352.

Sakuma, S., Kajiwara, T., Nakada, S., Uto, K. and Shimizu, H., 2008. Drilling and logging results of USDP-4—Penetration into the volcanic conduit of Unzen Volcano, Japan. Journal of Volcanology and Geothermal Research,175(1), pp.1-12.

Schock, R.N., Heard, H.C. and Stephens, D.R., 1973. Stress-strain behavior of a granodiorite and two graywackes on compression to 20 kilobars. Journal of Geophysical Research, 78(26), pp.5922-5941.

Sepúlveda, F., Lahsen, A., Bonvalot, S., Cembrano, J., Alvarado, A. and Letelier, P., 2005. Morpho-structural evolution of the Cordón Caulle geothermal region, Southern Volcanic Zone, Chile: Insights from gravity and 40 Ar/39 Ar dating. Journal of Volcanology and geothermal Research, 148(1), pp.165-189.

Setiawan, A., 2002. Modeling of Gravity Changes on Merapi Volcano: Observed Between 1997-2000. Ph.D. thesis, Darmstadt University of Technology, Darmstadt, Germany.

Shimada, M., Ito, K. and Cho, A., 1989. Ductile behavior of a fine-grained porous basalt at room temperature and pressures to 3 GPa. Physics of the Earth and Planetary Interiors, 55(3-4), pp.361-373.

Shteynberg, G.S. and Solov'yev, T., 1976. The shape of volcanoes and the position of subordinate vents. Izvestia Earth Phys, 5, pp.83-84.

Sigmundsson, F., Pinel, V., Lund, B., Albino, F., Pagli, C., Geirsson, H. and Sturkell, E., 2010. Climate effects on volcanism: influence on magmatic systems of loading and unloading from ice mass variations, with examples from Iceland. Philosophical Transactions of the Royal Society of London A: Mathematical, Physical and Engineering Sciences, 368(1919), pp.2519-2534.

Siratovich, P.A., Heap, M.J., Villeneuve, M.C., Cole, J.W., Kennedy, B.M., Davidson, J. and Reuschlé, T., 2016. Mechanical behaviour of the Rotokawa Andesites (New Zealand): Insight into permeability evolution and stress-induced behaviour in an actively utilised geothermal reservoir. Geothermics, 64, pp.163-179.

Sparks, R.S.J., 1978. The dynamics of bubble formation and growth in magmas: a review and analysis. Journal of Volcanology and Geothermal Research, 3(1-2), pp.1-37.

Sparks, R.S.J., 1997. Causes and consequences of pressurisation in lava dome eruptions. Earth and Planetary Science Letters, 150(3), pp.177-189.

Tiede, C., Camacho, A.G., Gerstenecker, C., Fernández, J. and Suyanto, I., 2005. Modeling the density at Merapi volcano area, Indonesia, via the inverse gravimetric problem. Geochemistry, Geophysics, Geosystems, 6(9).

Ulusay, R. and Hudson, J.A., 2007. The Complete ISRM Suggested Methods for Rock Characterization, Testing and Monitoring: 1974–2006. International Society for Rock Mechanics. ISBN 978-975-93675-4-1.

Vajdova, V., Baud, P. and Wong, T.-f., 2004. Permeability evolution during localized deformation in Bentheim sandstone. Journal of Geophysical Research: Solid Earth, 109(B10).

van Wyk de Vries, B., Kerle, N. and Petley, D., 2000. Sector collapse forming at Casita volcano, Nicaragua. Geology, 28(2), pp.167-170.

van Wyk de Vries, B.V.W. and Borgia, A., 1996. The role of basement in volcano deformation. Geological Society, London, Special Publications, 110(1), pp.95-110.

van Wyk de Vries, B.V.W. and Matela, R., 1998. Styles of volcano-induced deformation: numerical models of substratum flexure, spreading and extrusion. Journal of Volcanology and Geothermal Research, 81(1), pp.1-18.

Vinciguerra, S., Trovato, C., Meredith, P.G. and Benson, P.M., 2005. Relating seismic velocities, thermal cracking and permeability in Mt. Etna and Iceland basalts. International Journal of Rock Mechanics and Mining Sciences, 42(7), pp.900-910.

Violay, M., Gibert, B., Mainprice, D., Evans, B., Dautria, J.M., Azais, P. and Pezard, P., 2012. An experimental study of the brittle-ductile transition of basalt at oceanic crust pressure and temperature conditions. Journal of Geophysical Research: Solid Earth, 117(B3).

Violay, M., Gibert, B., Mainprice, D. and Burg, J.P., 2015. Brittle versus ductile deformation as the main control of the deep fluid circulation in oceanic crust. Geophysical Research Letters, 42(8), pp.2767-2773.

Voight, B., Hoblitt, R.P., Clarke, A.B., Lockhart, A.B., Miller, A., Lynch, L. and McMahon, J., 1998. Remarkable cyclic ground deformation monitored in real-time on Montserrat, and its use in eruption forecasting. Geophysical Research Letters, 25(18), pp.3405-3408.

Wadge, G., Mattioli, G.S. and Herd, R.A., 2006. Ground deformation at Soufrière Hills Volcano, Montserrat during 1998–2000 measured by radar interferometry and GPS. Journal of Volcanology and Geothermal Research,152(1), pp.157-173.

Wadsworth, F.B., Vasseur, J., Scheu, B., Kendrick, J.E., Lavallée, Y. and Dingwell, D.B., 2016. Universal scaling of fluid permeability during volcanic welding and sediment diagenesis. Geology, 44(3), pp.219-222.

Wong, T.-f., and Baud, P., 2012. The brittle-ductile transition in porous rock: A review. Journal of Structural Geology, 44, pp.25-53.

Wong, T.-f., David, C. and Zhu, W., 1997. The transition from brittle faulting to cataclastic flow in porous sandstones: Mechanical deformation. Journal of Geophysical Research: Solid Earth, 102(B2), pp.3009-3025

Woods, A. W., and Koyaguchi, T., 1994. Transitions between explosive and effusive eruptions of silicic magmas. Nature, 370(6491), 641-644.

Xiaochun, L., Manabu, T., Zhishen, W., Hitoshi, K. and Takashi, O., 2003. Faulting-induced permeability change in Shirahama sandstone and implication for $CO_2$ aquifer storage. Chinese Journal of Rock Mechanics and Engineering, 22(6), pp.995-1001.

[revised manuscript text omitted]
. To date, physical property data at depth in volcanic materials has been obtained predominantly by researchers assessing the suitability of volcanic deposits for hydrocarbon or geothermal energy exploitation. For example Chen et al. (2016, 2017a) investigated volcanic sequences in the Junggar Basin in western China, in order to determine their suitability as gas and petroleum reservoirs. At both sites, the authors note a general decrease in porosity with depth; for example, Chen et al. (2016) report a decrease of porosity from ∼0.30 to <0.10 between the surface and 1000 m depth. Often, the interpretation of logging data from volcanic materials (e.g. Millett et al., 2016) is non-trivial due to the variable quality of density logs, and variations in the relationship between density and porosity with depth, alteration, or the intersection of distinct facies (Li et al., 2009), as well as gaps in the stratigraphic record due to incomplete core recovery. An example of a study with an excellent degree of core recovery—99.7 %—is reported by Jónsson and Stefánsson (1982): these authors calibrate porosity and density data obtained by the Iceland Research Drilling Project from a borehole cored continuously to a depth of 1919 m near Reyðarfjörður in Iceland. To combat small-scale variability arising from the intersection of discrete geological units, Jónsson and Stefánsson (1982) calculate a running average of porosity and density against depth. Notably, the average porosity decreases from 0.13 at 400 m depth to 0.06 at 1200 m depth, corresponding to an increase in bulk rock density of approximately 200 kg m$^{-3}$.

The observation that porosity tends to decrease with increasing depth is borne out by experimental deformation studies, which show that the propensity for compactant – rather than dilatant – behaviour of volcanic rock is intrinsically linked to the confining pressure under which the sample is deformed (e.g. Shimada et al., 1989; Heap et al., 2015a; Zhu et al., 2016), as well as being heavily reliant on its initial porosity (Heap et al., 2015a; Zhu et al., 2016) and other factors such as temperature (Violay et al., 2012, 2015; Heap et al., 2017) and alteration (Siratovich et al., 2016). In detail, high effective pressures and/or high initial porosity promote ductile behaviour, whereas dilatant brittle failure is favoured in low-porosity volcanic rock deformed under a range of effective pressures. For an in-depth study regarding the influence of effective pressure and porosity on the failure mode of andesite, the reader is referred to Heap et al. (2015a).

While pre-failure permeability has been explored in plutonic (Zoback and Byerlee, 1975; Kiyama et al., 1996; Mitchell and Faulkner, 2008) and volcanic (Faoro et al., 2013) rocks, studies of post-failure permeability change have been generally limited to investigations into sedimentary and synthetic materials (e.g. Mordecai and Morris, 1970; Peach and Spiers, 1996; Zhu and Wong, 1997; Regnet et al., 2015). However, a recent study explored brittle failure in compression of low- to intermediate-porosity volcanic rock and the influence of progressive stress-induced dilation (Farquharson et al., 2016b). Permeability was found to increase with

ongoing strain accumulation under triaxial conditions. With regards to the influence of inelastic compaction, research has yielded both decreases (Zhu and Wong, 1997; Baud et al., 2012; Chen et al., 2017b) and increases (Xiaochun et al., 2003) in the permeability of porous sandstone. Alam et al. (2014) investigated the permeability evolution of welded tuff from Shikotsu (Hokkaidō Prefecture, Japan), finding that permeability decreased monotonically with triaxial compression (both in the dilation and compaction regimes) and that the rate of permeability decrease was tied to the effective pressure under which deformation was performed. Pilot experiments on porous andesite – described in Heap et al. (2015a) – also indicate permeability loss as a result of inelastic compaction. Building on the work of these studies, this contribution investigates the response of the physical properties of volcanic rock – i.e. porosity and permeability – as a function of inelastic compaction under conditions anticipated in volcanic environments. Using a simplified geometry, we use a layered flow model to discuss permeability reduction as a function of compaction localisation. We then expound these results in light of the potential influence of edifice rock compaction on volcanic activity.

**2 Materials and methods**

**2.1 Sample preparation and deformation**

To assess the influence of inelastic compaction on volcanic rock permeability, a porous andesite from Volcán de Colima (Mexico) was used. The construction history, geomorphology and eruptive style of Volcán de Colima make it a useful analogue for other active andesitic stratovolcanoes around the world, such as Gunung Merapi (Indonesia), Ruapehu (New Zealand), Volcán Rincon de la Vieja (Costa Rica), Santa María (Guatemala), Tungurahua (Ecuador) and many, many more. Core samples were prepared from a block of andesite approximately 1 m$^3$, collected in May 2014 from the La Lumbre debris-flow track (barranca) on the south-western flank of the volcano. The andesite – "LLB" – is a vesicular porphyritic andesite containing subhedral phenocrysts and microphenocrysts, of unknown age. Bulk geochemical analysis is given in Table 1.

This andesite was chosen because its relatively high initial connected porosity $\phi$ means that it can be deformed in the ductile regime under pressure conditions relevant to a volcanic edifice (Heap et al., 2015a). Ten sample cores were prepared with a diameter of 20 mm, and were ground flat and parallel to a nominal length of 40 mm. Samples were dried in a vacuum oven for at least 48 hours, and the following steps carried out (adopting the protocol of Farquharson et al., 2016b):

1. Physical properties (porosity, permeability) were measured,

2. Samples were saturated, then deformed triaxially in compression under a set effective pressure to a given degree of axial strain,

3. Samples were unloaded, dried for 48 hours, and their permeability was re-measured.

Each of these stages are described in more detail hereafter. Helium pycnometry was used to measure the bulk and powder densities of LLB samples ($\rho_b$ and $\rho_p$, respectively), whilst measurements of sample dimensions allow the calculation of the bulk volume $V$, and in turn the volumetric mass density $\rho_v$. In turn, porosity (connected $\phi$, total $\phi_t$, and unconnected $\phi_u$) can be calculated:

$$\phi = \left( \frac{\rho_b - \rho_v}{\rho_v} \right) \tag{1a}$$

$$\phi_t = 1 - \left( \frac{\rho_b}{\rho_p} \right) \tag{1b}$$

$$\phi_u = \phi_t - \phi \tag{1c}$$

where a value of $2653 \pm 0.17$ kg m$^{-3}$ is used for $\rho_p$. In detail, helium pycnometry is used to determine the solid volume $V_s$ of each sample, which subsequently allows the calculation of the connected gas porosity $\phi$ as described. Automated measurements of $V_s$ were performed iteratively until five consecutive measurements yielded results within a range of 0.01% of the sample volume, so precision of the pycnometer measurements is high ($< 0.005$ cm$^3$). A greater degree of error arises when manually measuring the sample dimensions, required in order to determine $\rho_v$. Repeat measurements allowed an estimation of error in the length and diameter, which typically amount to $< 0.05$ cm$^3$ in terms of volume. Adopting the notation that $\epsilon_\phi$ is the error on the porosity calculation and that $\epsilon_x$ and $\epsilon_y$ are the independently calculated errors for measurements of $V_s$ and $V$, then the propagated error can be approximated by:

$$\epsilon_\phi = \phi \times \left[ \left( \frac{\epsilon_z}{V - V_s} \right)^2 + \left( \frac{\epsilon_y}{V} \right)^2 \right]^{\frac{1}{2}}; \; \epsilon_z = \left( \epsilon_x^2 + \epsilon_y^2 \right)^{\frac{1}{2}} \tag{2}$$

Values for $\epsilon_\phi$ are generally $< 0.005$ for the samples described herein. As such, probable error on connected gas porosity measurements is low, and always contained within the symbol size when plotted graphically.

Gas permeability was measured under steady-state conditions with a confining pressure of 1 MPa using the setup described in Farquharson et al. (2016c). Where necessary, a correction was applied to the measured permeability values to account for turbulent flow (see Forchheimer, 1901), the effects of which can become non-negligible when measuring the permeability of high-porosity media. Appendix A contains further details on the determination of permeability, the application of corrections for inertial effects, and the sources and sizes of potential error in the measurements.

**Table 1.** Major element (oxide) composition, determined via X-ray fluorescence analysis. All values are given in weight percent (wt. %).

| $SiO_2$ | $Al_2O_3$ | $Fe_2O_3$ | MnO | MgO | CaO | $Na_2O$ |
|---|---|---|---|---|---|---|
| 61.260 | 17.330 | 5.745 | 0.100 | 3.725 | 5.505 | 4.455 |
| $\pm$ 0.270 | $\pm$ 0.070 | $\pm$ 0.085 | $\pm$ 0.001 | $\pm$ 0.295 | $\pm$ 0.025 | $\pm$ 0.015 |

| $K_2O$ | $TiO_2$ | $P_2O5$ | $Cr_2O_3$ | $V_2O_5$ | NiO | LOI |
|---|---|---|---|---|---|---|
| 1.505 | 0.565 | 0.200 | 0.020 | 0.015 | 0.004 | 0.185 |
| $\pm$ 0.035 | $\pm$ 0.015 | $\pm$ 0.020 | $\pm$ 0.005 | $\pm$ 0.005 | $\pm$ 0.001 | $\pm$ 0.215 |

$SiO_2$ = Silicon dioxide; $Al_2O_3$ = Aluminium oxide; $Fe_2O_3$ = Iron oxide; MnO = Manganese(II) oxide; MgO = Magnesium oxide; CaO =Calcium oxide; $Na_2O$ = Sodium oxide; $K_2O$ = Potassium oxide; $TiO_2$ = Titanium oxide; $P_2O_5$ = Phosphorus pentoxide; $Cr_2O_3$ = Chromium(III) oxide; $V_2O_5$ = Vanadium(V) oxide; NiO = Nickel(II) oxide; LOI = loss on ignition.

Samples were then encased in a copper foil jacket (which serves to retain bulk sample cohesion after deformation), saturated with distilled water, and loaded into the triaxial deformation rig at Université de Strasbourg (see Figure 1). Throughout deformation we assume a simple effective stress law, whereby the effective confining pressure $p_{eff}$ experienced by a sample is a function of the confining pressure $p_c$ around the sample and the pressure of pore fluid $p_p$ within the sample, such that $p_{eff} = p_c - \alpha \cdot p_p$. Recent experimental work (Farquharson et al., 2016a) shows that $\alpha = 1$ is a reasonable assumption for porous andesite.

For each test, the confining and pore pressures were increased slowly until a targeted effective pressure (i.e. hydrostatic pressurisation). Note that pressure is only controlled within the connected porous network (i.e. $\phi$). However, we assume that any influence of incomplete sample saturation is negligible, due to the relatively small volume of isolated porosity $\phi_u$ in these andesites (see Table 2). Assuming a pycnometry-derived value for bulk density $\rho_b$ of approximately 2100 kg m$^{-3}$, the imposed effective pressures of 10, 30, 50, and 70 MPa are analogous to depths ranging from the upper 500 m of the edifice to greater than 3 km in depth (given that $p_c \propto \rho_b \cdot g\hat{z}$, where $g$ and $\hat{z}$ are surface gravitational acceleration and depth, respectively). The sample would then be left overnight to allow microstructural equilibrium. During the deformation experiments, a differential stress was introduced in the direction of the sample axis by advancing an axial piston (see Figure 1) under servo-control, such that the sample is subjected to a constant strain rate of $10^{-5}$ s$^{-1}$ (note that hereafter "strain" refers to axial strain unless otherwise specified). We note that – on the edifice-scale – absolute strain rates resulting from magma migration and edifice displacement are generally of the order of $10^{-7}$ s$^{-1}$ or lower, for example as estimated from borehole strain-meters: Linde et al. (1993), or from spaceborne interferometry: Massonnet et al. (1995). However, strain and strain rates are undoubtedly highly variable throughout active volcanic systems. The chosen strain rate for these experiments (the international standard in rock mechanics Kovari et al., 1983; Ulusay and Hudson, 2007) is comparable to shear rates inferred to occur along conduit margins by Rust et al. (2003). Similarly, Cashman et al. (2008) estimate strain rates of 3 - 8 $\times$ $10^{-5}$ s$^{-1}$ for the formation of fault gouge at Mount St Helens (USA). Most importantly, a strain rate of $10^{-5}$ s$^{-1}$ ensures that our samples are drained (i.e. the product of the Darcy timescale and the strain rate is $\ll 1$: Heap and Wadsworth, 2016).

Confining pressure and pore pressure were servo-controlled throughout the experiments. During hydrostatic and nonhydrostatic loading, the response of the pore fluid pump reflects variations in pore volume (see Read et al., 1995), which – normalised to the initial sample volume – corresponds to the porosity change $\Delta\phi$. For a porous material, $\Delta\phi$ can be considered equal to the volumetric strain. Indeed, the response of the confining pressure pump provides an independent estimation of the volumetric strain, found to be in perfect agreement with the inferred $\Delta\phi$ (see Baud et al., 2014, for details). When this differential $\Delta\phi$ is positive it signifies dilation (an increase in porosity) and when it is negative, it indicates compaction (a decrease in porosity). Deformation was allowed to continue for different amounts of axial strain accumulation – sample shortening relative to its original length ($\varepsilon_t$) – then unloaded. The strain recovered during the unloading phase is subtracted from the total axial strain $\varepsilon_t$ to give the inelastic (non-recoverable) strain accrued by the sample ($\varepsilon_i$). Similarly, the elastic porosity change recovered during unloading is subtracted from the porosity change at $\varepsilon_t$, to give the inelastic porosity change $\Delta\phi_i$. Samples were vacuum-dried once again and gas permeability re-measured. Samples were all deformed at room temperature. We note that in natural volcanic environments there may be some influence of temperature on rock strength (due, for instance, to the closure of cracks driven by thermal expansion). Nevertheless, a recent study by Heap et al. (2017) shows that the influence of temperature on the physical and mechanical properties of andesite may not be significant at temperatures below the glass transition $T_g$. Importantly, this study showed that the failure mode and underlying microstructural mechanism driving ductile behaviour (cataclastic pore collapse) did not change below $T_g$, which is itself largely restricted to the

[Figure]

**Figure 1.** Schematic of triaxial deformation apparatus, including confining pressure $p_c$, pore pressure $p_p$, and axial pressure $p_{ax}$ circuits. Detail of sample assemblage is shown inset. (a) axial piston; (b) "blank" endcap; (c) sample; (d) copper foil jacket; (e) nitrile jacket; (f) pore fluid distributor endcap. Directions of major $\sigma_1$ and minor $\sigma_3$ principal stresses are as shown, such that $\sigma_1 > \sigma_2 = \sigma_3$. Not to scale. Numbered valves allow various parts of each circuit to be used at any given time.

magma conduit and rock in the immediate vicinity. This is in agreement with previous studies by Vinciguerra et al. (2005) and Heap et al. (2014), both of which noted only negligible changes in microcrack density and porosity after thermally stressing volcanic materials. These authors attribute this phenomenon to the high initial crack density resulting from the complex thermal histories of volcanic rocks.

**2.2 Post-deformation permeability**

It has been shown in recent studies (e.g. Vinciguerra et al., 2005; Nara et al., 2011) that the permeability of fractured volcanic materials is influenced by the effective pressure under which it is measured: permeability tends to decrease with increasing effective pressure. As such, we acknowledge the limitation that post-deformation measurements do not represent the permeability under the deformation conditions *sensu stricto*. Nevertheless, we choose to measure permeability under the conditions described above for a host of reasons. Investigations towards determining the influence of effective pressure on properties (including permeability) other than rock strength indicate that their evolution with pressure may differ as a function of porosity, pore geometry, and other factors; which is to say that the effective pressure coefficient for a given rock property may not be the same as the Biot-Willis coefficient $\alpha$ (Bernabé et al., 1986). Given the lack of constraint on the effective pressure effect for the permeability of volcanic rocks, permeability is measured at the lowest possible confining pressure (1 MPa, rather than at "*in situ*" pressures) and without imposing a differential stress, in order to allow comparison within and between sample sets (indeed, we compare our data with compiled literature data in Sect. 4.1). This procedure also avoids the potential for creep – a mechanism of time-dependent deformation whereby subcritical crack growth induces damage and possibly even failure at stresses below the short-term strength of the rock (e.g. Heap et al., 2011; Brantut et al., 2013) – as well as precluding other phenomena such as stress relaxation that may arise when measuring permeability under a differential stress.

Measuring permeability requires that the sample dimensions, specifically length and cross-sectional area, are well constrained. Prior to initial measurements of permeability, sample dimensions are measured accurately using digital callipers. However, samples are often barrelled and thus noncylindrical after mechanical deformation, making their mean radii nontrivial to determine. Assuming that the solid volume $V_s$ remains constant throughout deformation, then the post-deformation volume is equal to the sum of solid volume, the initial pore volume, and the pore volume change after deformation. The post-deformation cross-sectional area $A_{post}$ can therefore be determined such that

$$A_{post} = \underbrace{\left[ \frac{V_s}{1 - (\phi + \Delta\phi_i)} \right]}_{V_{post}} \times \left[ \frac{1}{l_{post}} \right] \qquad (3)$$

where $V_{post}$ is the post-deformation volume and $l_{post}$ is the mean sample length after deformation.

**3 Results**

[revised manuscript text omitted]

$\phi$ = connected porosity; $\phi_u$ = unconnected porosity; $p_{eff}$ = effective pressure; $\varepsilon_t$ = target (total) axial strain; $\varepsilon_i$ = inelastic axial strain; $\Delta\phi_i$ = inelastic porosity change; $\phi + \Delta\phi_i$ = post-deformation porosity; $k_0$ = initial permeability; $k_e$ = post-deformation permeability.

**4  Discussion**

**4.1  Microstructural controls on permeability evolution**

The underlying micromechanical mechanism driving inelastic compaction in volcanic rocks has been shown to be cataclastic pore collapse (Zhu et al., 2011; Heap et al., 2015a; Zhu et al., 2016). Figure 4 illustrates this process by showing images of an intact and a deformed sample. Figure 4a is a backscattered scanning electron microscope image of an as-collected sample of LLB andesite, whereas the images in Figure 4b and 4c (from the same sample suite) are of a samples that has accumulated high strain ($> 0.20$) under an effective pressure of 30 MPa. The undeformed sample (Figure 4a) is pervasively microcracked, with highly amœboid pores ranging from $< 10$ μm to around 80 μm in diameter. Cataclastic pore collapse involves intense microcracking, which develops in a concentric damage zone around a pore. As the process of cataclasis – progressive fracturing and comminution – continues, fragments can spall into the void space, thus reducing porosity (Zhu et al., 2010). Figure 4b clearly shows abundant fractures created during triaxial deformation, both within the groundmass and crystals. In many areas, fragments have been comminuted to the micron-scale. In these samples, as observed in previous experimental studies of volcanic rock (Loaiza et al., 2012; Adelinet et al., 2013; Heap et al., 2015a), cataclastic pore collapse is localised in the form of bands traversing the sample.

The occurrence of compaction bands in andesite has been shown to correspond to periodic stress drops (Heap et al., 2015a), which are abundant in the mechanical data of Figure 2a. Our experimental data (Figure 2b and Figure 3a) show that cataclastic pore collapse progressively reduces the porosity of these andesites. If we assume that compaction is perfectly localised in our samples, we can consider a compaction band-bearing sample as a layered medium where the band of porosity $\phi_b$ is embedded within an "intact" host with the initial rock properties (which is to say, a sample of porosity $\phi$). The intact material must have a pore volume $V_1^{\phi}$ of $\phi \times (l_{intact} \times A_{post})$, where $l_{intact}$ is the overall length of the sample that is undamaged ($l_{post} - w_b$), where $w_b$ is the width of the compaction band. The deformed sample contains a pore volume $V_2^{\phi}$ of $\phi + \Delta\phi_i \times (l_{post} \times A_{post})$. From this, we can relate the compaction band porosity $\phi_b$ to the width of the compaction band:

$$\phi_b = \left[\frac{V_2^{\phi} - V_1^{\phi}}{A_{post}}\right] \times \left[\frac{1}{w_b}\right]. \tag{4}$$

Solutions for $\phi_b$ and $w_b$ are non-unique (moreover, a greater value of $w_b$ could be a function of one wide band or a number of discrete, relatively thinner bands), but we can impose a lower bound on $\phi_b$ of zero, and an upper bound equal to the post-deformation porosity of the sample: $0 \leq \phi_b < (\phi + \Delta\phi_i)$. Assuming the compaction band porosity noted by Heap et al. (2015a) ($\sim 0.10$) is typical for compacted andesite, then Equation 4 yields compaction band widths of between 1.63 and 23.57 mm (i.e. between 4 and 70 % of the overall sample length). We note that the lower end of this range is in line with the observations of Heap et al. (2015a).

We note that porosity loss is seemingly tied to the effective pressure under which compaction occurs (as observed in previous studies concerned with triaxial rock deformation, e.g. Wong et al., 1997; Baud et al., 2006; Heap et al., 2015a): for a given increment of inelastic strain, the porosity lost by a sample is greater at a higher effective pressure. This phenomenon is true both for total porosity change (Figure 2b) and for inelastic porosity loss (Figure 3a), which is to say that the inelastic compaction factor $\Delta\phi_i/\varepsilon_i$ always decreases as effective pressure increases (Baud et al., 2006).

While the mechanism of cataclastic pore collapse is governed by the pore sizes (e.g. Zhu et al., 2010, 2011), it has also been demonstrated that the local stress field around a

[Figure]

[Figure]

[Figure]

**Figure 4.** Backscattered scanning electron microscope images of LLB andesite, showing as-collected (a) and post-deformation (b - c) microstructure. Void space appears as black. Dense (metal-rich) phenocrysts appear as white or light grey within a darker grey groundmass. Both (b) and (c) are images of LLB-13, which was taken to beyond both $C^*$ and $C^{*\prime}$. Cataclastic pore collapse associated with shear-enhanced compaction is shown in (b), while (c) shows part of the dilatant shear zone marking the transition from shear-enhanced compaction to dilation.

pore increases as a function of the incumbent confining pressure (Zhu et al., 2010). A study of fault gouge formation in sandstone (Engelder, 1974) shows that fault-zone fragments are smaller when generated at higher confining pressures, and a similar effect (albeit less pronounced) was noted by Kennedy et al. (2012), who investigated fault gouge formation in dacitic dome rock. It is reasonable to assume that cataclasis may become more efficient as the local stress field increases in line with the confining pressure; in turn, a finer distribution of fragments will more readily occlude the pores around which they develop. Whether a change in the mean fragment size generated during cataclastic pore collapse underlies the observed evolution of $\Delta\phi_i/\varepsilon_i$ remains open to a targeted microstructural study. Nevertheless, we can interrogate Equation 4 to glean an idea of the effect of $\phi_b$ and $w_b$ at different effective pressures and axial strains.

Figure 5 shows the calculated $w_b$ for our experimental data as a function of inelastic axial strain. Values of $w_b$ are calculated using Equation 4, using values of $\phi_b$ of 0.20, 0.15, 0.10, and 0.05. At relatively higher imposed values of $\phi_b$ (0.20 and 0.15: Figure 5a, 5b), many of the resulting values of $w_b/l$ are non-physical (i.e. $w_b/l \not> 0$ or $w_b/l \not< 1$). However, at lower imposed values of $\phi_b$ (0.10 and 0.05: Figure 5c, 5d), values fall between 0 and 1. Moreover, there appears to be a systematic effect of $p_{eff}$, with deformation under relatively higher effective pressure yielding a higher ratio of $w_b/l$ – hence, a thicker compaction band – for any given amount of inelastic axial strain accumulation.

As would be expected (e.g. Zhu and Wong, 1997), permeability reduction follows the same general trend (Figure 3b) as porosity reduction (Figure 3a), with samples accumulating high strains showing a correspondingly large reduction in permeability. Notably, there appears to be an influence of the effective pressure under which the sample was deformed and the change in permeability for a given increment of axial strain. A difference in measured post-deformation permeability $k_e$ may be due to (1) a variation in characteristic grain size, or (2) a variation in the thickness of compaction localisation features with respect to the sample length, as described above. Moreover, one may imagine that these two factors (characteristic grain size, band thickness) operate in tandem to reduce permeability as effective pressure increases. To test this theory, we again model the deformed samples as a layered medium, such that discrete bands of uniform permeability $k_b$ and thickness $w_b$ are embedded in a medium of permeability $k_0$ (see Figure 6a). A similar approach has previously been adopted by Vajdova et al. (2004) and Baud et al. (2012) to model the permeability of sandstones containing experimentally-induced compaction bands. Fluid flow through this simplified geometry may then be modelled by assuming conservation of mass (e.g. Freeze and Cherry, 1979), such that:

$$k_e = k_0 \times \left[ \left( \frac{w_b}{l} \right) \left( \frac{k_0}{k_b} - 1 \right) + 1 \right]^{-1} \qquad (5)$$

[Figure]

**Figure 5.** Calculated sample length ratio $w_b/l$ as a function of inelastic axial strain accumulation $\varepsilon_i$. Values of $w_b$ calculated from Equation 4 using values of compaction band porosity of $\phi_b = 0.20$ (a), $\phi_b = 0.15$ (b), $\phi_b = 0.10$ (c), and $\phi_b = 0.05$ (d). Shaded area indicates range of physical values of $w_b/l$.

Variables $k_e$, $k_0$, and $l$ are already constrained, allowing us to solve for combinations of $w_b$ and $k_b$. If we assume that a compaction band comprises a granular bed (Figure 6a *inset*), we can relate its permeability $k_b$ to surface area $s$ in the following manner after Martys et al. (1994), who determined a universal scaling of permeability of a system of packed spheres:

$$k_b = \frac{2(1-\phi^*)}{s^2}\phi^{*f} \qquad (6)$$

where $\phi^* = \phi_b - \phi_c$, and $f = 4.2$ (the value of $f$ is thought to be related to the initial grain geometry: Wadsworth et al., 2016). $\phi_c$ represents the percolation threshold, taken here as 0.03. Note that the characteristic porosity is taken as the porosity within a compaction band $\phi_b$ (i.e. the porosity of the granular layer with permeability $k_b$ and width $w_b$). In turn, we can relate $s$ to a characteristic grain size (e.g. Wadsworth et al., 2016):

$$s(r) = \frac{3(1-\phi_b)}{r} \qquad (7)$$

where $r$ is the monodisperse particle radius. Note that in reality, the porosities assumed within the compaction bands here are not compatible with a monodisperse packing of spheres. Nevertheless, this greatly simplified approach gives an indication of the relative influence of the difference constituent parameters $r$, $\phi_b$, and $w_b$.

The assumed geometry is illustrated in Figure 6a, including the corresponding values of permeability and porosity for each layer. Figure 6b highlights the effects of changing either the characterisitic particle (i.e. grain) radius or the compaction band porosity. Notably – for a given porosity – a change in particle radius of 1 order of magnitude results in a change in compaction band permeability of 2 orders of magnitude. At relatively high initial porosities, a reduction in porosity by a given volume (for example, from 0.20

to 0.15) has little influence on $k_b$. On the other hand, when the porosity is low, a change in porosity of the same absolute volume (for example, from 0.10 to 0.05) exerts a much greater influence over $k_b$ (in this case, a reduction by over 2 orders of magnitude). However, the bulk sample permeability (the equivalent permeability) depends not only on the porosity of the compaction band but also its width. Figures 6c – 6f show the equivalent permeability for different values of $\phi_b$ for changing values of $w_b/l$: the ratio of the band width to the overall length of the sample. Curves are modelled by combining Equations 5, 6, and 7, using the particle radii $r$ (noted on each figure panel) and a value of $k_0 = 5.0 \times 10^{-13}$ m². As there are multiple non-unique solutions for $\phi_b$ and $w_b$, we show model results for a range of potential $\phi_b$ values in panels 6c to 6f.

Evidently, the variation in our experimental data (for example, the difference between $k_e$ of samples deformed under different effective pressures) is not explained by a systematic evolution of $r$. This suggests that while $k_b$ is very sensitive to the characteristic grain radius, the tradeoff between $\phi_b$ and $w_b$ is more important in controlling the bulk sample permeability ($k_e$). Importantly, however, idealising the geometry of a compaction band in terms of a monodisperse particle size distribution cannot accurately represent its complex porous network. More accurate values of $\phi_b$ and $s$ (and hence, a better prediction of $k_b$ and $k_e$) may be acheived by adopting a polydisperse particle size distribution or by imposing a non-spherical characteristic particle shape, for example.

An evident weakness of employing the simple layered medium model outlined above is that we assume that the only operative mechanism is porosity- and permeability-reducing. However at low strains, there is no one-to-one relationship between permeability and porosity after deformation. Rather, permeability tends to increase moderately at inelastic axial strains less than around 0.05 (i.e. a 5% shortening in sample length): while initial values of permeability tended to be

[Figure]

**Figure 6.** Model geometry and results. (a) Deformed sample may be thought of as a granular bed (with permeability $k_b$ and porosity $\phi_b$) within a sample matrix with permeability $k_0$ and porosity $\phi + \Delta\phi_i$. Inset highlights the characterisitc grain radius that governs surface area (Equation 7). (b) The permeability of a compaction band composed of packed spheres as a function of sphere radius and bed porosity. Note that in reality, porosities $< 0.26$ would require a polydisperse packing of spheres, or a granular bed composed of non-spherical grains. (c) – (f) Modelled equivalent permeability $k_e$ of a compaction band-bearing sample, plotted against the ratio of compaction band width relative to the overall sample length ($w_b/l$), for compaction band porosities of $\phi_b = 0.20$ (c), $\phi_b = 0.15$ (d), $\phi_b = 0.10$ (e), and $\phi_b = 0.05$ (f). Also plotted are data from this study (Table 2), where values of $w_b$ are calculated after Equation 4. Note that of the ten $k_e, w_b/l$ data pairs from this study, not all are plotted on each of the panels (c) to (f). This is because certain combinations of $\phi_b$ and $w_b$ yield non-physical values (Equation 4). Refer to text for further discussion.

around $5 \times 10^{-13}$ m$^2$, the measured post-deformation permeability was often greater than $10^{-12}$ m$^2$ after accumulating a small amount of axial strain (Table 2). A similar phenomenon was also observed by Loaiza et al. (2012), who noted that permeability of Açores trachyandesite increased beyond a critical stress state during hydrostatic pressurisation. This critical stress – known as $P^*$: Zhang et al. (1990) – signals the onset of lithostatic inelastic compaction; Loaiza et al. (2012) show that stress-induced cracks coalesce between collapsed pores during hydrostatic compaction, improving connectivity and, in turn, increasing permeability. Prior to deformation, the samples of porous andesite used in our experiments – LLB – contained an isolated porosity of 0.01, on average (Table 2).

Similar to the mechanism posited by Loaiza et al. (2012), we suggest that distributed microcracking during the initial stages of ductile deformation serves to interconnect this isolated porosity, creating efficient pathways for fluid flow. The mechanical data of all the experiments (Figure 1a) exhibit intermittent stress drops – even in the instances where permeability was observed to increase relative to the initial value – which suggests that compaction localisation in these andesites does not necessarily equate to the formation of an effective barrier to fluid flow. Loaiza et al. (2012), Adelinet et al. (2013), Heap et al. (2015a), and Heap et al. (2016) each examine microstructure of compaction bands formed in porous volcanic rocks (trachyandesite, basalt, andesite, and dacite, respectively). In all cases, the bands are irregular in shape and thickness, but do not necessarily constitute a contiguous surface of collapse pores (i.e. a layer of reduced porosity). This is supported by the results from a recent study by Baud et al. (2015), which examined compaction band-bearing sandstones using X-ray computed tomography. The authors show that when compaction bands are formed in a rock with porosity clusters, their path is more tortuous than in material with homogeneous porosity; consequently the bands do not comprise efficient permeability barriers in three dimensions. Notably, the porosity of the andesites deformed in our study exhibit marked heterogeneity in terms of porosity, pore shape, and pore size distribution (Figure 4a). The characteristic tortuosity of compaction bands formed in heterogeneous volcanic material – 
[revised manuscript text omitted]
, possibly due to enhanced communion. By modelling a simple sample geometry where a compaction band is represented by a packed granular bed, we show that the permeability reduction within a discrete compaction band is sensitive to the characteristic grain size. The effect of the porosity of a compaction band has a variable influence on its permeability, with changes in band porosity becoming ever more important as porosity decreases. However, the overall trade-off between the width and porosity of a compaction band are far more important that grain size in controlling fluid flow throughout the bulk of a sample. At low strains ($< 0.05$), compaction tends to result in a moderate increase in permeability (not accounted for in our model), which we suggest is a result of increased pore connectivity due to distributed microcracking. This effect is outweighed by progressive compaction at higher strains, resulting in a general trend of decreasing permeability with ongoing inelastic compaction. There exists a physical limit to compaction, which we suggest is echoed in a limit to the potential for permeability reduction in a deforming sample. Compiled data show that at high strain, porosity and permeability tend to converge towards intermediate values (i.e. $0.10 \leq \phi \leq 0.20$; $10^{-14} \leq k \leq 10^{-13}$ m$^2$). Field evidence from the literature emphasises the importance of understanding the physical and mechanical properties of rock in active volcanic environments, in particular the evolution in a rock's capacity to effectively transmit magmatic volatiles.

**6 Data availability**

Data are presented in Table 1 and 2 in the text, or are available on request to the corresponding author.

**Appendix A: Measuring permeability**

The constitutive equation governing fluid transport in porous and granular media was originally derived from experiments performed by Henry Darcy in the 1850s on the flow of water at different levels through sand. Since Darcy's work (1856), the theoretical framework of fluid transport—which is based on Newton's second law—has been well established and expanded, such that flow of gas through a porous medium may be given thus:

$$Q_v = \frac{-kA}{\mu} \frac{(p_b - p_a)}{l} \tag{A1}$$

where $\mu$ is the fluid viscosity, $Q_v$ is the volumetric flow rate, $A$ is the cross-sectional area available for flow and $l$ is the distance over which fluid flow occurs (i.e. the sample length). In a fluid transport system, flow is driven towards the region of lowest potential energy: in the special case of horizontal

flow, this may be described by a differential between a region of relatively high pressure $p_b$ to one of relatively lower pressure $p_a$: a pressure differential or pressure drop $\nabla p$. Equation A1 is valid for all porous media as long as flow is laminar (two cases of non-laminar flow are discussed hereafter). While this expression is sufficient for the case of laminar (or "streamline") flow, when considering an ideal compressible gas measured under atmospheric conditions, it becomes convenient to present gas permeability $k_{gas}$ in the following manner (Klinkenberg, 1941; McPhee and Arthur, 1991):

$$k_{gas} = \frac{Q_v \mu l \cdot p_{atm}}{A \cdot \nabla p \bar{p}} \tag{A2}$$

where $p_{atm}$ is the atmospheric pressure at which $Q_v$ is measured, and the driving pressure is given as a product of the differential pressure $\nabla p$ and the mean pressure over the sample $\bar{p}$. The mean pressure $\bar{p}$ is determined by the upstream and downstream pressures $p_b$ and $p_a$ such that $\bar{p} = (p_b + p_a)/2$. Under ambient conditions, $p_a$ is equal to the atmospheric pressure $p_{atm}$, and $p_b$ is equal to $\nabla p + p_{atm}$. The mean pressure therefore simplifies to $\bar{p} = (\nabla p + 2p_{atm})/2$.

Herein, gas permeability was measured using a steady-state permeameter using the setup described in Farquharson et al. (2016c). The apparatus is a commercial benchtop permeameter from Vinci Technologies, modified by incorporating interchangeable El-Flow volumetric mass flowmeters (from Bronkhorst) to measure the volumetric flow rate of gas at the downstream end of the experimental samples. Gas permeability was measured using nitrogen as the permeant (pore fluid). A confining pressure of 1 MPa was applied radially to the sample, in order to ensure that no leakage occurred along its margins during measurement. The sample was then left under this confining pressure for 1 hour, to allow for any necessary microstructural equilibrium. Gas would then be flowed through the sample, whilst the volumetric flow rate $Q_v$ and the pressure differential $\nabla p$ across the sample were continuously monitored by means of customised data acquisition system and a LabVIEW program written for this purpose.

The pressure of gas entering the sample could be adjusted using a regulator attached to the permeant gas bottle. By altering the flow of gas, a range of different values of $\nabla p$ was imposed across the samples (typically between 0.001 and 0.2 MPa). Once steady-state flow was achieved, the volumetric flow rate was noted. Thus with knowledge of the gas viscosity and sample dimensions, permeability could be calculated using Equation A2. However, two scenarios can necessitate that post-measurement corrections need to be applied to the calculated values due to inertial effects: flow turbulence or gas slippage.

**A1 Non-laminar flow 1: Turbulence**

Forchheimer (1901) conducted fluid flow experiments through porous media, noting that the relationship between

the pressure differential $\nabla p$ and the volumetric flow rate $Q_v$ becomes nonlinear at high fluid velocities due to flow no longer being laminar. To account for this turbulence, an inertial term, here denoted $\iota$, must be introduced, such that:

$$\frac{1}{k_{fo}} = \frac{1}{k_{gas}} - \iota \cdot Q_v \tag{A3}$$

where $k_{fo}$ is the Forchheimer-corrected permeability value, and $k_{gas}$ is the as-measured value. In this scenario, the measured gas permeability would be lower than the true (corrected) permeability, as turbulence induces resistance to fluid flow.

**A2    Non-laminar flow 2: Gas slippage**

In his seminal 1941 paper, Klinkenberg showed that as the characteristic pore size or aperture approaches the mean free path of the permeant gas—the distance travelled between consecutive molecular collisions—interactions between the gas molecules and the pore (or crack) walls serve to reduce resistance to flow. Simply put, during liquid laminar flow, the layer of molecules adjacent to the pore (or crack) walls is static. However, for gases this molecule layer has a nonzero velocity due to molecular diffusion ("slip"). This slippage results in a higher flow rate at any given pressure differential for a gas than a liquid. Accordingly, the permeability measured using a gas would be artificially higher than if determined using a liquid.

The relationship of Klinkenberg (1941) is incorporated thus:

$$k_{gas} = k_{kl}\left(1 + \frac{b}{\bar{p}}\right) \tag{A4}$$

where $k_{gas}$ is the as-measured permeability calculated from gas flow experiments (note that in cases where a Forchheimer correction has been applied as in Equation A3, $k_{gas}$ is substituted by $k_{fo}$ in Equation A4), $\bar{p}$ is the mean flow pressure of gas in the system, $b$ is the Klinkenberg parameter (which depends on both the gas used but also the pore structure), and $k_{kl}$ is the Klinkenberg-corrected permeability value.

In the absence of inertial effects, plotting $Q_v$ against the driving pressure (i.e. $\nabla p \bar{p}$) yields a linear relationship. Deviations from linear behaviour indicate that one or both of the inertial phenomena described above are influencing the calculated permeability. In practice, the corrected permeability can be calculated using the slope and intercept of graphs of $Q_v$ against $k_{gas}^{-1}$, and $\bar{p}^{-1}$ against $k_{fo}$. In the main body of the text measured permeability, corrected for turbulence and/or gas slippage when necessary, is always presented as $k$. Figure A1 shows example data from volcanic rocks which exhibit laminar flow, turbulence, and gas slippage, respectively. The effects of inertial flow, especially in the case of gas slippage tends to be slight, although non-negligible.

**A3    Permeability and experimental error**

Sources of error in the permeability measurements include the sample dimensions, and the resolution of the pressure transducer and flowmeters. As mentioned, permeability is determined as a function of the $Q_v - \nabla p \bar{p}$ curve, which is a series of points fit by a simple linear regression (when flow is laminar). The respective precision of the transducer and flowmeter is thus encompassed by the coefficient of determination of the regression line (i.e. its $r^2$ value). If the data are unaffected by turbulence or gas slippage, then $r^2$ is generally greater than 0.99. If flow is nonlaminar, $r^2$ tends to be appreciably lower, and the permeability is determined using Equation A3 or  A4 as appropriate. Repeat measurements suggest that experimental error is always engirdled by the symbol size when plotted graphically.

Figure A1 shows flowrate and pressure data obtained during steady-state permeability measurements on three volcanic samples. For the first example, Figure A1a - c, flow is laminar, as evident from the linear relation between the volumetric flowrate ($Q_v$) and the driving pressure ($\nabla p \bar{p}$). Accordingly, the reciprocal permeability ($k_{gas}^{-1}$) versus $Q_v$ is negative and nonlinear, as is the measured permeability $k_{gas}$ against the reciprocal mean pressure $\bar{p}^{-1}$. In the second example, flow is turbulent, and the data in Figure A1d are nonlinear. Applying the correction derived from Figure A1e (Equation A3) yields Figure A1f, where the data are randomly distributed about the mean (no Klinkenberg correction is necessary). Finally, the data shown in Figure A1g - h highlight that a Klinkenberg correction is necessary (Equation A4). However, the correction is very slight, as indeed often tends to be the case in volcanic rocks (as opposed to "tight" materials such as granite). For the LLB data presented in Table 2, Forchheimer corrections were applied to the raw values where appropriate. Permeability values were affected only slightly, increasing by a factor of between 1.03 to 1.41.

*Author contributions.* J.F., P.B., and M.H. designed the experiments, which were performed by J.F. J.F. prepared the manuscript with contributions from all authors.

*Competing interests.* The authors declare that they have no conflict of interest.

*Acknowledgements.* Thierry Reuschlé, Alex Schubnel, Luke Griffiths, and Alexandra Kushnir are thanked for inspiring discussions. Fieldwork was funded in part by the framework of the LABEX   ANR-11-LABX-0050_G-EAU-THERMIE-PROFONDE and therefore benefits from a funding from the state managed by the French National Research Agency as part of the Investments for the future program. JIF acknowledges an Initiative d'Excellence (IDEX) "Contrats doctoraux" grant from the French State. MJH acknowledges Initiative d'Excellence (IDEX) Attractivité grant

[Figure]

**Figure A1.** Data obtained during permeability measurements on three rocks. a) - c) Laminar flow in a sample from Whakaari, New Zealand. d) - f) Turbulent flow, in a sample of Ruapehu andesite (New Zealand). g) - i) A small gas slippage effect in a sample of Açores trachyandesite.

"VOLPERM" and a CNRS INSU grant. We are grateful to Nick Varley and Oliver Lamb for field assistance at Volcán de Colima. Fabian Wadsworth and Antonella Longo are thanked for their constructive comments on the manuscript.